# Tracking Most Significant Shifts in Nonparametric Contextual Bandits

**Joe Suk**
Columbia University
`joe.suk@columbia.edu`

**Samory Kpotufe**
Columbia University
`samory@columbia.edu`

## Abstract

We study nonparametric contextual bandits where Lipschitz mean reward functions may change over time. We first establish the minimax dynamic regret rate in this less understood setting in terms of number of changes $L$ and total-variation $V$, both capturing all changes in distribution over context space, and argue that state-of-the-art procedures are suboptimal in this setting.

Next, we tend to the question of an *adaptivity* for this setting, i.e. achieving the minimax rate without knowledge of $L$ or $V$. Quite importantly, we posit that the bandit problem, viewed locally at a given context $X_t$, should not be affected by reward changes in other parts of context space $\mathcal{X}$. We therefore propose a notion of *change*, which we term *experienced significant shifts*, that better accounts for locality, and thus counts considerably less changes than $L$ and $V$. Furthermore, similar to recent work on non-stationary MAB [Suk and Kpotufe, 2022], *experienced significant shifts* only count the most *significant* changes in mean rewards, e.g., severe best-arm changes relevant to observed contexts.

Our main result is to show that this more tolerant notion of change can in fact be adapted to.

## 1 Introduction

Contextual bandits model sequential decision making problems where the reward of a chosen action depends on an observed context $X_t$ at time $t$, e.g., a consumer's profile, a medical patient's history. The goal is to maximize the total rewards over time of chosen actions, as informed by seen contexts. As such, one suitable measure of performance is that of *dynamic regret*, which compares earned rewards to a time-varying oracle maximizing mean rewards at $X_t$. While it is often assumed in the bulk of works in this setting that rewards distributions remain stationary over time, it is understood that in practice, environmental changes induce nontrivial changes in rewards.

In fact, the problem of non-stationary environments has received a surge of attention in the simpler non-contextual Multi-Arm-Bandits (MAB) setting, while the more challenging contextual case remains ill-understood. In particular in the contextual case, some recent works [Wu et al., 2018, Luo et al., 2018, Chen et al., 2019, Wei and Luo, 2021] consider *parametric settings*, i.e. where reward functions belong to fixed parametric family, and show that one may achieve rates adaptive to an unknown number of $L$ of shifts in rewards or to a notion of total-variation $V$, both acccounting for all changes over time and context space. Instead here, we consider a much larger class of reward functions, namely Lipschitz rewards, corresponding to the natural assumption that closeby contexts have similar rewards even as reward distributions change.

As a first result for this nonparametric setting, we establish some minimax lower-bounds as a baseline in terms of either $L$ or $V$, and argue that state-of-the-art procedures for the parametric case—extended to the class of Lipschitz functions—do not achieve these baselines.

37th Conference on Neural Information Processing Systems (NeurIPS 2023).

We then turn attention to whether such baselines may be achieved *adaptively*, i.e., without knowledge of $L$ or $V$. The answer as we show is affirmative, and more importantly, some much weaker notions of change may be adapted to; for intuition, while $L$ or $V$ accounts for any change at any time over the context space (say $\mathcal{X}$), it may be that all changes are relegated to parts of the space irrelevant to observed contexts $X_t$ at the time they are played. For instance, suppose at time $t$, we observe $X_t = x_0$, then it may not make sense to count changes that happen at some other $x_1$ far from $x_0$, or changes that happened at $x_0$ itself but far back in time.

We therefore propose a new parameterization of change, termed *experienced significant shifts* that better accounts for the locality of changes in time and space, and as such may register much less changes than either $L$ or $V$. As a sanity check, we show that an oracle policy which restarts only at experienced significant shifts can attain enhanced regret rates in terms of the number $\tilde{L} = \tilde{L}(X_1, \ldots, X_T)$ of such experienced shifts (Proposition 2), a rate always no worse that the baseline we first established in terms of $L$ and $V$.

Our main result is to show that *experienced significant shifts* can be adapted to (Theorem 3), i.e., with no prior knowledge of such shifts. Importantly, the result holds in both stochastic environments, and in (oblivious) adversarial ones with no change to our notion, algorithmic approach, nor analysis. Furthermore, similar to recent advances in the non-contextual case [Abbasi-Yadkori et al., 2022, Suk and Kpotufe, 2022], an *experienced shift* is only triggered under *severe changes* such as changes of best arms locally at a context $X_t$. An added difficulty in the contextual case is that we cannot hope to observe rewards for a given arm (action) repeatedly at $X_t$ as the context may only appear once, and have to rely on carefully chosen nearby points to identify unknown shifts in reward at $X_t$.

## 1.1    Other Related Work

**Nonparametric Contextual Bandits.**    The stationary bandits with covariates (where rewards and contexts follow a joint distribution) was first introduced in a one-armed bandit problem [Woodroofe, 1979, Sarkar, 1991], with the nonparametric model first studied by Yang et al. [2002]. Minimax regret rates, based on a margin condition, were first established for the two-armed bandit in Rigollet and Zeevi [2010] and generalized to any finite number of arms in Perchet and Rigollet [2013], with further insights thereafter [Qian and Yang, 2016a,b, Reeve et al., 2018, Guan and Jiang, 2018, Hu et al., 2020, Arya and Yang, 2020, Suk and Kpotufe, 2021, Gur et al., 2022, Cai et al., 2022]. However, the mentioned works all assume a stationary distribution of rewards over contexts. Blanchard et al. [2023] studies non-stationary nonparametric contextual bandits, but in the much-different context of universal learning, concerning when sublinear regret is achievable asymptotically. Lipschitz contextual bandits also appears as part of studies on broader infinite-armed settings [Lu et al., 2009, Krishnamurthy et al., 2019]. Related, Slivkins [2014] allows for non-stationary (i.e., obliviously adversarial) environments, but only studies regret to the (per-context) best arm in hindsight. *Realizable contextual bandits* posits that the regression function capturing mean rewards in contexts lies in some known class of regressors $\mathcal{F}$, over which one can do empirical risk minimization [Foster et al., 2018, Foster and Rakhlin, 2020, Simchi-Levi and Xu, 2021]. While this setting recovers Lipschitz contextual bandits, the only applicable non-stationary guarantee to our knowledge is Wei and Luo [2021], which yields suboptimal dynamic regret (see Table 1).

**Non-Stationary Bandits and RL.**    In the simpler non-contextual bandits, changing reward distributions (a.k.a. *switching bandits*) was introduced in Garivier and Moulines [2011] and further explored with various assumptions and formulations [Karnin and Anava, 2016, Allesiardo et al., 2017, Liu et al., 2018, Wei and Srivatsva, 2018, Besbes et al., 2019, Cao et al., 2019, Mukherjee and Maillard, 2019, Besson et al., 2022]. While these earlier works focused on algorithmic design assuming knowledge of non-stationarity, such a strong assumption was removed via the *adaptive* procedures of Auer et al. [2019], Chen et al. [2019]. In followup works, Abbasi-Yadkori et al. [2022], Suk and Kpotufe [2022] show that tighter dynamic regret rates are possible, scaling only with severe changes in best arm. The ideas from non-stationary MAB were extended to various contextual bandit settings by Wu et al. [2018] (for linear mean rewards in contexts), Luo et al. [2018], Chen et al. [2019] (for finite policy classes), and Wei and Luo [2021] (for realizable mean reward functions). There have also been extensions to various reinforcement learning setups [Jaksch et al., 2010, Gajane et al., 2018, Chi Cheung et al., 2019, Ortner et al., 2020, Fei et al., 2020, Cheung et al., 2020, Touati and Vincent, 2020, Domingues et al., 2021, Mao et al., 2021, Domingues et al., 2021, Wei and Luo, 2021, Zhou et al., 2022, Lykouris et al., 2021, Wei et al., 2022, Chen and Luo, 2022, Ding and Lavaei, 2023].

Of these works, only Domingues et al. [2021] can recover Lipschitz contextaul bandits, whereupon we find their dynamic regret bounds are suboptimal (see Table 1). Again, the typical aim of the aforementioned works on contextual bandits or RL is to minimize a notion of dynamic regret in terms of the number of changes $L$ or total-variation $V$. As such, the guarantees of such works don't involve tighter notions of experienced non-stationarity, such as those studied in this work.

## 2 Problem Formulation

### 2.1 Contextual Bandits with Changing Rewards

**Preliminaries.** We assume a finite set of arms $[K] \doteq \{1, 2 \ldots, K\}$. Let $Y_t \in [0, 1]^K$ denote the vector of rewards for arms $a \in [K]$ at round $t \in [T]$ (horizon $T$), and $X_t$ the observed context at that round, lying in $\mathcal{X} \doteq [0, 1]^d$, which have joint distribution $(X_t, Y_t) \sim \mathcal{D}_t$. We let $\mathbf{X}_t \doteq \{X_s\}_{s \leq t}, \mathbf{Y}_t \doteq \{Y_s\}_{s \leq t}$ denote the observed contexts and (observed and unobserved) rewards from rounds 1 to $t$. In our setting, an oblivious adversary decides a sequence of (independent) distributions on $\{(X_t, Y_t)\}_{t \in [T]}$ before the first round.

**Notation.** *The* **reward function** $f_t : \mathcal{X} \to [0, 1]^K$ *is* $f_t^a(x) \doteq \mathbb{E}[Y_t^a | X_t = x]$, $a \in [K]$, *and captures the mean rewards of arm $a$ at context $x$ and time $t$.*

A *policy* chooses actions at each round $t$, based on observed contexts (up to round $t$) and passed rewards, whereby at each round $t$ only the reward $Y_t^a$ of the chosen action $a$ is revealed. Formally:

**Definition 1** (Policy). *A policy $\pi \doteq \{\pi_t\}_{t \in \mathbb{N}}$ is a random sequence of functions $\pi_t : \mathcal{X}^t \times [K]^{t-1} \times [0, 1]^{t-1} \to [K]$. A* **randomized** *policy $\pi_t$ maps to distributions on $[K]$, In an abuse of notation, in the context of a sequence of observations till round $t$, we'll let $\pi_t \in [K]$ denote the (possibly random) action chosen at round $t$.*

The performance of a policy is evaluated using the *dynamic regret*, defined as follows:

**Definition 2.** *Fix a context sequence $X_T$. Define the* **dynamic regret** *of a policy $\pi$, as*

$$R_T(\pi, \boldsymbol{X}_T) \doteq \sum_{t=1}^{T} \max_{a \in [K]} f_t^a(X_t) - f_t^{\pi_t}(X_t).$$

So, we aim to minimize $\mathbb{E}[R_T(\pi, \mathbf{X}_T)]$ where the expectation is over $\mathbf{X}_T, \mathbf{Y}_T$, and randomness in $\pi$.

**Notation.** *As much of our analysis focuses on the gaps in mean rewards between arms at observed contexts $X_t$, the following notation will serve useful. Let $\delta_t(a', a) \doteq f_t^{a'}(X_t) - f_t^a(X_t)$ denote the* **relative gap** *of arms $a$ to $a'$ at round $t$ at context $X_t$. Define the* **worst gap** *of arm $a$ as $\delta_t(a) \doteq \max_{a' \in [K]} \delta_t(a', a)$, corresponding to the instantaneous regret of playing $a$ at round $t$ and context $X_t$. Thus, the dynamic regret can be written as $\sum_{t \in [T]} \mathbb{E}[\delta_t(\pi_t)]$. Additionally, we will occasionally talk about the* **gap functions** *in context $x \in \mathcal{X}$. In an abuse of notation, let $\delta_t^{a', a}(x) \doteq f_t^{a'}(x) - f_t^a(x)$ and $\delta_t^a(x) \doteq \max_{a' \in [K]} \delta_t^{a', a}(x)$. Generally, when a context $x$ is not specified, as in the quantities $\delta_t(a', a), \delta_t(a)$, it should be assumed that the context in question is $X_t$.*

### 2.2 Nonparametric Setting

We assume, as in prior work on nonparametric contextual bandits [Rigollet and Zeevi, 2010, Perchet and Rigollet, 2013, Slivkins, 2014, Reeve et al., 2018, Guan and Jiang, 2018, Suk and Kpotufe, 2021], that the reward function is 1-Lipschitz.

**Assumption 1** (Lipschitz $f_t$). *For all rounds $t \in \mathbb{N}$, $a \in [K]$ and $x, x' \in \mathcal{X}$,*

$$|f_t^a(x) - f_t^a(x')| \leq \|x - x'\|_\infty. \tag{1}$$

For ease of presentation, we assume the contextual marginal distribution $\mu_X$ remains the same across rounds. Furthermore, we make a standard *strong density assumption* on $\mu_X$, which is typical in this nonparametric setting [Audibert and Tsybakov, 2007, Perchet and Rigollet, 2013, Qian and Yang, 2016a,b, Gur et al., 2022, Hu et al., 2020, Arya and Yang, 2020, Cai et al., 2022]. This holds, e.g. if $\mu_X$ has a continuous Lebesgue density on $[0, 1]^d$, and ensures good coverage of the context space.

**Assumption 2** (Strong Density Condition). *There exist $C_d, c_d > 0$ s.t. $\forall \ell_\infty$ balls $B \subset [0,1]^d$ of diameter $r \in (0,1]$:*

$$C_d \cdot r^d \geq \mu_X(B) \geq c_d \cdot r^d. \tag{2}$$

**Remark 1.** *We can in fact relax Assumption 2 so that $\mu_{X,t}(\cdot)$ is changing in $t$ and (2) is satisfied with different constants $C_{d,t}, c_{d,t}$. Our procedures in the end will not require knowledge of any $C_{d,t}, c_{d,t}$.*

### 2.3 Model Selection

A common algorithmic approach in nonparametric contextual bandits, starting from earlier work [Rigollet and Zeevi, 2010, Perchet and Rigollet, 2013], is to discretize or partition the context space $\mathcal{X}$ into *bins* where we can maintain local reward estimates. These bins have a natural hierarchical tree structure which we first elaborate.

**Definition 3** (Partition Tree). *Let $\mathcal{R} \doteq \{2^{-i} : i \in \mathbb{N} \cup \{0\}\}$, and let $\mathcal{T}_r, r \in \mathcal{R}$ denote a regular partition of $[0,1]^d$ into hypercubes (which we refer to as **bins**) of side length (a.k.a. bin size) $r$. We then define the dyadic **tree** $\mathcal{T} \doteq \{\mathcal{T}_r\}_{r \in \mathcal{R}}$, i.e., a hierarchy of nested partitions of $[0,1]^d$. We will refer to the **level** $r$ of $\mathcal{T}$ as the collection of bins in partition $\mathcal{T}_r$. The **parent** of a bin $B \in \mathcal{T}_r, r < 1$ is the bin $B' \in \mathcal{T}_{2r}$ containing $B$; **child**, **ancestor** and **descendant** relations follow naturally. We'll use $T_r(x)$ to refer to the bin at level $r$ containing $x$, while $r(B)$ is the side length of bin $B$.*

Note that, while in the above definition, $\mathcal{T}$ has infinite levels $r \in \mathcal{R}$, at any round $t$ in a procedure, we implicitly only operate on the subset of $\mathcal{T}$ containing data.

Key in securing good regret is then finding the optimal level $r \in \mathcal{R}$ of discretization (balancing local regression bias and variance), which over $T$ stationary rounds is known to be $\propto (K/T)^{\frac{1}{2+d}}$ [Rigollet and Zeevi, 2010]. The intuition for this choice is that a bias of $T^{-\frac{1}{2+d}}$ is safe to pay for $T$ rounds to maintain a minimax regret rate of $T^{\frac{1+d}{2+d}}$ (see rates in Theorem 1). We introduce the following general notation, useful later in the non-stationary problem, for associating a level with an intervals of rounds.

**Notation 1** (Level). *For $n \in \mathbb{N} \cup \{0\}$, let $r_n$ be the largest $2^{-m} \in \mathcal{R}$ such that $(K/n)^{\frac{1}{2+d}} \geq 2^{-m}$. When $I$ is an interval of rounds, we use $r_I$ as shorthand for $r_{|I|}$, where $|I|$ is the length of $I$.*

*We use $\mathcal{T}_m, T_m(x)$ as shorthand to denote (respectively) the tree $\mathcal{T}_r$ of level $r = r_m$ and the (unique) bin at level $r_m$ containing $x$.*

## 3 Results Overview

### 3.1 Minimax Lower Bounds Under Global Shifts

| | Dynamic Regret Upper Bound |
|---|---|
| ADA-ILTCB [Chen et al., 2019] | $\left(L^{1/2} \cdot T^{\frac{1+d}{2+d}}\right) \wedge \left(V_T^{1/3} \cdot T^{\frac{2+d}{3+d} + \frac{d}{3(2+d)(3+d)}}\right)$ |
| MASTER with FALCON [Wei and Luo, 2021] | $\left(L^{1/2} \cdot T^{\frac{1+d}{2+d}}\right) \wedge \left(V_T^{1/3} \cdot T^{\frac{2+d}{3+d} + \frac{d}{3(2+d)(3+d)}}\right)$ |
| KeRNS [Domingues et al., 2021] (non-adaptive) | $V_T^{1/3} T^{\frac{2+d}{3+d} + O(1/d)}$ |
| **Minimax Lower-Bound** | $\left(L^{\frac{1}{2+d}} T^{\frac{1+d}{2+d}}\right) \wedge \left(V_T^{\frac{1}{3+d}} T^{\frac{2+d}{3+d}}\right)$ |

Table 1: Existing dynamic regret upper-bounds are suboptimal in the Lipschitz setting.

As a baseline, we start with some basic lower-bounds under the simplest parametrizations of changes in rewards which have appeared in the literature, namely a *global number of shifts*, and *total variation*.

**Definition 4** (Global Number of Shifts). *Let $L \doteq \sum_{t=2}^{T} \mathbf{1}\{\exists x \in \mathcal{X}, a \in [K] : f_t^a(x) \neq f_{t-1}^a(x)\}$ be the number of* global shifts*, i.e., it counts every change in mean-reward overtime and over $\mathcal{X}$ space.*

**Definition 5** (Total Variation). *Define $V_T \doteq \sum_{t=2}^{T} \|\mathcal{D}_t - \mathcal{D}_{t-1}\|_{TV}$ where recall $\mathcal{D}_t \in \mathcal{X} \times [0,1]^K$ is the joint distribution on context and rewards at time $t$.*

We have the following initial result (for two-armed bandits) to serve as baseline for this study.

**Theorem 1** (Dynamic Regret Lower Bound). *Suppose there are $K = 2$ arms. For $V, L \in [0, T]$, let $\mathcal{P}(V, L, T)$ be the family of joint distributions $\mathcal{D} \doteq \{\mathcal{D}_t\}_{t \in [T]}$ with either total variation $V_T \leq V$ or at most $L$ global shifts. Then, there exists a constant $c > 0$ such that for any policy $\pi$:*

$$\sup_{\mathcal{D} \in \mathcal{P}(V, L, T)} \mathbb{E}_{\mathcal{D}}[R(\pi, \boldsymbol{X}_T)] \geq c \left( T^{\frac{1+d}{2+d}} + T^{\frac{2+d}{3+d}} \cdot V^{\frac{1}{3+d}} \right) \wedge \left( (L+1)^{\frac{1}{2+d}} T^{\frac{1+d}{2+d}} \right). \tag{3}$$

**Remark 2.** *Note setting $d = 0$ in Theorem 1 recovers the minimax rate $(\sqrt{T} + T^{2/3} V_T^{1/3}) \wedge \sqrt{(L+1) \cdot T}$ for non-contextual bandits [Besbes et al., 2019].*

**Achievability of Minimax Lower-Bound (3).** We are interested in whether the rates of (3) are achievable, with, or without knowledge of relevant parameters. First, we note that no existing algorithm currently guarantees a rate that matches (3). See Table 1 for a rate comparison (details for specializing to our setting found in Appendix A).

In particular, the prior adaptive works [Chen et al., 2019, Wei and Luo, 2021] both rely on the approach of randomly scheduling *replays* of stationary algorithms to detect unknown non-stationarity. However, the scheduling rate is designed to safeguard against the parametric $\sqrt{LT} \wedge V_T^{1/3} T^{2/3}$ regret rates and thus lead to suboptimal dependence on $L$ and $V_T$.

However, a simple back of the envelope calculation indicates that the rate in (3) may be attainable, at least given some distributional knowledge: a minimax-optimal stationary procedure restarted at each shift will incur regret, over $L$ equally spaced shifts, $(L+1) \cdot \left( \frac{T}{L+1} \right)^{\frac{1+d}{2+d}} \approx (L+1)^{\frac{1}{2+d}} \cdot T^{\frac{1+d}{2+d}}$.

As it turns out as we will show in the next section, (3) is indeed attainable, even adaptively; in fact, this is shown via a more optimistic problem parametrization as described next.

### 3.2 A New Problem Parametrization: Experienced Significant Shifts.

As discussed in Subsection 2.3, typical approaches [Rigollet and Zeevi, 2010, Perchet and Rigollet, 2013] in our setting discretize the context space $\mathcal{X}$ into bins, each of which is treated as an MAB instance. At a high level, our new measure of non-stationarity will trigger an **experienced significant shift** when the observed context $X_t$ arrives in a bin $B \in \mathcal{T}$ where there has been a severe change in local best arm, *w.r.t. the observed data in that bin*.

We first define a notion of **significant regret** for an arm $a \in [K]$ *locally* within a bin $B \in \mathcal{T}$. We say arm $a$ incurs **significant regret** in bin $B$ on interval $I$ if:

$$\sum_{s \in I} \delta_s(a) \cdot \mathbf{1}\{X_s \in B\} \geq \sqrt{K \cdot n_B(I)} + r(B) \cdot n_B(I), \tag{$\star$}$$

where $n_B(I) \doteq \sum_{s \in I} \mathbf{1}\{X_s \in I\}$ and recall $r(B)$ is the side length of bin $B$. The intuition for ($\star$) is as follows: suppose that, over $n$ separate rounds, we observe the same context $X_s = x_0$ in bin $B$. Then, arm $a$ would be considered unsafe in the local bandit problem at context $x_0$ if its regret exceeds $\sqrt{K \cdot n}$ (i.e., the first term on the above R.H.S.), which is a safe regret to pay for the non-contextual problem. Our broader notion ($\star$) extends this over the bin $B$ by also accounting for the bias (i.e., the second term on the above R.H.S.) of observing $X_s$ near a given context $x_0 \in B$.

**Remark 3** (Significant Regret Occurs at Critical Levels). *If ($\star$) holds for some bin $B$, then in fact it also roughly holds for the bin $B'$, with $B' \cap B \neq \emptyset$, at the critical level $r_{|I|}$ (see Notation 1) w.r.t. interval $I$ (Lemma 9 and Proposition 15). Thus, it suffices to only check ($\star$) for the critical levels $r_{|I|}$, which will be crucial in the analysis. Additionally, all such critical levels $r_{|I|}$ are above the optimal level $T^{-\frac{1}{2+d}}$ for $T$ stationary rounds (see Subsection 2.3). Thus, only $O(\log(T))$ levels play a role in in checking ($\star$) for all intervals of time $I$ and bins $B$.*

We then propose to record an *experienced significant shift* when we experience a context $X_t$, for which there is no safe arm to play in the sense of ($\star$). The following notation will be useful.

**Notation.** *For the sake of succinctly identifying regions of time with experienced significant shifts, we will conflate the closed, open, and half-closed intervals of real numbers $[a, b]$, $(a, b)$, and $[a, b)$, respectively, with the corresponding rounds contained therein, i.e. $[a, b] \equiv [a, b] \cap \mathbb{N}$.*

**Definition 6.** *Fix the context sequence $X_1, X_2, \ldots, X_T$.*

● *We say an arm $a \in [K]$ is **unsafe at context** $x \in \mathcal{X}$ **on** $I$ if there exists a bin $B \in \mathcal{T}$ containing $x$ such that arm $a$ incurs significant regret ($\star$) in bin $B$ on $I$.*

*We then have the following recursive definition:*

● *Let $\tau_0 = 1$. Define the $(i+1)$-th **experienced significant shift** as the earliest time $\tau_{i+1} \in (\tau_i, T]$ such that every arm $a \in [K]$ is unsafe at $X_t$ on some interval $I \subset [\tau_i, \tau_{i+1}]$. We refer to intervals $[\tau_i, \tau_{i+1}), i \geq 0$, as **experienced significant phases**. The unknown number of such shifts (by time $T$) is denoted $\tilde{L}$, whereby $[\tau_{\tilde{L}}, \tau_{\tilde{L}+1})$, for $\tau_{\tilde{L}+1} \doteq T + 1$, is the last phase.*

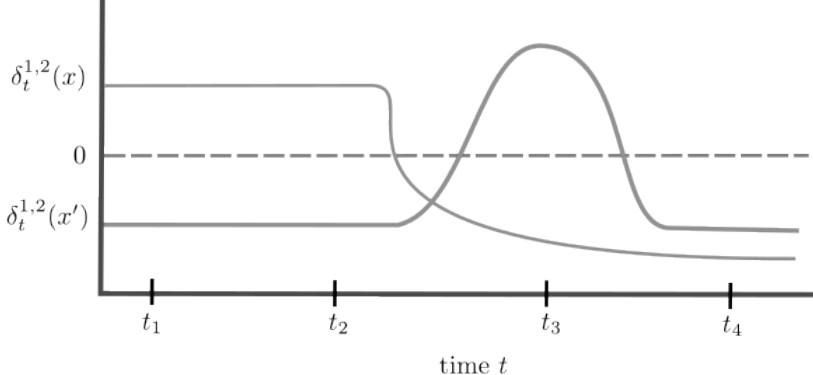

Figure 1: Shown are the gap functions $\delta_t^{1,2}(\cdot)$ ($K = 2$ arms) in time $t$ at two contexts $x, x'$ assumed to be far apart. Suppose at times $t_1$ and $t_3$, we observe context $x$, and at times $t_2$ and $t_4$, we observe $x'$. Then, the change in best arm at context $x$ is experienced. To contrast, the change in best arm at $x'$ is *not* experienced and hence not counted in Definition 6.

**Remark 4** (Definition 6 Only Counts Most Essential Changes). *It's clear from Definition 6 that only changes in mean rewards $f_t^a(X_t)$ at experienced contexts $X_t$ are counted, and only counted when experienced (see example in Figure 1).*

An experienced significant shift $\tau_i$ in fact implies a best-arm change at $X_{\tau_i}$ since, by smoothness (Assumption 1) and ($\star$), we have

$$\sum_{s \in I} \delta_s^a(X_{\tau_i}) \cdot \mathbf{1}\{X_s \in B\} \geq \sum_{s \in I} \delta_s(a) \cdot \mathbf{1}\{X_s \in B\} - r(B) \sum_{s \in I} \mathbf{1}\{X_s \in B\} > 0.$$

Thus, $\tilde{L} \leq L + 1$, the global count of shifts, and can in fact be much smaller. On the other hand, so long as an experienced significant shift does not occur, there will be arms safe to play at each context $X_t$. Thus, procedures need not restart exploration so long as unsafe arms can be quickly ruled out.

As a warmup to presenting our main regret bounds and algorithms, we'll first consider an oracle procedure which knows the experienced significant shift times $\tau_i$. The strategy will be to mimic a successive elimination algorithm, restarted at each experienced significant shift.

**Definition 7** (Oracle Procedure). *For each round $t$ in phase $[\tau_i, \tau_{i+1})$, define a good arm set $\mathcal{G}_t$ as the set of **safe** arms, i.e., arms which do not yet satisfy ($\star$) in the bin $T_r(X_t)$ at level $r = r_{\tau_{i+1}-\tau_i}$ containing $X_t$. Here, recall from Subsection 2.3 that $r_{\tau_{i+1}-\tau_i}$ is the oracle choice of level over phase $[\tau_i, \tau_{i+1})$). Then, define an **oracle procedure** $\pi$: playing a uniformly random arm $a \in \mathcal{G}_t$ at time $t$.*

We then claim such an oracle procedure attains an enhanced dynamic regret rate in terms of the significant shifts $\{\tau_i\}_i$ which recovers the minimax lower bound in terms of global shifts $L$ and total variation $V_T$ from before. The following proposition (see Appendix C for proof) captures this.

**Proposition 2** (Sanity Check). *We have the oracle procedure $\pi$ of Definition 7 satisfies with probability at least $1 - 1/T^2$ w.r.t. the randomness of $\mathbf{X}_T$: for some $C > 0$*

$$\mathbb{E}_\pi[R_T(\pi, \mathbf{X}_T) \mid \mathbf{X}_T] \leq C \log(K) \log(T) \sum_{i=0}^{\tilde{L}(\mathbf{X}_T)} (\tau_{i+1}(\mathbf{X}_T) - \tau_i(\mathbf{X}_T))^{\frac{1+d}{2+d}} \cdot K^{\frac{1}{2+d}}.$$

By Jensen's inequality the above regret rate is at most $(\tilde{L}(\mathbf{X}_T) + 1)^{\frac{1}{2+d}} \cdot T^{\frac{1+d}{2+d}} \ll (L + 1)^{\frac{1}{2+d}} \cdot T^{\frac{1+d}{2+d}}$.
At the same time, the rate is also faster than $V_T^{\frac{1}{3+d}} T^{\frac{2+d}{3+d}}$ (see Corollary 5). Thus, the oracle procedure above attains the dynamic regret lower bound of Theorem 1. We next aim to design an algorithm which can attain same order regret without knowledge of $\tau_i$ or $\tilde{L}$.

## 3.3 Main Results: Adaptive Upper-bounds

Our main result is a dynamic regret upper bound of similar order to Proposition 2 *without knowledge of the environment, e.g., the significant shift times, or the number of significant phases*. It is stated for our algorithm CMETA (Algorithm 1 of Section 4), which, for simplicity, requires knowledge of the time horizon $T$ (knowledge of $T$ removable using doubling tricks).

**Theorem 3.** *Let $\{\tau_i(\mathbf{X}_T)\}_{i=0}^{\tilde{L}+1}$ denote the unknown experienced significant shifts (Definition 6). We then have w.p. at least $1 - 1/T^2$ w.r.t. the randomness of $\mathbf{X}_T$, for some $C > 0$:*

$$\mathbb{E}[R_T(\mathsf{CMETA}, \mathbf{X}_T) \mid \mathbf{X}_T] \leq C \log(K) \log^3(T) \sum_{i=1}^{\tilde{L}(\mathbf{X}_T)} (\tau_i(\mathbf{X}_T) - \tau_{i-1}(\mathbf{X}_T))^{\frac{1+d}{2+d}} \cdot K^{\frac{1}{2+d}}.$$

**Corollary 4** (Adapting to Experienced Significant Shifts). *Under the conditions of Theorem 3, with probability at least $1 - 1/T^2$ w.r.t. the randomness in $\mathbf{X}_T$:*

$$\mathbb{E}[R_T(\mathsf{CMETA}, \mathbf{X}_T) \mid \mathbf{X}_T] \leq C \log(K) \log^3(T) \cdot (K \cdot (\tilde{L}(\mathbf{X}_T) + 1))^{\frac{1}{2+d}} \cdot T^{\frac{1+d}{2+d}}.$$

Note, this is tighter than the earlier mentioned $(L + 1)^{\frac{1}{2+d}} T^{\frac{1+d}{2+d}}$ rate. The next corollary asserts that Theorem 3 also recovers the optimal rate in terms of total-variation $V_T$.

**Corollary 5** (Adapting to Total Variation). *Under the conditions of Theorem 3:*

$$\mathbb{E}[R_T(\mathsf{CMETA}, \mathbf{X}_T)] \leq C \log(K) \log^3(T) \left( T^{\frac{1+d}{2+d}} \cdot K^{\frac{1}{2+d}} + (V_T \cdot K)^{\frac{1}{3+d}} \cdot T^{\frac{2+d}{3+d}} \right).$$

**Remark 5.** *Our regret bound can straightforwardly be generalized to $\min\{(\tilde{L} + 1)^{\frac{\beta}{2\beta+d}} \cdot T^{\frac{\beta+d}{2\beta+d}} \cdot K^{\frac{\beta}{2\beta+d}}, (V_T \cdot K)^{\frac{\beta}{3\beta+d}} \cdot T^{\frac{2\beta+d}{3\beta+d}} + T^{\frac{\beta+d}{2\beta+d}} \cdot K^{\frac{\beta}{2\beta+d}}\}$ for $\beta$-Hölder reward functions with $\beta \leq 1$, provided the notion ($\star$) is modified to take into account a bias of $r^\beta(B)$.*

## 4  Algorithm

We take a similar algorithmic approach to Suk and Kpotufe [2022], with several important modifications for our setting. The high-level strategy is to schedule multiple copies of a *base algorithm* (Algorithm 2) at random times and durations, in order to ensure updated and reliable estimation of the gaps in ($\star$). This allows fast enough detection of unknown experienced significant shifts.

**Overview of Algorithm Hierarchy.** Our main algorithm CMETA (Algorithm 1) proceeds in episodes, each of which begins by playing according to an initially scheduled base algorithm of possible duration equal to the number of rounds left till $T$. Base algorithms occasionally activate their own base algorithms of varying durations (Line 9 of Algorithm 2), called *replays*, according to a random schedule (set via variables $\{Z_{m,s}\}$). We refer to the currently playing base algorithm as the *active base algorithm*. This induces a hierarchy of base algorithms, from *parent* to *child* instances.

**Choice of Level.** Focusing on a single base algorithm now, each Base-Alg manages its own discretization of the context space $\mathcal{X} = [0, 1]^d$, corresponding to a level $r \in \mathcal{R}$ (see Definition 3). Within each bin $B \in \mathcal{T}_r$ at the level $r$, candidate arms, maintained in a set $\mathcal{A}(B)$, are evicted according to importance-weighted estimates (4) of local gaps.

As discussed in Subsection 2.3, key in algorithmic design is determining the optimal level $r \in \mathcal{R}$. An immediate difficulty is that the oracle procedure's choice of level (see Definition 7) depends on the unknown significant phase length $\tau_{i+1} - \tau_i$. To circumvent this, and as in previous works on Lipschitz contextual bandits [Perchet and Rigollet, 2013, Slivkins, 2014], we rely on an adaptive time-varying choice of level $r_t$. Specifically, each base algorithm uses the level $r_{t-t_{\text{start}}}$ based on the time elapsed since the time $t_{\text{start}}$ it was first activated.

**Managing Multiple Base Algorithms.** Instances of Base-Alg and CMETA share information, in the form of *global variables* as listed below:

- All variables defined in CMETA: $t_\ell, t, \{\mathcal{A}_{\mathrm{master}}(B)\}_{B \in \mathcal{T}}, \{Z_{m,t}\}$ (see Lines 3–6 of Algorithm 1).

- The choice of arm played at each round $t$, along with observed rewards $Y_t^a$, and the candidate arm set $\mathcal{A}_t$ which takes value the set $\mathcal{A}(B)$ of the active Base-Alg at round $t$ and bin $B = T_r(X_t)$ used.

By sharing these global variables, any Base-Alg can trigger a new episode: once an arm is evicted from a Base-Alg's $\mathcal{A}(B)$, it is also evicted from $\mathcal{A}_{\mathrm{master}}(B)$, which is essentially the candidate arm set for the current episode. A new episode is triggered at time $t$ when $\mathcal{A}_{\mathrm{master}}(B)$ becomes empty for some bin $B$ (necessarily a currently experienced bin), i.e., there is no *safe* arm left to play at the context $X_t$ in the sense of Definition 6. Note that $\mathcal{A}(B)$ are *local variables* internal to each Base-Alg (the owner of which will be clear from context in usage).

To ensure consistent behavior while using a time-varying choice of level, we enforce further regularity in arm evictions across $\mathcal{X}$: arms evicted from $\mathcal{A}(B')$ are also evicted from child bins $B \subseteq B'$ to ensure $\mathcal{A}(B) \subseteq \mathcal{A}(B')$.

---

**Algorithm 1:** **C**ontextual **M**eta-**E**limination while **T**racking **A**rms (CMETA)

---
**Input:** horizon $T$, set of arms $[K]$, tree $\mathcal{T}$ with levels $r \in \mathcal{R}$.
1 **Initialize:** round count $t \leftarrow 1$.
2 **Episode Initialization (setting global variables):**
3    $t_\ell \leftarrow t$.            // $t_\ell$ indicates start of $\ell$-th episode.
4    For each bin $B \in \mathcal{T}$, set $\mathcal{A}_{\mathrm{master}}(B) \leftarrow [K]$.    // Initialize master candidate arm sets
5    For each $m = 2, 4, \ldots, 2^{\lceil \log(T) \rceil}$ and $s = t_\ell + 1, \ldots, T$:
6       Sample and store $Z_{m,s} \sim \mathrm{Bernoulli}\left( \left(\frac{1}{m}\right)^{\frac{1}{2+d}} \cdot \left(\frac{1}{s-t_\ell}\right)^{\frac{1+d}{2+d}} \right)$.    // Set replay schedule.

7 Run Base-Alg $(t_\ell, T + 1 - t_\ell)$.
8 **if** $t < T$ **then** restart from Line 2 (i.e. start a new episode). ;

---

**Algorithm 2:** Base-Alg $(t_{\mathrm{start}}, m_0)$: Adaptively Binned Elimination with randomized arm-pulls

---
**Input**: starting round $t_{\mathrm{start}}$, scheduled duration $m_0$.
1 **Initialize**: $t \leftarrow t_{\mathrm{start}}$ For each bin $B$ at any level in $\mathcal{T}$, set $\mathcal{A}(B) \leftarrow [K]$
2 **while** $t \leq t_{\mathrm{start}} + m_0$ **do**
3    **Choose level in $\mathcal{R}$:** $r \leftarrow r_{t - t_{\mathrm{start}}}$.
4    Let $\mathcal{A}_t \leftarrow \mathcal{A}(B)$ and let $B \leftarrow T_r(X_t)$.
5    Play a random arm $a \in \mathcal{A}_t$ selected with probability $1/|\mathcal{A}_t|$.
6    Increment $t \leftarrow t + 1$.
7    **if** $\exists m$ such that $Z_{m,t} > 0$ **then**
8       Let $m \doteq \max\{m \in \{2, 4, \ldots, 2^{\lceil \log(T) \rceil}\} : Z_{m,t} > 0\}$.    // Set maximum replay length.
9       Run Base-Alg $(t, m)$.                      // Replay interrupts.
10    **Evict bad arms in bin $B$:**
11       $\mathcal{A}(B) \leftarrow \mathcal{A}(B) \backslash \{a \in [K] :$
      $\exists$ rounds $[s_1, s_2] \subseteq [t_{\mathrm{start}}, t)$ s.t. (5) holds for bin $T_{s_2 - s_1}(X_t)\}$.
12       $\mathcal{A}_{\mathrm{master}}(B) \leftarrow \mathcal{A}_{\mathrm{master}}(B) \backslash \{a \in [K] :$
      $\exists$ rounds $[s_1, s_2] \subseteq [t_\ell, t)$ s.t. (5) holds for bin $T_{s_2 - s_1}(X_t)\}$.
13    **Refine candidate arms:**       // Discard arms previously discarded in ancestor bins.
14       $\mathcal{A}(B) \leftarrow \cap_{B' \in \mathcal{T}, B \subseteq B'} \mathcal{A}(B')$.
15       $\mathcal{A}_{\mathrm{master}}(B) \leftarrow \cap_{B' \in \mathcal{T}, B \subseteq B'} \mathcal{A}_{\mathrm{master}}(B')$.
16    **Restart criterion: if** $\mathcal{A}_{\mathrm{master}}(B) = \emptyset$ for some bin $B$ **then** RETURN.;
17 RETURN.

---

**Estimating Aggregate Local Gaps.** $\sum_{s=s_1}^{s_2} \delta_s(a', a) \cdot \mathbf{1}\{X_s \in B\}$ is estimated by $\sum_{s=s_1}^{s_2} \hat{\delta}_s^B(a', a)$, where $\delta_s(a', a) \cdot \mathbf{1}\{X_s \in B\}$ is estimated by importance weighting as:

$$\hat{\delta}_s^B(a', a) \doteq |\mathcal{A}_t| \cdot \left(Y_s^{a'} \cdot \mathbf{1}\{\pi_s = a'\} - Y_s^a \cdot \mathbf{1}\{\pi_s = a\}\right) \cdot \mathbf{1}\{a \in \mathcal{A}_s\} \cdot \mathbf{1}\{X_s \in B\}. \quad (4)$$

Note that the above is an unbiased estimate of $\delta_s(a', a) \cdot \mathbf{1}\{X_s \in B\}$ whenever $a'$ and $a$ are both in $\mathcal{A}_s$ at time $s$, conditional on the context $X_s$. It then follows that, conditional on $\mathbf{X}_T$, the difference $\sum_{s=s_1}^{s_2} \left(\hat{\delta}_s^B(a', a) \cdot \mathbf{1}\{X_s \in B\} - \delta_s(a', a)\right)$ is a martingale that concentrates at a rate of order roughly $\sqrt{K \cdot n_B([s_1, s_2])}$, where recall from earlier that $n_B(I) \doteq \sum_{s \in I} \mathbf{1}\{X_s \in I\}$ is the context count in bin $B$ over interval $I$.

An arm $a$ is then evicted at round $t$ if, for some fixed $C_0 > 0$[1], $\exists$ rounds $s_1 < s_2 \le t$ such that at level $r_{s_2 - s_1}$ and letting $B \doteq T_{s_2 - s_1}(X_t)$ (i.e., the bin at level $r_{s_2 - s_1}$ containing $X_t$)

$$\max_{a' \in [K]} \sum_{s=s_1}^{s_2} \hat{\delta}_s^B(a', a) > \sqrt{C_0 \cdot K \log(T) \cdot (n_B([s_1, s_2]) \vee K \log(T))} + r_{s_2 - s_1} \cdot n_B([s_1, s_2]). \quad (5)$$

## 5 Key Technical Highlights of Analysis

While a full analysis is deferred to Appendix D, we highlight key novelties and intuitive calculations.

- **Local Safety in Bins implies Safe Total Regret.** We first argue that the notion of significant regret ($\star$) within a bin $B$ captures the total allowable regret rates $T^{\frac{1+d}{2+d}}$ we wish to compete with over $T$ "safe" rounds. If ($\star$) holds for no intervals $[s_1, s_2]$ in all bins $B$, arm $a$ would be safe and incur little regret over any $[s_1, s_2]$. As it turns out, bounding the per-bin regret by ($\star$) implies a total regret of $T^{\frac{1+d}{2+d}}$ as seen from the following rough calculation: via concentration and the strong density assumption (Assumption 2) to conflate $n_B([1, T]) \approx r(B)^d \cdot T$ and the fact that there are $\approx r^{-d}$ bins at level $r$, we have:

$$\sum_{B \in T_r} \sqrt{K \cdot n_B([1, T])} + r \cdot n_B([1, T]) \le K^{1/2} \cdot T^{1/2} \cdot r^{-d/2} + T \cdot r. \quad (6)$$

In particular taking $r \propto (K/T)^{\frac{1}{2+d}}$ makes the above R.H.S. the desired rate $K^{\frac{1}{2+d}} T^{\frac{1+d}{2+d}}$.

- **Significant Regret Threshold is Estimation Error.** At the same time, the R.H.S. of ($\star$) is a standard variance and bias bound on the regression error of estimating the cumulative regret $\sum_{s=s_1}^{s_2} \delta_s^a(x) \cdot \mathbf{1}\{X_s \in B\}$ at any context $x \in B$, conditional on $\mathbf{X}_T$ (see Lemma 6). Thus, intuitively, large gaps of magnitude above the threshold $\sqrt{K \cdot n_B(I)} + r(B) \cdot n_B(I)$ in ($\star$) are detectable via the estimates of (4).

Combining the above two points, we conclude that the notion of significant regret ($\star$) balances both (1) detection of unsafe arms and (2) regret of playing non-evicted arms. We next argue the randomized scheduling of multiple base algorithms is suitable for detecting experienced significant shifts.

- **A New Balanced Replay Scheduling.** As mentioned earlier in Subsection 3.1, previous adaptive works on contextual bandits fail to attain the optimal regret in this setting due to an inappropriate frequency of scheduling re-exploration. We introduce a novel scheduling (Line 6 of Algorithm 1) of replays which carefully balances exploration and fast detection of significant regret in the sense of ($\star$). In particular, the determined probability $(1/m)^{\frac{1}{2+d}}(1/t)^{\frac{1+d}{2+d}}$ of scheduling a new Base-Alg$(t, m)$ comes from the following intuitive calculation: a single replay of duration $m$ will, if scheduled, incur an additional regret of about $m^{\frac{1+d}{2+d}}$. Then, summing over all possible replays, the total extra regret incurred due to scheduled replays is roughly upper bounded by

$$\sum_{t=1}^{T} \sum_{m=2,4,\dots,T} \left(\frac{1}{m}\right)^{\frac{1}{2+d}} \left(\frac{1}{t}\right)^{\frac{1+d}{2+d}} \cdot m^{\frac{1+d}{2+d}} \lesssim \sum_{t=1}^{T} T^{\frac{d}{2+d}} \cdot (1/t)^{\frac{1+d}{2+d}} \lesssim T^{\frac{1+d}{2+d}}.$$

---

[1]$C_0 > 0$ needs to be sufficiently large, but is a universal constant free of the horizon $T$ or any distributional parameters.

In other words, the cost of replays only incurs extra constants in the regret. Surprisingly, we find this scheduling rate is also sufficient for detecting significant regret in *any* experienced subregion $B$ of the context space $\mathcal{X}$, i.e. there is no need to do additional exploration on a localized per-bin basis.

Key in this observation is the fact that, to detect significant regret over interval $I$ in any bin $B$, it suffices to check it at the *critical level* $r_I \propto (K/|I|)^{\frac{1}{2+d}}$, where $|I|$ is the length of $I$. In particular, a well-timed Base-Alg running on the interval $I$ will use this level $r_I$ (Line 2 of Algorithm 2) and is, thus, equipped to detect significant regret at all experienced bins over $I$.

• **Suffices to Only Check ($\star$) at Critical Levels $r_I$.**   At first glance, detecting experienced significant shifts (Definition 6) appears difficult as an arm $a$ may incur significant regret over a different bin $B'$ from the bin $B$ that is currently being used by the algorithm.

We in fact show that it suffices to only estimate the R.H.S. of ($\star$) in bins $B'$ at the critical level $r_I$ (Lemma 9 and Proposition 15). We give a rough argument for why this is the case: first, note that ($\star$) may be rewritten as

$$\frac{1}{n_B(I)} \sum_{s \in I} \delta_s(a) \cdot \mathbf{1}\{X_s \in B\} \geq \sqrt{\frac{K}{n_B(I)}} + r(B). \tag{7}$$

We next relate the two sides of the above display across different levels $r(B)$.

- By concentration and the strong density assumption (Assumption 2), the R.H.S. of (7) is in fact of order $\propto 1/\sqrt{|I| \cdot r(B)^d} + r(B)$, which is minimized at the critical level $r(B) \propto r_I$.
- The L.H.S. of (7) is an estimate of the average gap $\frac{1}{|I|} \sum_{s \in I} \delta_s^a(x)$ at any particular $x \in B$ using nearby contexts. In fact, by concentration, the two can be conflated up to error terms of order the R.H.S. of (7).

Combining the above two points, we see that if (7) holds for some bin $B$, then it will also hold for the "critical bin" $B'$, with $B' \cap B \neq \emptyset$, at the critical level $r_I \propto (K/|I|)^{\frac{1}{2+d}}$. In other words, significant regret at any experienced bin implies significant regret at the critical level, thus allowing us to constrain attention to these critical levels.

On the other hand, we observe that the calculations in (6) would hold if we only checked ($\star$) for bins $B$ at level $r_I$. Thus, it also suffices to only use the levels $r_I$ for regret minimization over intervals $I$ with no experienced significant shift.

Yet, even still, the analysis is challenging as there may be "missing data problems": arms $a \in \mathcal{A}(B)$ in contention at $B$ may have been evicted from sibling bins inside the parent $B' \supset B$ at the critical level. In other words, it is not a priori obvious how to do reliable estimation of arms $a \in \mathcal{A}(B)$ across a larger bin $B'$ which may contain sub-regions where $a$ has already been evicted. We show it is in fact possible to identify a subclass of intervals of rounds (Definition 12) and an associated class of replays (Definition 13) which can quickly evict arm $a$ in the critical bin $B'$ before there are missing data problems for bin $B \subseteq B'$. The details of this can be found in Proposition 15 of Appendix D.2.

## 6   Conclusion

We have shown that it is possible to adapt optimally to an unknown number of experienced significant shifts – a new notion introduced here – which captures severe changes in best-arm, only at observed contexts. An interesting future direction is to explore other notions of experienced shifts which may yield even more optimistic rates. For example, suppose changes in best arm occur at every round, but are localized to a sub-region $\Xi$ of the context space $\mathcal{X}$. Then, a procedure which discretizes $\mathcal{X}$ into bins at level $T^{-\frac{1}{2+d}}$ and runs local instantiations of a suitable non-stationary MAB algorithm (e.g., META of Suk and Kpotufe [2022]) can attain faster rates than those of Theorem 3 for some choices of $\Xi$. At the same time, such a strategy cannot always attain the $\tilde{L}(\mathbf{X}_T)^{\frac{1}{2+d}} \cdot T^{\frac{1+d}{2+d}}$ rate in general as using the level $T^{-\frac{1}{2+d}}$ is insufficient to detect experienced significant shifts occurring at short intervals $I$ of length $|I| \ll T$ (which require larger levels). This prompts the questions of whether there exists a unified notion of experienced shift which captures the most optimistic rates in these scenarios and whether such a notion can be adapted to.

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

# A Details for Specializing Previous Contextual Bandit Results to Lipschitz Contextual Bandits

## A.1 Finite Policy Class Contextual Bandits

In the finite policy class setting[2], one is given access to a known finite class $\Pi$ of policies $\pi : \mathcal{X} \to [K]$, and in the non-stationary variant, seeks to minimize regret to the time-varying benchmark of best policies $\pi_t^* \doteq \mathrm{argmax}_{\pi \in \Pi} \mathbb{E}_{(X,Y) \in \mathcal{D}_t}[Y(\pi(X))]$. In other words, the "dynamic regret" in this setting is defined by (for chosen policies $\{\hat{\pi}_t\}_t$)

$$\sum_{t=1}^{T} \max_{\pi \in \Pi} \mathbb{E}_{(X,Y) \in \mathcal{D}_t}[Y(\pi(X))] - \sum_{t=1}^{T} \mathbb{E}[Y_t(\hat{\pi}_t)]. \tag{8}$$

We can in fact recover the Lipschitz contextual bandit setting and relate the above to our notion of dynamic regret (Definition 2). To do so, we let $\Pi$ be the class of policies which uses a level $r \in \mathcal{R}$ and discretizes decision-making across individual bins $B \in \mathcal{T}_r$. Then, we claim there is an *oracle sequence of policies* $\{\pi_t^{\mathrm{oracle}}\}_t$ which attains the minimax regret rate of Theorem 1. So, it remains to bound the regret to the sequence $\{\pi_t^{\mathrm{oracle}}\}_t$ in the sense above.

• **Parametrizing in Terms of Global Number $L$ of Shifts.** Suppose there are $L + 1$ stationary phases of length $T/(L + 1)$. Then, we first claim there is an oracle sequence of policies $\pi_t^{\mathrm{oracle}}$ which attains reget $(L + 1)^{\frac{1}{2+d}} \cdot T^{\frac{1+d}{2+d}}$.

First, recall from Subsection 2.3 the *oracle choice of level $r_n$* for a stationary period of $n$ rounds, or the level $r_n \propto (K/n)^{\frac{1}{2+d}}$. Now, define $\{\pi_t^{\mathrm{oracle}}\}_t$ as follows: at each round $t$, $\pi_t^{\mathrm{oracle}}$ uses the oracle level $r \doteq r_{T/(L+1)} \propto \left( \frac{K(L+1)}{T} \right)^{\frac{1}{2+d}}$ and plays in each bin $B \in \mathcal{T}_r$, the arm maximizing the average reward in that bin $\mathbb{E}[f_t^a(X_t) \mid X_t \in B]$. As this is a biased version of the actual bandit problem $\{f_t^a(X_t)\}_{a \in [K]}$ at context $X_t$, it will follow that $\pi_t^{\mathrm{oracle}}$ incurs regret of order the bias of estimation in $B$ which is $r$.

Concretely, suppose $X_t$ falls in bin $B$ at level $r$, and let $\pi_t^{\mathrm{oracle}}(B)$ be the arm selected at round $t$ by $\pi_t^{\mathrm{oracle}}$ in bin $B$. Then, as mean rewards are Lipschitz, each policy $\pi_t^{\mathrm{oracle}}$ suffers regret:

$$\max_{a \in [K]} f_t^a(X_t) - f_t^{\pi_t^{\mathrm{oracle}}(B)}(X_t) \le \max_{a \in [K]} \mathbb{E}[f_t^a(X_t) - f_t^{\pi_t^{\mathrm{oracle}}(B)}(X_t) \mid X_t \in B] + r = r.$$

Thus, the sequence of policies $\{\pi_t^{\mathrm{oracle}}\}_t$ achieves dynamic regret (in the sense of Definition 2)

$$\mathbb{E}\left[ \sum_{t=1}^{T} \max_{a \in [K]} f_t^a(X_t) - f_t^{\pi_t^{\mathrm{oracle}}(X_t)}(X_t) \right] \lesssim (L+1) \cdot \left( \frac{T}{L+1} \right) \cdot \left( \frac{K}{(L+1)T} \right)^{\frac{1}{2+d}} \propto (L+1)^{\frac{1}{2+d}} \cdot T^{\frac{1+d}{2+d}}.$$

Thus, it suffices to minimize dynamic regret in the sense of (8) to this oracle policy $\pi_t^{\mathrm{oracle}}$. The state-of-the-art adaptive guarantee in this setting is that of the ADA-ILTCB algorithm of Chen et al. [2019], which achieves a dynamic regret of $\sqrt{KLT \log(|\Pi|)}$. Thus, it remains to compute $|\Pi|$.

We first observe that we need only consider levels in $\mathcal{R}$ of size at least $(K/T)^{\frac{1}{2+d}}$, which is the oracle choice of level for one stationary phase of length $T$. Thus, the size of the policy class $\Pi$ is

$$|\Pi| = \sum_{r \in \mathcal{R}} K^{r^{-d}} \propto K^{(T/K)^{\frac{d}{2+d}}} \implies \log(|\Pi|) = \left( \frac{T}{K} \right)^{\frac{d}{2+d}} \log(K).$$

Plugging this into $\sqrt{KLT \log(|\Pi|)}$ gives a regret rate of $K^{\frac{1}{2+d}} \cdot L^{1/2} T^{\frac{1+d}{2+d}}$, which has a worse dependence on the global number of shifts $L$ than the minimax optimal rate of $L^{\frac{1}{2+d}} \cdot T^{\frac{1+d}{2+d}}$ (see Theorem 1).

---

[2]While there are matters of efficiency and what offline learning guarantees may be assumed in this broader *agnostic* setting, we do not discuss these here, and readers are deferred to Langford and Zhang [2008], Dudik et al. [2011], Agarwal et al. [2014].

• **Parametrizing in Terms of Total-Variation** $V_T$. Fix any positive real number $V \in [T^{-\frac{3+d}{2+d}}, T]$. Then, the lower bound construction of Theorem 1 reveals that there exists an environment with $L + 1 = T/\Delta$ stationary phases of length $\Delta \doteq \left\lceil \left(\frac{T}{V}\right)^{\frac{2+d}{3+d}} \right\rceil$ and total-variation of order $V$.

Then, the earlier defined oracle sequence of policies $\{\pi_t^{\text{oracle}}\}_t$ attains the optimal dynamic regret rate in terms of $V_T$ (see Theorem 1) since

$$(L + 1)^{\frac{1}{2+d}} \cdot T^{\frac{1+d}{2+d}} \propto T^{\frac{2+d}{3+d}} \cdot V^{\frac{1+d}{3+d}}.$$

Meanwhile, the state-of-the-art adaptive regret guarantee in this parametrization is Theorem 2 of Chen et al. [2019], which shows ADA-ILTCB's regret bound is:

$$(K \cdot \log(|\Pi|) \cdot V)^{1/3} T^{2/3} + \sqrt{K \log(|\Pi|) \cdot T} \propto K^{\frac{2}{3(2+d)}} \cdot V^{\frac{1}{3}} \cdot T^{\frac{2+d}{3+d} + \frac{d}{3(2+d)(3+d)}} + K^{\frac{1}{2+d}} \cdot T^{\frac{1+d}{2+d}}.$$

We claim this rate is no better than our rate in Corollary 5, in all parameters $V, K, T$. For $K \geq T$, both rates imply linear regret. Assume $K < T$. Then, note by elementary calculations that for all $d \in \mathbb{N} \cup \{0\}$:

$$\frac{2}{3} + \frac{d}{3(2+d)} = \frac{2+d}{3+d} + \frac{1}{3+d} - \frac{2}{3(2+d)}.$$

Then, it follows that rate of Corollary 5 is smaller using the fact that $K < T$:

$$K^{\frac{2}{3(2+d)}} \cdot V^{1/3} \cdot T^{\frac{2}{3} + \frac{d}{3(2+d)}} \geq K^{\frac{2}{3(2+d)}} \cdot V^{\frac{1}{3+d}} \cdot T^{\frac{2+d}{3+d}} \cdot K^{\frac{1}{3+d} - \frac{2}{3(2+d)}} \geq (KV)^{\frac{1}{3+d}} \cdot T^{\frac{2+d}{3+d}}.$$

## A.2 Realizable Contextual Bandits

Lipschitz contextual bandits is also a special case of contextual bandits with *realizability*. In this broader setting, the learner is given a function class $\Phi$ which contains the true regression function $\phi_t^* : \mathcal{X} \times [K] \to [0, 1]$ describing mean rewards of context-arm pairs at round $t$. The goal is to compete with the time-varying benchmark of policies $\pi_{\phi_t^*}(x) \doteq \arg\max_{a \in [K]} \phi_t^*(x, a)$, using calls to a regression oracle over $\Phi$.

While the natural choice for $\Phi$ is the infinite class of all Lipschitz functions from $\mathcal{X} \times [K] \to [0, 1]$, the state-of-the-art non-stationary algorithm only provides guarantees for finite $\Phi$ [Wei and Luo, 2021, Appendix I.7].

However, it is still possible to recover the Lipschitz contextual bandit setting, by defining $\Phi$ similarly to how we defined the finite class of policies $\Pi$ above. Let $\Phi$ be the class of all piecewise constant functions which depends on a level $r \in \mathcal{R}$, and are constant on bins $B \in \mathcal{T}_r$ at level $r$, taking values which are multiples of $T^{-\frac{1}{2+d}}$ (there are $O(T)$ many such values in $[0, 1]$). Here, $\Phi$ is essentially the class of different discretization-based regression estimates for the true mean rewards, as appears in prior works [Rigollet and Zeevi, 2010, Perchet and Rigollet, 2013].

For this specification of $\Phi$, the realizability assumption is false. Rather, this is a mildly misspecified regression class which is allowed by the stationary guarantees of FALCON [Simchi-Levi and Xu, 2021, Section 3.2]. In particular, by smoothness, at each round $t \in [T]$ there is a function $\phi_t^* \in \Phi$ such that

$$\sup_{x \in \mathcal{X}, a \in [K]} |\phi_t^*(x, a) - f_t^a(x)| \lesssim \left(\frac{1}{T}\right)^{\frac{1}{2+d}}.$$

Specifically, we can let $\phi_t^*(x, a) \propto \mathbb{E}_X[f_t^a(X) | X \in B]$ be the smoothed version of $f_t^a(x)$ in the bin $B$ at level $T^{-\frac{1}{2+d}}$ containing $x$. Then, the above misspecification introduces an additive term in the regret bound of FALCON of order $T^{\frac{1+d}{2+d}}$ which is of the right order in our setting.

In this setting, the current state-of-the-art MASTER black-box algorithm using FALCON Simchi-Levi and Xu [2021] as a base algorithm can obtain dynamic regret upper bounded by [see Wei and Luo, 2021, Theorem 2]:

$$\min \left\{ \sqrt{\log(|\Phi|) \cdot L \cdot T}, \log^{1/3}(|\Phi|) \cdot \Delta^{1/3} \cdot T^{2/3} + \sqrt{\log(|\Phi|) \cdot T} \right\}.$$

As $\Phi$ is essentially the same size as the policy class $\Pi$ defined in the previous section, the above regret bound specializes to similar rates as those of ADA-ILTCB derived above, and are ultimately suboptimal in light of Theorem 1.

# B    Useful Lemmas

Throughout the appendix, $c_1, c_2, \ldots$ will denote universal positive constants not depending on $T, K$ or any of the significant shifts $\{\tau_i(\mathbf{X}_T)\}_i$.

## B.1    Concentration of Aggregate Gap over an Interval within a Bin

We'll first establish some concentration bounds for the local gap estimators $\hat{\delta}_s^B(a', a)$ defined in (4). For this purpose, we recall Freedman's inequality.

**Lemma 6** (Theorem 1 of Beygelzimer et al. [2011]). *Let $X_1, \ldots, X_n \in \mathbb{R}$ be a martingale difference sequence with respect to some filtration $\mathcal{F}_0, \mathcal{F}_1, \ldots$. Assume for all $t$ that $X_t \leq R$ a.s.. Then for any $\delta \in (0, 1)$, with probability at least $1 - \delta$, we have:*

$$\sum_{i=1}^n X_i \leq (e - 1) \left( \sqrt{\log(1/\delta) \sum_{i=1}^n \mathbb{E}[X_i^2 | \mathcal{F}_{i-1}]} + R \log(1/\delta) \right). \tag{9}$$

Recall from Section 4 that for round $t$, the local gap estimate $\hat{\delta}_t^B(a', a)$ in bin $B$ at round $t$ between arms $a', a$ is:

$$\hat{\delta}_t^B(a', a) \doteq |\mathcal{A}_t| \cdot (Y_t(a') \cdot \mathbf{1}\{\pi_t = a'\} - Y_t(a) \cdot \mathbf{1}\{\pi_t = a\}) \cdot \mathbf{1}\{a \in \mathcal{A}_t\} \cdot \mathbf{1}\{X_t \in B\}.$$

We next apply Lemma 6 to our aggregate estimator from Section 4.

**Proposition 7.** *With probability at least $1 - 1/T^2$ w.r.t. the randomness of $\mathbf{Y}_T, \{\pi_t\}_t \mid \mathbf{X}_T$, we have for all bins $B \in \mathcal{T}$ and rounds $s_1 < s_2$ and all arms $a \in [K]$ that for large enough $c_1 > 0$:*

$$\left| \sum_{s=s_1}^{s_2} \hat{\delta}_s^B(a', a) - \sum_{s=s_1}^{s_2} \mathbb{E}[\hat{\delta}_s^B(a', a) | \mathcal{F}_{s-1}] \right| \leq c_1 \left( \sqrt{K \log(T) \cdot n_B([s_1, s_2])} + K \log(T) \right), \tag{10}$$

*where $\mathcal{F} \doteq \{\mathcal{F}_t\}_{t=1}^T$ is the filtration with $\mathcal{F}_t$ generated by $\{\pi_s, Y_s^{\pi_s}\}_{s=1}^t$.*

*Proof.* The martingale difference $\hat{\delta}_s^B(a', a) - \mathbb{E}[\hat{\delta}_s^B(a', a) \mid \mathcal{F}_{s-1}]$ is clearly bounded above by $2K$ for all bins $B$, rounds $s$, and all arms $a, a'$. We also have a cumulative variance bound:

$$\sum_{s=s_1}^{s_2} \mathbb{E}[(\hat{\delta}_s^B(a', a))^2 \mid \mathcal{F}_{s-1}] \leq \sum_{s=s_1}^{s_2} \mathbf{1}\{X_s \in B\} \cdot |\mathcal{A}_s|^2 \cdot \mathbb{E}[\mathbf{1}\{\pi_s = a \text{ or } a'\} | \mathcal{F}_{s-1}]$$

$$\leq \sum_{s=s_1}^{s_2} \mathbf{1}\{X_s \in B\} \cdot 2|\mathcal{A}_s|$$

$$\leq 2K \cdot n_B([s_1, s_2]).$$

Then, the result follows from (9), and taking union bounds over bins $B$ (note there are at most $T$ levels and at most $T$ bins per level), arms $a, a'$, and rounds $s_1, s_2$.    $\square$

Since the error probability of Proposition 7 is negligible with respect to regret, we assume going forward in the analysis that (10) holds for all arms $a, a' \in [K]$ and rounds $s_1, s_2$. Specifically, let $\mathcal{E}_1$ be the good event over which the bounds of Proposition 7 hold for all all arms and intervals $[s_1, s_2]$.

## B.2    Concentration of Context Counts

We'll next establish concentration w.r.t. the distribution of contexts $\mathbf{X}_T$. This will ensure that all bins $B \in \mathcal{T}$ have sufficient observed data.

**Notation.** *To ease notation throughout the analysis, we'll henceforth use $\mu(\cdot)$ to refer to the context marginal distribution $\mu_X(\cdot)$.*

**Lemma 8.** *Let $\{i_t\}_{t=1}^T$ be a random sequence of arms whose distribution depends on $\mathbf{X}_T$. With probability at least $1 - 1/T^2$ w.r.t. the randomness of $\mathbf{X}_T$, we have for all bins $B \in \mathcal{T}$, all arms $a', a \in [K]$, and rounds $s_1 < s_2$, for some large enough $c_2 > 0$ the following inequalities hold:*

$$|n_B([s_1, s_2]) - (s_2 - s_1 + 1) \cdot \mu(B)| \leq c_2 \left( \log(T) + \sqrt{\log(T)\mu(B) \cdot (s_2 - s_1 + 1)} \right) \quad (11)$$

$$\left| \sum_{s=s_1}^{s_2} \delta_s(i_s, a) \cdot (\mathbf{1}\{X_s \in B\} - \mu_s(B)) \right| \leq c_2 \left( \log(T) + \sqrt{\log(T)\mu(B) \cdot (s_2 - s_1 + 1)} \right) \quad (12)$$

$$\left| \sum_{s=s_1}^{s_2} \delta_s(a) \cdot (\mathbf{1}\{X_s \in B\} - \mu_s(B)) \right| \leq c_2 \left( \log(T) + \sqrt{\log(T)\mu(B) \cdot (s_2 - s_1 + 1)} \right) \quad (13)$$

*Proof.* The first inequality (11) follow from Lemma 6 since $\sum_{s=s_1}^{s_2} \mathbf{1}\{X_s \in B\} - \mu(B)$ is a martingale, which has predictable quadratic variation is at most $(s_2 - s_1 + 1) \cdot \mu(B)$.

The other two inequalities are trickier since the corresponding sums are not necessarily martingales. Indeed, note $\delta_s(a)$ depends on $X_s$ while $\delta_s(i_s, a)$ may not even be adapted to the canonical filtration generated by $\mathbf{X}_T$ (i.e., $i_s$ may depend on $X_t$ for $t > s$). Nevertheless, we observe that for any random variable $W_s = W_s(\mathbf{X}_T) \in [-1, 1]$:

$$-(\mathbf{1}\{X_s \in B\} - \mu(B)) \leq W_s \cdot (\mathbf{1}\{X_s \in B\} - \mu(B)) \leq \mathbf{1}\{X_s \in B\} - \mu(B).$$

The upper and lower bounds above are both martingale differences with respect to the canonical filtration of $\mathbf{X}_T$ and thus, summing the above over $s$ we have via Lemma 6:

$$\left| \sum_{s=s_1}^{s_2} W_s \cdot (\mathbf{1}\{X_t \in B\} - \mu(B)) \right| \leq \left| \sum_{s=s_1}^{s_2} \mathbf{1}\{X_s \in B\} - \mu(B) \right|$$

$$\leq c_2 \left( \log(T) + \sqrt{\log(T)\mu(B) \cdot (s_2 - s_1 + 1)} \right).$$

Then, taking union bounds over rounds $s_1, s_2$, bins $B \in \mathcal{T}$, and arms $a \in [K]$ gives the result. $\qquad\square$

**Notation 2** (good event). *Recall from earlier that $\mathcal{E}_1$ is the good event over which the bounds of Proposition 7 hold for all rounds $s_1, s_2 \in [T]$ and arms $a', a \in [K]$. Thus, on $\mathcal{E}_1$, our estimated gaps in each bin will concentrate around their conditional means.*

*Let $\mathcal{E}_2$ be the good event on which bounds of Lemma 8 holds for all bins $B$, arms $a \in [K]$, rounds $s_1, s_2 \in [T]$. Thus, on $\mathcal{E}_2$, our covariate counts $n_B([s_1, s_2])$ will concentrate and we will be able to relate the empirical quantities $\sum_{s=s_1}^{s_2} \delta_s(a) \cdot \mathbf{1}\{X_s \in B\}$ with their expectations.*

Next, we establish a lemma which allows us to relate significant regret ($\star$) and thus our eviction criterion (5) between different bins and levels.

**Lemma 9** (Relating Aggregate Gaps Between Levels). *On event $\mathcal{E}_2$, if for rounds $s_1 < s_2$, bin $B'$ at level $r_{s_2 - s_1}$ and arm $a$, for some $c_3 > 0$:*

$$\sum_{s=s_1}^{s_2} \delta_s(a) \cdot \mathbf{1}\{X_s \in B'\} \leq c_3 \left( \sqrt{K \log(T) \cdot (n_{B'}([s_1, s_2]) \vee K \log(T))} + r(B') \cdot n_{B'}([s_1, s_2]) \right),$$

*then for any bin $B \subseteq B'$ and some $c_4 > 0$:*

$$\sum_{s=s_1}^{s_2} \delta_s(a) \cdot \mathbf{1}\{X_s \in B\} \leq c_4 \left( \log^{1/2}(T) \cdot r(B)^d \cdot K^{\frac{1}{2+d}} \cdot (s_2 - s_1)^{\frac{1+d}{2+d}} \right.$$

$$\left. + K \log(T) + \sqrt{\log(T)\mu(B)(s_2 - s_1 + 1)} \right).$$

*The same applies for $\delta_s(a)$ replaced with $\delta_s(a', a)$ for any fixed arm $a' \in [K]$.*

*Proof.* We have using (13) and the strong density assumption (Assumption 2):

$$\sum_{s=s_1}^{s_2} \delta_s(a) \cdot \mathbf{1}\{X_s \in B\} \leq \sum_{s=s_1}^{s_2} \delta_s(a) \cdot \mu(B)$$

$$+ c_2 \left( \log(T) + \sqrt{\log(T)\mu(B) \cdot (s_2 - s_1 + 1)} \right)$$

$$\leq \frac{C_d \cdot r(B)^d}{c_d \cdot r(B')^d} \sum_{s=s_1}^{s_2} \delta_s(a) \cdot \mu(B')$$

$$+ c_2 \left( \log(T) + \sqrt{\log(T)\mu(B) \cdot (s_2 - s_1 + 1)} \right) \qquad (14)$$

Again using (13)

$$\sum_{s=s_1}^{s_2} \delta_s(a) \cdot \mu_s(B') \leq \sum_{s=s_1}^{s_2} \delta_s(a) \cdot \mathbf{1}\{X_s \in B'\} + c_2 \left( \log(T) + \sqrt{\log(T)\mu(B') \cdot (s_2 - s_1 + 1)} \right)$$

$$\leq c_5 \left( \sqrt{K\log(T) \cdot (n_{B'}([s_1, s_2]) \vee K\log(T))} + r(B') \cdot n_{B'}([s_1, s_2]) \right.$$

$$\left. + \log(T) + \sqrt{\log(T)\mu(B') \cdot (s_2 - s_1 + 1)} \right).$$

Next, applying (11) to $n_{B'}([s_1, s_2])$ and using the strong density assumption (Assumption 2) to bound the mass $\mu(B')$ above by $C_d \cdot r(B')^d$, the above R.H.S. is further upper bounded by

$$c_6 \left( \log^{1/2}(T) K^{\frac{1+d}{2+d}} \cdot (s_2 - s_1)^{\frac{1}{2+d}} + K\log(T) \right). \qquad (15)$$

Finally, plugging (15) into (14) and using the fact that $(r(B')/2)^d \geq (K/(s_2 - s_1))^{\frac{d}{2+d}}$, we have that (14) is of the desired order. The proof of the same inequalities with $\delta_s(a', a)$ is analogous. $\qquad \square$

The following lemma relating the bias and variance terms in the notion of significant regret ($\star$) will serve useful many places in the analysis. They all follow from concentration and similar calculations via the strong density assumption (Assumption 2) as done previously.

**Lemma 10** (Relating Bias and Variance Error Terms via Strong Density Assumption). *Let $r \doteq r_{s_2 - s_1}$ for some $s_2 > s_1$. Then, on event $\mathcal{E}_2$, for any bin $B \in T_r$:*

$$\sqrt{(s_2 - s_1 + 1) \cdot \mu(B)} \leq c_7 (s_2 - s_1)^{\frac{1}{2+d}} \cdot K^{\frac{d/2}{2+d}}$$

$$\sqrt{n_B([s_1, s_2])} \leq c_8 \left( (s_2 - s_1)^{\frac{1}{2+d}} \cdot K^{\frac{d/2}{2+d}} + \log^{1/2}(T) \right.$$

$$\left. + \log^{1/4}(T) \cdot K^{\frac{d/4}{2+d}} \cdot (s_2 - s_1)^{\frac{1/2}{2+d}} \right)$$

### B.3   Useful Facts about Levels $r \in \mathcal{R}$ and their Durations in Play

The following basic facts about the level selection procedure on Line 2 of Algorithm 2 will be useful as we will decompose the analysis into the blocks, or different periods of rounds, where different levels are used.

**Definition 8.** *Namely, for $r \in \mathcal{R}$, let $s_\ell(r)$ and $e_\ell(r)$ denote the first and last rounds when level $r$ is used by the master* Base-Alg *in episode $[t_\ell, t_{\ell+1})$, i.e. rounds $t \in [t_\ell, t_{\ell+1})$ such that $r_{t-t_\ell} = r$. Then, we call $[s_\ell(r), e_\ell(r)]$ a* **block**.

The proofs of the following facts all follow from the definition of the level $r_n$ (see Notation 1) and basic calculations.

**Fact 1** (Relating Level to Interval Length). *The level $r_{s_2 - s_1} = 2^{-m}$ satisfies for $s_2 - s_1 \geq K$:*

$$2^{-(m-1)} > \left( \frac{K}{s_2 - s_1} \right)^{\frac{1}{2+d}} \geq 2^{-m},$$

*and hence*

$$K \cdot 2^{(m-1)(2+d)} < s_2 - s_1 \leq K \cdot 2^{m(2+d)}.$$

**Fact 2** (First Block). *The first block $[s_\ell(1), e_\ell(1)]$ consists of rounds $[t_\ell, t_\ell + K]$.*

**Fact 3** (Start and End Times of a Block). *For $r < 1$, the **start time** or first round $s_\ell(r)$ of the block corresponding to level $r$ in episode $[t_\ell, t_{\ell+1})$ is $s_\ell(r) = t_\ell + \left\lceil K \cdot (2r)^{-(2+d)} \right\rceil$ and the **anticipated end time**, or last round of the block if no new episode is triggered in said block, is $e_\ell(r) = t_\ell + \left\lceil K \cdot r^{-(2+d)} \right\rceil - 1$.*

**Fact 4** (Length of a Block). *Each block $[s_\ell(r), e_\ell(r)]$ is at least $K$ rounds long. For the first block $[s_\ell(1), e_\ell(1)]$, this is already clear. Otherwise, suppose $r < 1$ in which case:*

$$
\begin{aligned}
e_\ell(r) - s_\ell(r) + 1 &= \left\lceil K \cdot r^{-(2+d)} \right\rceil - \left\lceil K \cdot (2r)^{-(2+d)} \right\rceil \\
&\geq K \cdot r^{-(2+d)}(1 - 2^{-(2+d)}) - 1 \\
&\geq K \cdot 2^{2+d}(1 - 2^{-(2+d)}) - 1.
\end{aligned}
$$

*In particular, since $2^{2+d}(1 - 2^{-(2+d)}) \geq 2$ for all $d \geq 0$, we have the above is at least $K$.*

*We also have the above implies*

$$
2 \cdot (e_\ell(r) - s_\ell(r)) \geq \frac{K \cdot r^{-(2+d)} \cdot (1 - 2^{-(2+d)})}{2}.
$$

*Rearranging, this becomes for some constant $c_9 > 0$ depending only on d:*

$$
c_9^{-1} \cdot r \leq \left( \frac{K}{e_\ell(r) - s_\ell(r)} \right)^{\frac{1}{2+d}} < c_9 \cdot r.
$$

*Note we can make $c_9$ large enough so that the above also holds for level $r = 1$.*

*The above along with the definition of $r_{t-t_\ell}$ (see Notation 1) implies that the block length $e_\ell(r) - s_\ell(r)$ and the episode length $e_\ell(r) - t_\ell(r)$ up to the end of block $[s_\ell(r), e_\ell(r)]$ can be conflated up to constants*

$$
c_{10}^{-1} \cdot (e_\ell(r) - s_\ell(r)) \leq e_\ell(r) - t_\ell \leq c_{10} \cdot (e_\ell(r) - s_\ell(r)).
$$

## C   Proof of Oracle Regret Bound (Proposition 2)

Recall that $\mathcal{E}_2$ is the good event on which our covariate counts concentrate by Lemma 8. It suffices to show our desired regret bound for any fixed context sequence $\mathbf{X}_T$ on this event.

Fix a phase $[\tau_i, \tau_{i+1})$ and let $r \doteq r_{\tau_{i+1} - \tau_i}$. Fix a bin $B \in \mathcal{T}_r$ and let $\tau_i^a$ be the last round $t \in [\tau_i, \tau_{i+1})$ such that $X_t \in B$ and arm $a$ is included in $\mathcal{G}_t$. If $a$ is never excluded from $\mathcal{G}_t$ for all such $t$, let $\tau_i^a \doteq \tau_{i+1} - 1$. WLOG suppose $\tau_i^1 \leq \tau_i^2 \leq \cdots \leq \tau_i^K$. Then, letting $B'$ be the bin at level $r_{\tau_i^a - \tau_i}$ containing covariate $X_{\tau_i^a}$, we have by $(\star)$ that:

$$
\sum_{t=\tau_i}^{\tau_i^a} \delta_t(a) \cdot \mathbf{1}\{X_t \in B'\} \leq \sqrt{K \cdot n_{B'}([\tau_i, \tau_i^a])} + r(B') \cdot n_{B'}([\tau_i, \tau_i^a]).
$$

From Lemma 9, we conclude $\sum_{t=\tau_i}^{\tau_i^a} \frac{\delta_t(a)\mathbf{1}\{X_t \in B\}}{|\mathcal{G}_t|}$ is at most

$$
\frac{c_4 \left( \log^{1/2}(T) r^d K^{\frac{1}{2+d}} (\tau_{i+1} - \tau_i)^{\frac{1+d}{2+d}} + K\log(T) + \sqrt{\log(T)\mu(B)(\tau_i^a - \tau_i + 1)} \right)}{K + 1 - a},
$$

where we use the fact that $|\mathcal{G}_t| \geq K + 1 - a$ for $t \leq \tau_i^a$ such that $X_t \in B$. Summing over arms $a \in [K]$ with $\sum_{a \in [K]} \frac{1}{K+1-a} \leq \log(K)$, we obtain from summing the above over $K$:

$$
c_4 \log(K) \left( \log^{1/2}(T) r^d K^{\frac{1}{2+d}} (\tau_{i+1} - \tau_i)^{\frac{1+d}{2+d}} + K\log(T) + \sqrt{\log(T)\mu(B) \cdot (\tau_i^a - \tau_i + 1)} \right).
$$
(16)

Next, we claim that each significant phase $[\tau_i, \tau_{i+1})$ is at least $K$ rounds long or $K \leq \tau_{i+1} - \tau_i$. This follows from the definition of significant regret $(\star)$ since for $[s_1, s_2] \subseteq [\tau_i, \tau_{i+1})$:

$$
n_B([s_2, s_2]) \geq \sum_{s=s_1}^{s_2} \delta_s(a) \cdot \mathbf{1}\{X_s \in B\} \geq \sqrt{K \cdot n_B([s_1, s_2])} \implies \tau_{i+1} - \tau_i \geq n_B([s_1, s_2]) \geq K.
$$

Then $K \leq \tau_{i+1} - \tau_i$ implies (via Fact 1 about the level $r_{\tau_{i+1}-\tau_i}$)

$$\sum_{B \in \mathcal{T}_r} K \log(T) \leq K \log(T) \cdot r^{-d} \leq c_{11} \log(T) K^{\frac{2}{2+d}} (\tau_{i+1}-\tau_i)^{\frac{d}{2+d}} \leq c_{11} \log(T) K^{\frac{1}{2+d}} \cdot (\tau_{i+1}-\tau_i)^{\frac{1+d}{2+d}}.$$

Additionally, we have by Lemma 10:

$$\sqrt{(\tau_i^a - \tau_i) \cdot \mu(B)} \leq \sqrt{(\tau_{i+1}-\tau_i) \cdot \mu(B)} \leq c_7 (\tau_{i+1}-\tau_i)^{\frac{1}{2+d}} K^{\frac{d/2}{2+d}} \leq c_8 K^{\frac{1}{2+d}} (\tau_{i+1}-\tau_i)^{\frac{1+d}{2+d}}.$$

Then, plugging the above into (16) and summing over bins $B$ at level $r$, we have the regret in episode $[\tau_i, \tau_{i+1})$ is with probability at least $1 - 1/T^2$ w.r.t. the distribution of $\mathbf{X}_T$:

$$\mathbb{E}\left[ \sum_{t=\tau_i}^{\tau_{i+1}-1} \delta_t(\pi_t) \middle| \mathbf{X}_T \right] = \mathbb{E}\left[ \sum_{B \in \mathcal{T}_r} \sum_{t=\tau_i}^{\tau_{i+1}-1} \sum_{a \in \mathcal{G}_t} \frac{\delta_t(a) \cdot \mathbf{1}\{X_t \in B\}}{|\mathcal{G}_t|} \middle| \mathbf{X}_T \right]$$

$$= \mathbb{E}\left[ \sum_{B \in \mathcal{T}_r} \sum_{a \in [K]} \sum_{t=\tau_i}^{\tau_i^a} \frac{\delta_t(a) \cdot \mathbf{1}\{X_t \in B\}}{|\mathcal{G}_t|} \middle| \mathbf{X}_T \right]$$

$$\leq c_{12} \log(K) \sum_{B \in \mathcal{T}_r} \log^{1/2}(T) r^d (\tau_{i+1}-\tau_i)^{\frac{1+d}{2+d}} K^{\frac{1}{2+d}} + K \log(T)$$

$$\leq c_{13} \log(K) \log(T) \cdot (\tau_{i+1}-\tau_i)^{\frac{1+d}{2+d}} \cdot K^{\frac{1}{2+d}},$$

where we use the strong density assumption to bound $\sum_{B \in \mathcal{T}_r} r^d \leq \sum_{B \in \mathcal{T}_r} c_d^{-1} \cdot \mu(B) \leq c_d^{-1}$ in the last inequality. Summing the regret over all experienced significant phases $[\tau_i, \tau_{i+1})$ gives the desired result. $\qquad \square$

## D  Proof of CMETA Regret Upper Bound (Theorem 3)

Recall from Line 3 of Algorithm 1 that $t_\ell$ is the first round of the $\ell$-th episode. WLOG, there are $T$ total episodes and, by convention, we let $t_\ell \doteq T + 1$ if only $\ell - 1$ episodes occurred by round $T$.

We first quickly handle the simple case of $T < K$. In this case, the regret bound of Theorem 3 is vacuous since by the sub-additivity of $x \mapsto x^{\frac{1+d}{2+d}}$:

$$\sum_{i=0}^{\tilde{L}} (\tau_{i+1}-\tau_i)^{\frac{1+d}{2+d}} \cdot K^{\frac{1}{2+d}} \geq (\tau_{\tilde{L}+1} - \tau_0)^{\frac{1+d}{2+d}} \cdot K^{\frac{1}{2+d}} \geq T^{\frac{1+d}{2+d}} \cdot T^{\frac{1}{2+d}} = T.$$

Thus, it remains to show Theorem 3 for $T \geq K$.

We first transform the expected regret into a more suitable form, which will allow us to analyze regret in a similar fashion to the proof of the oracle regret bound (Appendix C).

### D.1  Decomposing the Regret

We first transform the regret into a more convenient form. Let $\mathcal{F} \doteq \{\mathcal{F}_t\}_{t=1}^T$ be the filtration with $\mathcal{F}_t$ generated by $\{\pi_s, Y_s^{\pi_s}\}_{s=1}^t$ conditional on a fixed $\mathbf{X}_T$. Then,

$$\mathbb{E}[R_T(\pi, \mathbf{X}_T) \mid \mathbf{X}_T] = \sum_{t=1}^T \mathbb{E}[\mathbb{E}[\delta_t(\pi_t) \mid \mathcal{F}_{t-1}] \mid \mathbf{X}_T]$$

$$= \sum_{t=1}^T \mathbb{E}\left[ \sum_{a \in \mathcal{A}_t} \frac{\delta_t(\pi_t)}{|\mathcal{A}_t|} \cdot \middle| \mathbf{X}_T \right]$$

$$= \mathbb{E}\left[ \sum_{t=1}^T \sum_{a \in \mathcal{A}_t} \frac{\delta_t(a)}{|\mathcal{A}_t|} \middle| \mathbf{X}_T \right].$$

Now, it suffices to bound the above R.H.S. on the good event $\mathcal{E}_1 \cap \mathcal{E}_2$ where the bounds of Lemmas 8 and 9 hold. Going forward in the rest of the analysis, we will assume said bounds hold wherever convenient.

Next, as alluded to in defining the oracle procedure (Definition 7), until the end of a significant phase $[\tau_i, \tau_{i+1})$, there is a safe arm in each bin $B$ at level $r_{\tau_{i+1} - \tau_i}$ which is experienced.

**Definition 9** (local last safe arm in each phase $a_t^\sharp$). *For a round $t \in [\tau_i, \tau_{i+1})$, let $B$ be the bin at level $r_{\tau_{i+1} - \tau_i}$ which contains $X_t$ and let $t_i(B)$ be the last round in $[\tau_i, \tau_{i+1})$ such that $X_{t_i(B)} \in B$. Then, by Definition 6, there is a **(local) last safe arm** $a_t^\sharp$ which does not yet incur significant regret in bin $B$ in the following sense: for all $[s_1, s_2] \subseteq [\tau_i, t_i(B)]$ letting $r = r_{s_2 - s_1}$ and $B' \in \mathcal{T}_r$ such that $B' \supseteq B$ we have:*

$$\sum_{s=s_1}^{s_2} \delta_s(a_t^\sharp) \cdot \mathbf{1}\{X_s \in B'\} < \sqrt{K \cdot n_{B'}([s_1, s_2])} + r \cdot n_{B'}([s_1, s_2]).$$

**Remark 6.** *The local last safe arms $\{a_t^\sharp\}_t$ only depend on the distribution of $X_T$ and **not** on the realized rewards $Y_T$. In particular, the sequence $\{a_t^\sharp\}_t$ is fixed conditional on $X_T$.*

We first decompose the regret at round $t$ as (a) the regret of the local last safe arm $a_t^\sharp$ and (b) the regret of arm $a$ to $a_t^\sharp$. In other words, it suffices to bound:

$$\mathbb{E}\left[\sum_{t=1}^T \sum_{a \in \mathcal{A}_t} \frac{\delta_t(a)}{|\mathcal{A}_t|} \mathbf{1}\{\mathcal{E}_1 \cap \mathcal{E}_2\} \,\Big|\, \mathbf{X}_T\right] = \sum_{t=1}^T \delta_t(a_t^\sharp) \mathbf{1}\{\mathcal{E}_1 \cap \mathcal{E}_2\} + \mathbb{E}\left[\sum_{t=1}^T \sum_{a \in \mathcal{A}_t} \frac{\delta_t(a_t^\sharp, a)}{|\mathcal{A}_t|} \mathbf{1}\{\mathcal{E}_1 \cap \mathcal{E}_2\} \,\Big|\, \mathbf{X}_T\right].$$

Note that the expectation on the first sum disappears since $a_t^\sharp$ is only a function of $\mathbf{X}_T$ and the mean reward functions $\{f_t^a(\cdot)\}_{t,a}$.

## D.2 Bounding the Regret of the Local Last Safe Arm

Bounding $\sum_{t=1}^T \delta_t(a_t^\sharp) \cdot \mathbf{1}\{\mathcal{E}_1 \cap \mathcal{E}_2\}$ will be similar to the proof of Proposition 2. We show that the oracle procedure could have essentially just played arm $a_t^\sharp$ every round.

Fix a phase $[\tau_i, \tau_{i+1})$ and let $r = r_{\tau_{i+1} - \tau_i}$. Fix a bin $B \in \mathcal{T}_r$ and let $a_i(B)$ be the local last safe arm $a_t^\sharp$ of the last round $t \in [\tau_i, \tau_{i+1})$ such that $X_t \in B$. Then, $a_t^\sharp = a_i(B)$ for every round $t \in [\tau_i, \tau_{i+1})$ such that $X_t \in B$. Then, we have by Definition 6 that for bin $B' \supseteq B$ at level $r_{t-\tau_i}$:

$$\sum_{s=\tau_i}^t \delta_s(a_i(B)) \cdot \mathbf{1}\{X_s \in B'\} \le \sqrt{K \cdot n_{B'}([\tau_i, t])} + r(B') \cdot n_{B'}([\tau_i, t]).$$

Then, by Lemma 9, we have:

$$\sum_{s=\tau_i}^t \delta_s(a_i(B)) \cdot \mathbf{1}\{X_s \in B\} \le c_4 \Big( \log^{1/2}(T) \cdot r^d \cdot (\tau_{i+1} - \tau_i)^{\frac{1+d}{2+d}} \cdot K^{\frac{1}{2+d}}$$
$$+ K \log(T) + \sqrt{\log(T) \mu(B) \cdot (t - \tau_i + 1)} \Big). \qquad (17)$$

Then, summing the above over bins in the same fashion as the proof of Proposition 2 gives:

$$\sum_{t=\tau_i}^{\tau_{i+1}-1} \delta_t(a_t^\sharp) = \sum_{B \in \mathcal{T}_r} \sum_{s=\tau_i}^{\tau_{i+1}-1} \delta_s(a_i(B)) \cdot \mathbf{1}\{X_s \in B\} \le c_{14} \log(T) \cdot (\tau_{i+1} - \tau_i)^{\frac{1+d}{2+d}} \cdot K^{\frac{1}{2+d}}.$$

Finally, summing over phases $[\tau_i, \tau_{i+1})$ we have $\sum_{t=1}^T \delta_t(a_t^\sharp)$ is of the right order.

## D.3 Relating Episodes to Significant Phases

We next show that w.h.p. a restart occurs (i.e., a new episode begins) only if a significant shift has occurred sometime within the episode. Recall from Definition 6 that $\tau_1, \tau_2, \ldots, \tau_{\tilde{L}}$ are the times of the significant shifts and that $t_1, \ldots, t_T$ are the episode start times.

**Lemma 11** (**Restart Implies Significant Shift**). *On event $\mathcal{E}_1$, for each episode $[t_\ell, t_{\ell+1})$ with $t_{\ell+1} \le T$ (i.e., an episode which concludes with a restart), there exists a significant shift $\tau_i \in [t_\ell, t_{\ell+1}]$.*

*Proof.* Fix an episode $[t_\ell, t_{\ell+1})$. Then, by Line 11 of Algorithm 1, there is a bin $B$ such that every arm $a \in [K]$ was evicted from $B$ at some round in the episode, i.e. (5) is true for each arm $a$ on some interval $[s_1, s_2] \subseteq [t_\ell, t_{\ell+1})$. It suffices to show that this implies a significnat shift has occurred between rounds $t_\ell$ and $t_{\ell+1}$.

Suppose (5) first triggers the eviction of arm $a$ at time $t$ in $B' \supseteq B$ over interval $[s_1, s_2]$ where $r(B') = r_{s_2-s_1}$. By concentration (10) and our eviction criteria (5), we have that there is an arm $a' \neq a$ such that (using the notation of Proposition 7) for large enough $C_0 > 0$ and some $c_{14} > 0$:

$$\sum_{s=s_1}^{s_2} \mathbb{E}\left[\hat{\delta}_s^B(a', a) \mid \mathcal{F}_{s-1}\right] \geq c_{14} \left(\sqrt{K \log(T) \cdot n_{B'}([s_1, s_2]) + (K \log(T))^2}\right.$$
$$\left. + r(B') \cdot n_{B'}([s_1, s_2])\right). \tag{18}$$

Next, if arm $a$ is evicted from $\mathcal{A}(B')$ at round $t$, then we have by the definition of $\hat{\delta}_s^{B'}(a', a)$ (4):

$$\mathbb{E}[\hat{\delta}_s^{B'}(a', a) \mid \mathcal{F}_{s-1}] = \begin{cases} \delta_s(a', a) \cdot \mathbf{1}\{X_s \in B'\} & a, a' \in \mathcal{A}_s \\ -f_s^a(X_s) \cdot \mathbf{1}\{X_s \in B\} & a \in \mathcal{A}_s, a' \notin \mathcal{A}_s \\ 0 & a \notin \mathcal{A}_s \end{cases}.$$

In any case, the above L.H.S. conditional expectation is bounded above by $\delta_s(a) \cdot \mathbf{1}\{X_s \in B'\}$. Thus, (18) implies arm $a$ incurs significant regret $(\star)$ in $B'$ on $[s_1, s_2]$:

$$\sum_{s=s_1}^{s_2} \delta_s(a) \cdot \mathbf{1}\{X_s \in B'\} \geq \sqrt{K \cdot n_{B'}([s_2, s_2])} + r(B') \cdot n_{B'}([s_1, s_2]).$$

Then, since every arm $a$ is evicted in bin $B$ by round $t$, a significant shift must have occurred in episode $[t_\ell, t_{\ell+1}]$. $\qquad\square$

### D.4 Regret of CMETA to the Last Safe Arm

It remains to bound $\mathbb{E}[\sum_{t=1}^T \sum_{a \in \mathcal{A}_t} \delta_t(a_t^\sharp, a)/|\mathcal{A}_t| \mid \mathbf{X}_t]$. We further decompose this sum over $t$ into episodes and then the blocks (see Definition 8) where a particular choice of level is used within the episode. The following notation will be useful.

**Definition 10.** *Let* $\text{PHASES}(\ell, r) \doteq \{i \in [\tilde{L}] : [\tau_i, \tau_{i+1}) \cap [s_\ell(r), e_\ell(r)] \neq \emptyset\}$ *be the phases which intersect block* $[s_\ell(r), e_\ell(r))$, *let* $T(i, r, \ell) \doteq |[\tau_i, \tau_{i+1}) \cap [s_\ell(r), e_\ell(r)]|$ *be the effective length of the phase as observed in block* $[s_\ell(r), e_\ell(r)]$.

*Similarly, define* $\text{PHASES}(\ell) \doteq \{i \in [\tilde{L}] : [\tau_i, \tau_{i+1}) \cap [t_\ell, t_{\ell+1}) \neq \emptyset\}$ *as the phases which intersect episode* $[t_\ell, t_{\ell+1})$.

It will in fact suffice to show w.h.p. w.r.t. the distribution of $\mathbf{X}_T$, for each episode $[t_\ell, t_{\ell+1})$, each block $[s_\ell(r), e_\ell(r)]$ in $[t_\ell, t_{\ell+1})$, and each bin $B \in \mathcal{T}_r$:

$$\mathbb{E}\left[\sum_{t=s_\ell(r)}^{e_\ell(r)} \sum_{a \in \mathcal{A}_t} \frac{\delta_t(a_t^\sharp, a)}{|\mathcal{A}_t|} \cdot \mathbf{1}\{X_t \in B\} \cdot \mathbf{1}\{\mathcal{E}_1 \cap \mathcal{E}_2\} \middle| \mathbf{X}_T\right]$$
$$\leq c_{15} \log(K) \mathbb{E}\left[\mathbf{1}\{\mathcal{E}_1 \cap \mathcal{E}_2\} \left(\log(T) + \log^2(T) \sum_{i \in \text{PHASES}(\ell,r)} r(B)^d \cdot T(i,r,\ell)^{\frac{1+d}{2+d}} \cdot K^{\frac{1}{2+d}}\right) \middle| \mathbf{X}_T\right]$$
$$\tag{19}$$

### D.5 Summing the Per-(Bin, Block, Episode) Regret over Bins, Blocks, and Episodes.

Admitting (19), we show that the total dynamic regret over $T$ rounds is of the desired order.

Recall from earlier that there are WLOG $T$ total episodes with the convention that $t_\ell \doteq T + 1$ if only $\ell$ episodes occur by round $T$. Then, summing our per-bin regret bound (19) over all the bins $B \in \mathcal{T}_r$

at level $r$ gives (using strong density to bound $\sum_{B \in r} r^d \le \frac{C_d}{c_d}$):

$$\mathbb{E}\left[\sum_{B \in \mathcal{T}_r} \sum_{t=s_\ell(r)}^{e_\ell(r)} \sum_{a \in \mathcal{A}_t} \frac{\delta_t(a_t^\sharp, a)}{|\mathcal{A}_t|} \cdot \mathbf{1}\{X_t \in B\} \cdot \mathbf{1}\{\mathcal{E}_1 \cap \mathcal{E}_2\} \middle| \mathbf{X}_T\right]$$

$$\le c_{16} \log(K) \mathbb{E}\left[\mathbf{1}\{\mathcal{E}_1 \cap \mathcal{E}_2\} \left(\sum_{B \in \mathcal{T}_r} \log(T) + \log^2(T) \sum_{i \in \text{PHASES}(\ell, r)} T(i, r, \ell)^{\frac{1+d}{2+d}} \cdot K^{\frac{1}{2+d}}\right) \middle| \mathbf{X}_T\right].$$

$$\tag{20}$$

Next, summing over the different levels $r$ (of which there are at most $\log(T)$ used in any episode), we obtain by Jensen's inequality on the concave function $z \mapsto z^{\frac{1+d}{2+d}}$:

$$\sum_{r \in \mathcal{R}} \sum_{i \in \text{PHASES}(\ell, r)} T(i, r, \ell)^{\frac{1+d}{2+d}} = \sum_{i \in \text{PHASES}(\ell)} \sum_{r \in \mathcal{R}: i \in \text{PHASES}(\ell, r)} T(i, r, \ell)^{\frac{1+d}{2+d}}$$

$$\le \sum_{i \in \text{PHASES}(\ell)} \left(\log(T) \sum_{r \in \mathcal{R}: i \in \text{PHASES}(\ell, r)} T(i, r, \ell)\right)^{\frac{1+d}{2+d}}.$$

Now, we have

$$\sum_{r \in \mathcal{R}: i \in \text{PHASES}(\ell, r)} T(i, r, \ell) = \sum_{r \in \mathcal{R}: i \in \text{PHASES}(\ell, r)} |[\tau_i, \tau_{i+1}) \cap [s_\ell(r), e_\ell(r)]| = \tau_{i+1} - \tau_i + 1.$$

We also have (via Fact 1 about level $r_{t_{\ell+1} - t_\ell}$ which is the smallest level used in episode $[t_\ell, t_{\ell+1})$).

$$\sum_{r \in \mathcal{R}} \sum_{B \in \mathcal{T}_r} \log(T) \le \sum_{r \in \mathcal{R}} r^{-d} \cdot \log(T)$$

$$\le c_{17} \log^2(T) \left(\frac{t_{\ell+1} - t_\ell}{K}\right)^{\frac{d}{2+d}}$$

$$\le c_{18} \log^2(T) \sum_{i \in \text{PHASES}(\ell)} (\tau_{i+1} - \tau_i)^{\frac{1+d}{2+d}} \cdot K^{\frac{1}{2+d}}.$$

Thus, combining the above inequalities with (20), we obtain overall bound:

$$c_{18} \log(K) \log^3(T) \mathbb{E}\left[\mathbf{1}\{\mathcal{E}_1 \cap \mathcal{E}_2\} \sum_{i \in \text{PHASES}(\ell)} (\tau_{i+1} - \tau_i)^{\frac{1+d}{2+d}} \cdot K^{\frac{1}{2+d}}\right].$$

Recall now that $\mathcal{E}_1$ is the good event over which the concentration bounds of Proposition 7 hold. Then, using the fact that, on event $\mathcal{E}_1$, each phase $[\tau_i, \tau_{i+1})$ intersects at most two episodes (Lemma 11), summing the above R.H.S over episodes $\ell \in [T]$ gives us (since at most $\log(T)$ blocks per episode) order

$$2 \log(K) \log^3(T) \sum_{i=1}^{\tilde{L}} (\tau_{i+1} - \tau_i)^{\frac{1+d}{2+d}} \cdot K^{\frac{1}{2+d}}.$$

It then remains to show the per-(bin, block, episode) regret bound (19).

## D.6 Bounding the Per-(Bin, Block, Episode) Regret to the Last Safe Arm

To show (19), we first fix a block $[s_\ell(r), e_\ell(r)]$ and a bin $B \in \mathcal{T}_r$. We then further decompose $\delta_t(a_t^\sharp, a)$ in two parts:

(a) The regret of $a$ to the *local last master arm*, denoted by $a_r(B)$, to be evicted from $\mathcal{A}_{\text{master}}(B)$ in block $[s_\ell(r), e_\ell(r)]$ (ties are broken arbitrarily).

(b) The regret of the local last master arm $a_r(B)$ to the last safe arm $a_t^\sharp$.

In other words, the L.H.S. of (19) is decomposed as:

$$\underbrace{\mathbb{E}\left[\sum_{t=s_\ell(r)}^{e_\ell(r)}\sum_{a\in\mathcal{A}_t}\frac{\delta_t(a_r(B),a)}{|\mathcal{A}_t|}\cdot\mathbf{1}\{X_t\in B\}\,\middle|\,\mathbf{X}_T\right]}_{(a)}+\underbrace{\mathbb{E}\left[\sum_{t=s_\ell(r)}^{e_\ell(r)}\delta_t(a_t^\sharp,a_r(B))\cdot\mathbf{1}\{X_t\in B\}\,\middle|\,\mathbf{X}_T\right]}_{(b)}.$$

We will show both (a) and (b) are of order (19).

• **Bounding the Regret of Other Arms to the Local Last Master Arm** $a_r(B)$. We start by partitioning the rounds $t$ such that $X_t\in B$ and $a\in\mathcal{A}_t$ in (a) according to before or after they are evicted from $\mathcal{A}_{\text{master}}(B)$. Suppose arm $a$ is evicted from $\mathcal{A}_{\text{master}}(B)$ at round $t_r^a\in[s_\ell(r),e_\ell(r)]$ (formally, we let $t_r^a\doteq e_\ell(r)$ if $a$ is not evicted in block $[s_\ell(r),e_\ell(r)]$). Then, it suffices to bound:

$$\mathbb{E}\left[\sum_{a=1}^{K}\sum_{t=s_\ell(r)}^{t_r^a-1}\frac{\delta_t(a_r(B),a)}{|\mathcal{A}_t|}\cdot\mathbf{1}\{X_t\in B\}+\sum_{a=1}^{K}\sum_{t=t_r^a}^{e_\ell(r)}\frac{\delta_t(a_r(B),a)}{|\mathcal{A}_t|}\cdot\mathbf{1}\{a\in\mathcal{A}_t\}\cdot\mathbf{1}\{X_t\in B\}\,\middle|\,\mathbf{X}_T\right].$$
(21)

Suppose WLOG that $t_r^1\le t_r^2\le\cdots\le t_r^K$. Then, for each round $t<t_r^a$ all arms $a'\ge a$ are retained in $\mathcal{A}_{\text{master}}(B)$ and thus retained in the candidate arm set $\mathcal{A}_t$ for all rounds $t$ where $X_t\in B$. Importantly, at each round $t$ a level of at least $r$ is used since a child Base-Alg can only use a higher level than the master Base-Alg. Thus, $|\mathcal{A}_t|\ge K+1-a$ for all $t\le t_r^a$.

Next, we bound the first double sum in (21), i.e. the regret of playing $a$ to $a_r(B)$ from $s_\ell(r)$ to $t_r^a-1$. Applying our concentration bounds (Proposition 7), since arm $a$ is not evicted from $\mathcal{A}(B)$ till round $t_r^a$, on event $\mathcal{E}_1$ we have for some $c_{19}>0$ and any other arm $a'\in\mathcal{A}(B)$ through round $t_r^a-1$ (i.e., $a'\in\mathcal{A}_t$ for all $t\in[t_\ell,t_r^a)$ such that $X_t\in B$ since we always use level at least $r$ at such a round $t$): for bin $B'\supseteq B$ at level $r_{t_r^a-1-s_\ell(r)}$: on event $\mathcal{E}_1$ (note that we necessarily always have $\mathcal{A}(B')\supseteq\mathcal{A}(B)$ for $B'\supseteq B$ by Line 14 of Algorithm 2):

$$\sum_{t=s_\ell(r)}^{t_r^a-1}\mathbb{E}[\hat\delta_s^{B'}(a',a)\mid\mathcal{F}_{t-1}]\le c_{19}\sqrt{K\log(T)\cdot(n_{B'}([s_\ell(r),t_r^a))\vee K\log(T))}+r(B')\cdot n_{B'}([s_\ell(r),t_r^a)).$$

Next, since $a,a'\in\mathcal{A}_t$ for each $t\in[s_\ell(r),t_r^a-1)$ such that $X_t\in B$, we have:

$$\forall t\in[s_\ell(r),t_r^a),X_t\in B:\mathbb{E}[\hat\delta_t^B(a',a)\mid\mathcal{F}_{t-1}]=\delta_t(a',a).$$

Thus, we conclude by (5):

$$\sum_{t=s_\ell(r)}^{t_r^a-1}\delta_t(a',a)\cdot\mathbf{1}\{X_t\in B\}\le c_{19}\sqrt{K\log(T)\cdot(n_{B'}([s_\ell(r),t_r^a))\vee K\log(T))}+r(B')\cdot n_{B'}([s_\ell(r),t_r^a)).$$

Thus, by Lemma 9, and since $B'\supseteq B$, we conclude for any such $a'$ on event $\mathcal{E}_1$: $\sum_{t=s_\ell(r)}^{t_r^a-1}\frac{\delta_t(a',a)}{|\mathcal{A}_t|}\cdot$ $\mathbf{1}\{X_t\in B\}$ is at most

$$\frac{c_4\left(\log^{1/2}(T)r^d\cdot K^{\frac{1}{2+d}}\cdot(t_r^a-s_\ell(r))^{\frac{1+d}{2+d}}+K\log(T)+\sqrt{\log(T)(t_r^a-s_\ell(r))\cdot\mu(B)}\right)}{K+1-a},\quad(22)$$

where we use the fact that $|\mathcal{A}_t|\ge K+1-a$ for all $t\in[s_\ell(r),t_r^a)$. Since this last bound holds uniformly for all $a'\in\mathcal{A}(B)$ through round $t_r^a-1$, it must hold for $a'=a_r(B)$, the local last master arm.

Then, summing over all arms $a$, we have on event $\mathcal{E}_1$:

$$\sum_{a=1}^{K}\sum_{t=s_\ell(r)}^{t_r^a-1}\frac{\delta_t(a_r(B),a)}{|\mathcal{A}_t|}\cdot\mathbf{1}\{X_t\in B\}\le c_4\log(K)\left(\log^{1/2}(T)\cdot r^d\cdot(e_\ell(r)-s_\ell(r))^{\frac{1+d}{2+d}}\cdot K^{\frac{1}{2+d}}+\right.$$

$$\left.K\log(T)+\sqrt{\log(T)(t_r^a-s_\ell(r))\cdot\mu(B)}\right).$$
(23)

Next note that by Lemma 10:

$$\sqrt{(t_r^a - s_\ell(r)) \cdot \mu(B)} \le \sqrt{(e_\ell(r) - s_\ell(r)) \cdot \mu(B)}$$
$$\le c_7 K^{\frac{d/2}{2+d}} (e_\ell(r) - s_\ell(r))^{\frac{1}{2+d}}$$
$$\le c_7 K^{\frac{1}{2+d}} \cdot r^d \cdot (e_\ell(r) - s_\ell(r))^{\frac{1}{2+d}}.$$

Additionally, since $K \le e_\ell(r) - s_\ell(r)$ (Fact 4), we have:

$$K \log(T) \le (e_\ell(r) - s_\ell(r))^{\frac{1}{2+d}} K^{\frac{1+d}{2+d}} \propto r^d \cdot (e_\ell(r) - s_\ell(r))^{\frac{1+d}{2+d}} \cdot K^{\frac{1}{2+d}}.$$

Thus, combining the above two displays with (23) gives us

$$\sum_{a=1}^{K} \sum_{t=s_\ell(r)}^{t_r^a-1} \frac{\delta_t(a_r(B), a)}{|\mathcal{A}_t|} \cdot \mathbf{1}\{X_t \in B\} \le c_{20} \log(K) \log(T) \cdot r^d \cdot (e_\ell(r) - s_\ell(r))^{\frac{1+d}{2+d}} \cdot K^{\frac{1}{2+d}}.$$

We next show the second double sum in (21) has an upper bound similar to the above. For this, we first observe that if arm $a$ is played in bin $B$ after round $t_r^a$, then it must be due to an active replay. The difficulty here is that replays may interrupt each other and so care must be taken in managing the contribution of $\sum_t \delta_t(a_r(B), a)$ (which may be negative) by different overlapping replays.

Our strategy, similar to that of Section B.1 in Suk and Kpotufe [2022], is to partition the rounds when $a$ is played by a replay after round $t_r^a$ according to which replay is active and not accounted for by another replay. This involves carefully identifying a subclass of replays whose durations while playing $a$ in $B$ span all the rounds where $a$ is played in $B$ after $t_r^a$. Then, we cover the times when $a$ is played by a collection of intervals corresponding to the schedules of this subclass of replays, on each of which we can employ the eviction criterion (5) and concentration bound as done earlier.

For this purpose, we first define the following terminology (which is all w.r.t. a fixed arm $a$, along with fixed block $[e_\ell(r), s_\ell(r)]$ and bin $B \in \mathcal{T}_r$):

**Definition 11.**

*(i) For each scheduled and activated* Base-Alg $(s, m)$, *let the round $M(s, m)$ be the minimum of two quantities: (a) the last round in $[s, s + m]$ when arm $a$ is retained in $\mathcal{A}(B)$ by* Base-Alg $(s, m)$ *and all of its children, and (b) the last round that* Base-Alg $(s, m)$ *is active and not permanently interrupted by another replay. Call the interval $[s, M(s, m)]$ the **active interval** of* Base-Alg $(s, m)$.*

*(ii) Call a replay* Base-Alg $(s, m)$ **proper** *if there is no other scheduled replay* Base-Alg $(s', m')$ *such that $[s, s + m] \subset (s', s' + m')$ where* Base-Alg $(s', m')$ *will become active again after round $s + m$. In other words, a proper replay is not scheduled inside the scheduled range of rounds of another replay. Let* PROPER$(s_\ell(r), e_\ell(r))$ *be the set of proper replays scheduled to start in the block $[s_\ell(r), e_\ell(r)]$.*

*(iii) Call a scheduled replay* Base-Alg $(s, m)$ *a **sub-replay** if it is non-proper and if each of its ancestor replays (i.e., previously scheduled replays whose durations have not concluded)* Base-Alg $(s', m')$ *satisfies $M(s', m') < s$. In other words, a sub-replay either permanently interrupts its parent or does not, but is scheduled after its parent (and all its ancestors) stops playing arm $a$ in $B$. Let* SUBPROPER$(s_\ell(r), s_\ell(r))$ *be the set of all sub-replays scheduled before round $t_{\ell+1}$.*

Equipped with this language, we now show some basic claims which essentially reduce analyzing the complicated hierarchy of replays to analyzing the active intervals of replays in PROPER$(s_\ell(r), e_\ell(r)) \cup$ SUBPROPER$(s_\ell(r), s_\ell(r))$.

**Proposition 12.** *The active intervals*

$$\{[s, M(s, m)] : \text{Base-Alg}\,(s, m) \in \text{PROPER}(s_\ell(r), e_\ell(r)) \cup \text{SUBPROPER}(s_\ell(r), s_\ell(r))\},$$

*are mutually disjoint.*

*Proof.* Clearly, the classes of replays PROPER$(t_\ell, t_{\ell+1})$ and SUBPROPER$(s_\ell(r), s_\ell(r))$ are disjoint. Next, we show the respective active intervals $[s, M(s, m)]$ and $[s', M(s', m')]$ of any two Base-Alg $(s, m)$ and Base-Alg $(s', m') \in$ PROPER$(s_\ell(r), e_\ell(r)) \cup$ SUBPROPER$(s_\ell(r), s_\ell(r))$ are disjoint.

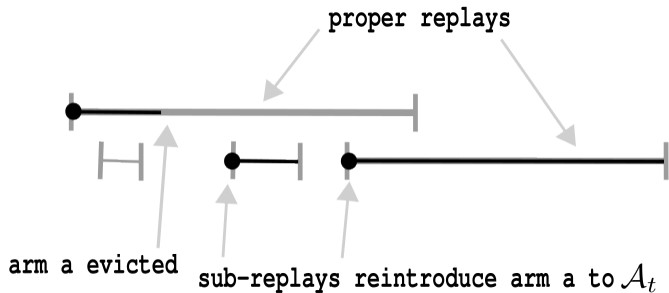

proper replays

arm a evicted    sub-replays reintroduce arm a to $\mathcal{A}_t$

Figure 2: Shown are replay scheduled durations (in gray) with dots marking when arm $a$ is reintroduced to $\mathcal{A}_t$. Black segments indicate the active intervals $[s, M(s,m)]$ for proper replays and sub-replays. Note that the rounds where $a \in \mathcal{A}_t$ in the left unlabeled replay's duration are accounted for by the larger proper replay.

1. Proper replay vs. sub-replay: a sub-replay can only be scheduled after the round $M(s,m)$ of the most recent proper replay Base-Alg $(s,m)$ (which is necessarily an ancestor). Thus, the active intervals of proper replays and sub-replays are disjoint.

2. Two distinct proper replays: two such replays can only intersect by one permanently interrupting the other, and since $M(s,m)$ always occurs before the permanent interruption of Base-Alg $(s,m)$, we have the active intervals of two such replays are disjoint.

3. Two distinct sub-replays: consider two non-proper replays Base-Alg $(s,m)$, Base-Alg $(s',m') \in \text{SUBPROPER}(s_\ell(r), s_\ell(r))$ with $s' > s$. Suppose their active intervals intersect and that Base-Alg $(s,m)$ is an ancestor of Base-Alg $(s',m')$. Then, if Base-Alg $(s',m')$ is a sub-replay, we must have $s' > M(s,m)$, which means that $[s', M(s',m')]$ and $[s, M(s,m)]$ are disjoint.

$\square$

Next, we claim that the active intervals $[s, M(s,m)]$ for Base-Alg $(s,m) \in \text{PROPER}(t_\ell, t_{\ell+1}) \cup \text{SUBPROPER}(s_\ell(r), s_\ell(r))$ contain all the rounds where $a$ is played in $B$ after being evicted from $\mathcal{A}_{\text{master}}(B)$. To show this, we first observe that for each round $t$ when a replay is active, there is a unique proper replay associated to $t$, namely the proper replay scheduled most recently. Next, note that any round $t > t_r^a$ where $X_t \in B$ and where arm $a \in \mathcal{A}_t$ must belong to the active interval $[s, M(s,m)]$ of this unique proper replay Base-Alg $(s,m)$ associated to round $t$, or else satisfies $t > M(s,m)$ in which case a unique sub-replay Base-Alg $(s',m') \in \text{SUBPROPER}(s_\ell(r), s_\ell(r))$ is active at round $t$ and not yet permanently interrupted by round $t$. Thus, it must be the case that $t \in [s', M(s',m')]$.

Overloading notation, we'll let $\mathcal{A}_t(B)$ be the value of $\mathcal{A}(B)$ for the Base-Alg active at round $t$. Next, note that every round $t \in [s, M(s,m)]$ for a proper or subproper Base-Alg $(s,m)$ is clearly a round where $a \in \mathcal{A}_t(B)$ and no such round is accounted for twice by Proposition 12. Thus,

$$\{t \in (t_r^a, e_\ell(r)] : a \in \mathcal{A}_t(B)\} = \bigsqcup_{\text{Base-Alg}\,(s,m)\in\text{PROPER}(s_\ell(r),e_\ell(r))\cup\text{SUBPROPER}(s_\ell(r),s_\ell(r))} [s, M(s,m)].$$

Then, we can rewrite the second double sum in (21) as:

$$\sum_{a=1}^{K} \sum_{\text{Base-Alg}\,(s,m)\in\text{PROPER}(s_\ell(r),e_\ell(r))\cup\text{SUBPROPER}(s_\ell(r),s_\ell(r))} Z_{m,s} \cdot \sum_{t=s\vee t_r^a}^{M(s,m)} \frac{\delta_t(a_r(B),a)}{|\mathcal{A}_t|} \cdot \mathbf{1}\{X_t \in B\}.$$

Recall in the above that the Bernoulli R.V. $Z_{m,s}$ (see Line 6 of Algorithm 1) decides whether Base-Alg $(s,m)$ is scheduled.

Further bounding the sum over $t$ above by its positive part, we can expand the middle sum above over Base-Alg $(s,m) \in \text{PROPER}(t_\ell, t_{\ell+1}) \cup \text{SUBPROPER}(s_\ell(r), s_\ell(r))$ to instead be over all Base-Alg $(s,m)$, or obtain:

$$\sum_{a=1}^{K} \sum_{\text{Base-Alg}\,(s,m)} Z_{m,s} \cdot \left( \sum_{t=s\vee t_r^a}^{M(s,m)} \frac{\delta_t(a_r(B),a)}{|\mathcal{A}_t|} \cdot \mathbf{1}\{X_t \in B\} \right)_+ ,$$

where the sum is over all replays $\mathsf{Base\text{-}Alg}\,(s, m)$, i.e. $s \in \{t_\ell + 1, \ldots, t_{\ell+1} - 1\}$ and $m \in \{2, 4, \ldots, 2^{\lceil \log(T) \rceil}\}$. It then remains to bound the contributed relative regret of each $\mathsf{Base\text{-}Alg}\,(s, m)$ in the interval $[s \vee t_r^a, M(s, m)]$, which will follow similarly to the previous steps in bounding the first double sum of (21).

We first have (now overloading the notation $M(s, m)$ as $M(s, m, a)$ for clarity), i.e. combining our concentration bound (10) with the eviction criterion (5) and applying Lemma 9:

$$\sum_{t=s \vee t_r^a}^{M(s,m,a)} \frac{\delta_t(a_r(B), a)}{|\mathcal{A}_t|} \cdot \mathbf{1}\{X_t \in B\}$$

$$\leq \frac{c_{21} \left( \log^{1/2}(T) r^d \cdot K^{\frac{1}{2+d}} \cdot M(s, m, a)^{\frac{1+d}{2+d}} + K \log(T) + \sqrt{\log(T) \cdot M(s, m, a) \cdot \mu(B)} \right)}{\min_{t \in [s, M(s,m,a)]} |\mathcal{A}_t|}$$

Thus, it remains to bound

$$\sum_{a,s,m} Z_{m,s} \left( \frac{\log^{1/2}(T) r^d K^{\frac{1}{2+d}} M(s, m, a)^{\frac{1+d}{2+d}} + K \log(T) + \sqrt{\log(T) M(s, m, a) \mu(B)}}{\min_{t \in [s, M(s,m,a)]} |\mathcal{A}_t|} \right).$$

Swapping the outer two sums and, similar to before, recognizing that $\sum_{a=1}^{K} \frac{1}{\min_{t \in [s, M(s,m,a)]} |\mathcal{A}_t|} \leq \log(K)$ by summing over arms in the order they are evicted by $\mathsf{Base\text{-}Alg}\,(s, m)$, we have that it remains to bound:

$$\log(K) \sum_{\mathsf{Base\text{-}Alg}\,(s,m)} Z_{m,s} \cdot c_{21} \left( \log^{1/2}(T) \cdot r^d \cdot K^{\frac{1}{2+d}} \cdot \tilde{m}^{\frac{1+d}{2+d}} + K \log(T) + \sqrt{\log(T) \cdot \tilde{m} \cdot \mu(B)} \right),$$

$$(24)$$

where $\tilde{m} \doteq m \wedge (e_\ell(r) - s_\ell(r))$ (note we may freely restrict all active intervals to the current block $[s_\ell(r), e_\ell(r)]$). Let

$$R(m, B) \doteq \left( c_{21} \left( \log^{1/2}(T) \cdot r^d \cdot K^{\frac{1}{2+d}} \cdot \tilde{m}^{\frac{1+d}{2+d}} + (K \wedge \tilde{m}) \log(T) \right. \right.$$

$$\left. \left. + \sqrt{\log(T) \cdot \tilde{m} \cdot \mu(B)} \right) \right) \wedge n_B([s, s+m]).$$

Then, in light of the previous calculations, $R(m, B)$ is an upper bound on the within-bin $B$ regret contributed by a replay of total duration $m$ (note we can always coarsely upper bound this regret by $n_B([s, s+m])$.

Then, plugging $R(m, B)$ into (24) gives via tower law:

$$\mathbb{E} \left[ \mathbb{E} \left[ \sum_{s,m} Z_{m,s} \cdot R(m, B) \middle| s_\ell(r) \right] \middle| \mathbf{X}_T \right] = \mathbb{E} \left[ \sum_{s=s_\ell(r)}^{T} \sum_{m} \mathbb{E}[Z_{m,s} \mathbf{1}\{s \leq e_\ell(r)\} \mid s_\ell(r)] R(m, B) \middle| \mathbf{X}_T \right]$$

$$(25)$$

Next, we observe that $Z_{m,s}$ and $\mathbf{1}\{s \leq e_\ell(r)\}$ are independent conditional on $s_\ell(r)$ since $\mathbf{1}\{s \leq e_\ell(r)\}$ only depends on the scheduling and observations of base algorithms scheduled before round $s$. Additionally, conditional on $s_\ell(r)$, the episode start time $t_\ell$ is also fixed since the two are deterministically related (see Fact 3). Then, we have that:

$$\mathbb{P}(Z_{m,s} = 1) = \mathbb{E}[Z_{m,s} \mid s_\ell(r)] = \mathbb{E}[Z_{m,s} \mid t_\ell, s_\ell(r)] = \left( \frac{1}{m} \right)^{\frac{1}{2+d}} \cdot \left( \frac{1}{s - t_\ell} \right)^{\frac{1+d}{2+d}}.$$

Thus,

$$\mathbb{E}[Z_{m,s} \cdot \mathbf{1}\{s \leq e_\ell(r)\} \mid s_\ell(r)] = \mathbb{E}[Z_{m,s} \mid s_\ell(r), t_\ell] \cdot \mathbb{E}[\mathbf{1}\{s \leq e_\ell(r)\} \mid s_\ell(r), t_\ell]$$

$$= \left( \frac{1}{m} \right)^{\frac{1}{2+d}} \cdot \left( \frac{1}{s - t_\ell} \right)^{\frac{1+d}{2+d}} \cdot \mathbb{E}[\mathbf{1}\{s \leq e_\ell(r)\} \mid s_\ell(r)].$$

Plugging this into (25) and unconditioning, we obtain:

$$\mathbb{E} \left[ \sum_{s=s_\ell(r)}^{e_\ell(r)} \sum_{n=1}^{\lceil \log(T) \rceil} \left( \frac{1}{2^n} \right)^{\frac{1}{2+d}} \left( \frac{1}{s - t_\ell} \right)^{\frac{1+d}{2+d}} \cdot R(2^n, B) \middle| \mathbf{X}_T \right]$$

$$(26)$$

We first evaluate the inner sum over $n$. Note that

$$\sum_{n=1}^{\lceil \log(T) \rceil} \left( \frac{1}{2^n} \right)^{\frac{1}{2+d}} \cdot (2^n \wedge (e_\ell(r) - s_\ell(r))^{\frac{1+d}{2+d}} \leq \log(T) \cdot (e_\ell(r) - s_\ell(r))^{\frac{d}{2+d}}$$

$$\sum_{n=1}^{\lceil \log(T) \rceil} \left( \frac{1}{2^n} \right)^{\frac{1}{2+d}} \sqrt{2^n \wedge (e_\ell(r) - s_\ell(r))} \leq (e_\ell(r) - s_\ell(r))^{\frac{d/2}{2+d}}$$

$$\sum_{n=1}^{\lceil \log(T) \rceil} \left( \frac{1}{2^n} \right)^{\frac{1}{2+d}} (K \wedge 2^n) \leq \log(T) \cdot K^{\frac{1+d}{2+d}}.$$

Next, we plug in the above displays into (26). In particular, multiplying the above displays by $(s - t_\ell)^{-\frac{1+d}{2+d}}$ and taking a further sum over $s \in [s_\ell(r), e_\ell(r)]$ gives an upper bound of:

$$(e_\ell(r) - t_\ell)^{\frac{1}{2+d}} \left( (e_\ell(r) - s_\ell(r))^{\frac{d}{2+d}} K^{\frac{1}{2+d}} \cdot r^d \cdot \log^{3/2}(T) \right.$$

$$\left. + (e_\ell(r) - s_\ell(r))^{\frac{d/2}{2+d}} \sqrt{\log(T) \cdot r^d} + K^{\frac{1+d}{2+d}} \log(T) \right).$$

First, we note the first term inside the parentheses above inside dominates the second term for all values of $K, e_\ell(r), s_\ell(r), T$.

Next, note from Fact 4 that $e_\ell(r) - t_\ell \leq c_{10}(e_\ell(r) - s_\ell(r))$ and so the above is at most:

$$r^d \cdot (e_\ell(r) - s_\ell(r))^{\frac{1+d}{2+d}} K^{\frac{1}{2+d}} \log^{3/2}(T) + \log(T) K^{\frac{1+d}{2+d}} \cdot (e_\ell(r) - s_\ell(r))^{\frac{1}{2+d}}. \tag{27}$$

We next recall from Fact 4 that each block $[s_\ell(r), e_\ell(r)]$ is at least $K$ rounds long. Thus,

$$r^d \cdot (e_\ell(r) - s_\ell(r))^{\frac{1+d}{2+d}} \cdot K^{\frac{1}{2+d}} \geq c_{22} \cdot (e_\ell(r) - s_\ell(r))^{\frac{1}{2+d}} \cdot K^{\frac{1+d}{2+d}}.$$

Thus, the second term of (27) is at most the order of the first term.

Showing (a) is order (19) then follows from writing $e_\ell(r) - s_\ell(r)$ as the sum of effective phase lengths $T(i, r, \ell)$ (see Definition 10) of the phases $[\tau_i, \tau_{i+1})$ intersecting block $[s_\ell(r), e_\ell(r)]$, and using the sub-additivity of $x \mapsto x^{\frac{1+d}{2+d}}$.

• **Bounding the Regret of the Last Master Arm** $a_r(B)$ **to the Last Safe Arm** $a_t^\sharp$. Before we proceed, we first convert $\sum_{t=s_\ell(r)}^{e_\ell(r)} \delta_t(a_t^\sharp, a_r(B)) \cdot \mathbf{1}\{X_t \in B\}$ into a more convenient form in terms of the masses $\mu(B)$. By concentration (12) of Proposition 7, we have

$$\sum_{t=s_\ell(r)}^{e_\ell(r)} \delta_t(a_t^\sharp, a_r(B)) \cdot \mathbf{1}\{X_t \in B\} \leq \sum_{t=s_\ell(r)}^{e_\ell(r)} \delta_t(a_t^\sharp, a_r(B)) \cdot \mu(B)$$

$$+ c_2 \left( \log(T) + \sqrt{\log(T)(e_\ell(r) - s_\ell(r)) \cdot \mu(B)} \right).$$

We first show the two concentration error terms on the R.H.S. above are negligible with respect to the desired bound (19). The $\log(T)$ term is clearly of the right order, whereas the other term is handled by Lemma 10, by which

$$\sqrt{(e_\ell(r) - s_\ell(r)) \cdot \mu(B)} \leq c_7(e_\ell(r) - s_\ell(r))^{\frac{1}{2+d}} \cdot K^{\frac{d/2}{2+d}} \leq c_{23} \cdot r^d \cdot (e_\ell(r) - s_\ell(r))^{\frac{1+d}{2+d}} \cdot K^{\frac{1}{2+d}}.$$

By similar arguments to before, where we write $e_\ell(r) - s_\ell(r) = \sum_{i \in \text{PHASES}(\ell, r)} T(i, r, \ell)$ and use the sub-additivity of the function $x \mapsto x^{\frac{1+d}{2+d}}$, the above is of the right order w.r.t. (19).

Thus, going forward, by the strong density assumption (Assumption 2) and in light of (19), it suffices to show for any fixed arm $a \in [K]$ (which we will take to be $a_r(B)$ in the end):

$$\sum_{t=s_\ell(r)}^{e_\ell(r) \wedge E(B,a)} \delta_t(a_t^\sharp, a) \lesssim \sum_{i \in \text{PHASES}(\ell, r)} (\tau_{i+1} - \tau_i)^{\frac{1+d}{2+d}} K^{\frac{1}{2+d}}, \tag{28}$$

where $E(B, a)$ is the last round in block $[s_\ell(r), e_\ell(r)]$ for which $a \in \mathcal{A}_{\text{master}}(B)$.

This aggregate gap is the most difficult quantity to bound since arm $a_t^\sharp$ may have been evicted from $\mathcal{A}_{\text{master}}(B)$ before round $t$ and, thus, we rely on our replay scheduling (Line 6 of Algorithm 2) to bound the regret incurred while waiting to detect a large aggregate value of $\delta_t(a_t^\sharp, a)$.

In an abuse of notation, we'll conflate $e_\ell(r)$ with the *anticipated block end time* based on $s_\ell(r)$; that is, the end block time if no episode restart occurs within the block. Now, for each phase $[\tau_i, \tau_{i+1})$ which intersects the block $[s_\ell(r), e_\ell(r)]$, our strategy will be to map out in time the *local bad segments* or subintervals of $[\tau_i, \tau_{i+1})$ where a fixed arm $a$ incurs significant regret to arm $a_t^\sharp$ in bin $B$, roughly in the sense of ($\star$). The argument will conclude by arguing that a well-timed replay is scheduled w.h.p. to detect some local bad segment in $B$, before too many elapse.

In particular, conditional on just the block start time $s_\ell(r)$, we define the bad segments for a fixed arm $a$ and then argue that if too many bad segments w.r.t. $a$ elapse in the block's anticipated set of rounds $[s_\ell(r), e_\ell(r)]$, then arm $a$ will be evicted in bin $B$. Crucially, this will hold uniformly over all arms $a$ and, in particular, for arm $a \doteq a_r(B)$. This will then bound the regret of $a_r(B)$ in block $[s_\ell(r), e_\ell(r)]$ in the sense of (28).

**Notation.** *Going forward, we will drop the dependence on the level $r$, block $[s_\ell(r), e_\ell(r)]$, and episode $[t_\ell, t_{\ell+1})$ in certain definitions as they are fixed momentarily. Recall from Appendix D.2 that $a_t^\sharp$ is the local last safe arm of the last round $t_i(B) \in [\tau_i, \tau_{i+1})$ such that $X_{t_i(B)} \in B$ where $B$ is the bin at level $r_{\tau_{i+1} - \tau_i}$ containing $X_t$ (see Definition 9).*

We first introduce the notion of a *bad segment* of rounds which is a minimal period where large regret in the sense of ($\star$) within bin $B$ is detectable by a well-timed replay.

**Definition 12.** *Fix an arm $a$ and $s_\ell(r)$, and let $[\tau_i, \tau_{i+1})$ be any phase intersecting $[s_\ell(r), e_\ell(r)]$. Define rounds $s_{i,0}(a), s_{i,1}(a), s_{i,2}(a) \ldots \in [t_\ell \vee \tau_i, \tau_{i+1})$ recursively as follows: let $s_{i,0}(a) \doteq t_\ell \vee \tau_i$ and define $s_{i,j}(a)$ as the smallest round in $(s_{i,j-1}(a), \tau_{i+1} \wedge e_\ell(r))$ such that arm $a$ satisfies for some fixed $c_{21} > 0$:*

$$\sum_{t=s_{i,j-1}(a)}^{s_{i,j}(a)} \delta_t(a_t^\sharp, a) \geq c_{24} \log(T) \cdot (s_{i,j}(a) - s_{i,j-1}(a))^{\frac{1+d}{2+d}} \cdot K^{\frac{1}{2+d}}. \tag{29}$$

*Otherwise, we let the $s_{i,j}(a) \doteq \tau_{i+1} - 1$. We refer to the interval $[s_{i,j-1}(a), s_{i,j}(a))$ as a **bad segment**. We call $[s_{i,j-1}(a), s_{i,j}(a))$ a **proper bad segment** if (29) above holds.*

It will in fact suffice to constrain our attention to proper bad segments, since non-proper bad segments $[s_{i,j-1}(a), s_{i,j}(a))$ (where $s_{i,j}(a) = \tau_{i+1} - 1$ and (29) is reversed) will be negligible in the regret analysis since there is at most one non-proper bad segment per phase $[\tau_i, \tau_{i+1})$ (i.e., the regret of each non-proper bad segment is at most the R.H.S. of (28)).

We first establish some elementary facts about proper bad segments which will later serve useful in analyzing the detectability of ($\star$) along such segments of time.

**Lemma 13.** *Let $[s_{i,j}(a), s_{i,j+1}(a))$ be a proper bad segment defined w.r.t. arm $a$. Let $m \in \mathbb{N} \cup \{0\}$ be such that $r_{s_{i,j+1}(a) - s_{i,j}(a)} = 2^{-m}$. Then, for some $c_{25} = c_{25}(d) > 0$ depending on the dimension $d$:*

$$\sum_{t=s_{i,j+1}(a) - K2^{(m-2)(2+d)-1}}^{s_{i,j+1}(a)} \delta_t(a_t^\sharp, a) \geq c_{25} \log(T) \cdot K^{\frac{1}{2+d}} (s_{i,j+1}(a) - s_{i,j}(a))^{\frac{1+d}{2+d}}. \tag{30}$$

*Proof.* First, we may assume $s_{i,j+1}(a) - s_{i,j}(a) \geq 4 \cdot K$ by choosing $c_{24}$ in (29) large enough (this will make $m - 1 \geq 0$).

First, observe by the definition of $r_{s_{i,j+1}(a) - s_{i,j}(a)}$ (Notation 1) that

$$K2^{(m-1)(2+d)} \leq s_{i,j+1}(a) - s_{i,j}(a) < K2^{m(2+d)}. \tag{31}$$

Now, let $\tilde{s} \doteq s_{i,j+1}(a) - K2^{(m-2)(2+d)-1}$. Then, we have by (29) in the construction of the $s_{i,j}(a)$'s (Definition 12) that:

$$\sum_{t=\tilde{s}}^{s_{i,j+1}(a)} \delta_t(a_t^\sharp, a) = \sum_{t=s_{i,j}(a)}^{s_{i,j+1}(a)} \delta_t(a_t^\sharp, a) - \sum_{t=s_{i,j}(a)}^{\tilde{s}} \delta_t(a_t^\sharp, a)$$

$$\geq c_{24} \log(T) K^{\frac{1}{2+d}} \left( (s_{i,j+1}(a) - s_{i,j}(a))^{\frac{1+d}{2+d}} - (\tilde{s} - s_{i,j}(a))^{\frac{1+d}{2+d}} \right)$$

Let $m_{i,j}(a) \doteq s_{i,j+1}(a) - s_{i,j}(a)$. Then, we have by (31) that:

$$m_{i,j}(a) \leq K2^{m(2+d)} \implies \tilde{s} - s_{i,j}(a) = m_{i,j}(a) - K2^{(m-2)(2+d)-1} \leq m_{i,j}(a) \cdot (1 - 2^{-2(2+d)-1}).$$

Plugging this into our earlier bound the constants in our updated lower bound scale like:

$$1 - \left( 1 - \frac{1}{2^{2(2+d)+1}} \right)^{\frac{1+d}{2+d}} > 0.$$

But, this last term is positive for all $d \in \mathbb{N} \cup \{0\}$ and only depends on $d$. $\qquad\square$

**Lemma 14** (Aggregate Gap Dominates Concentration Error). *Fix a bin $B$ at level $r$. Let $[s_{i,j}(a), s_{i,j+1}(a))$ be a proper bad segment and let $B' \supseteq B$ be the bin at level $r_{s_{i,j+1}(a)-\tilde{s}}$ where $\tilde{s} \doteq s_{i,j+1}(a) - K2^{(m-2)(2+1)-1}$ is as in Lemma 13. Then, for some $c_{26} > 0$:*

$$\sum_{t=\tilde{s}}^{s_{i,j+1}(a)} \delta_t(a_t^\sharp, a) \cdot \mathbf{1}\{X_t \in B'\} \geq c_{26} \left( \log(T)\sqrt{K \cdot (n_{B'}([\tilde{s}, s_{i,j+1}(a)]) \vee K)} \right.$$

$$\left. + r(B') \cdot n_{B'}([\tilde{s}, s_{i,j+1}(a)]) \right).$$

*Proof.* We have via concentration ((12) of Lemma 8), Lemma 13, and the strong density assumption (Assumption 2):

$$\sum_{t=\tilde{s}}^{s_{i,j+1}(a)} \delta_t(a_t^\sharp, a) \cdot \mathbf{1}\{X_t \in B'\} \geq \sum_{t=\tilde{s}}^{s_{i,j+1}(a)} \delta_t(a_t^\sharp, a) \cdot \mu(B')$$

$$- c_2 \left( \log(T) + \sqrt{\log(T)(s_{i,j+1}(a) - \tilde{s}) \cdot \mu(B')} \right)$$

$$\geq c_{25} \log(T)(s_{i,j+1}(a) - s_{i,j}(a))^{\frac{1}{2+d}} \cdot K^{\frac{1+d}{2+d}}$$

$$- c_2 \left( \log(T) + \sqrt{\log(T)(s_{i,j+1}(a) - \tilde{s}) \cdot \mu(B')} \right).$$

Now, the first term on the final R.H.S. above dominates the other two terms for large enough $c_{25}$ and via strong density assumption (Assumption 2).

Thus, it suffices to show

$$c_{25} \log(T)(s_{i,j+1}(a) - s_{i,j}(a))^{\frac{1}{2+d}} \cdot K^{\frac{1+d}{2+d}} \geq$$

$$c_{27} \left( \log(T)\sqrt{K \cdot (n_{B'}([\tilde{s}, s_{i,j+1}(a)]) \vee K)} + r(B') \cdot n_{B'}([\tilde{s}, s_{i,j+1}(a)]) \right). \qquad (32)$$

We first upper bound the "variance" term, or the first term on the R.H.S. above. Let $W \doteq s_{i,j+1}(a) - \tilde{s}$. By Lemma 10, we have

$$\log(T)\sqrt{K \cdot (n_{B'}([\tilde{s}, s_{i,j+1}(a)]) \vee K)} \leq c_8 \left( \log(T) \cdot W^{\frac{1}{2+d}} \cdot K^{\frac{1+d}{2+d}} + \log^{3/2}(T) + K\log(T) \right.$$

$$\left. + \log^{5/4}(T) \cdot K^{\frac{1+3d/4}{2+d}} \cdot W^{\frac{1/2}{2+d}} \right).$$

Now, we also have

$$s_{i,j+1}(a) - s_{i,j}(a) \geq W \implies \log(T) \cdot (s_{i,j+1}(a) - s_{i,j}(a))^{\frac{1}{2+d}} \cdot K^{\frac{1+d}{2+d}} \geq \log(T) \cdot W^{\frac{1}{2+d}} \cdot K^{\frac{1+d}{2+d}}$$

Next, we note that by the definition of a proper bad segment ((29) in Definition 12) that

$$2 \cdot (s_{i,j+1}(a) - s_{i,j}(a)) \geq \sum_{t=s_{i,j}(a)}^{s_{i,j}(a)} \delta_t(a_t^\sharp, a) \geq c_{24} \log(T) \cdot (s_{i,j+1}(a) - s_{i,j}(a))^{\frac{1+d}{2+d}} \cdot K^{\frac{1}{2+d}}.$$

This implies $(s_{i,j+1}(a) - s_{i,j}(a))^{\frac{1}{2+d}} \geq \frac{c_{24}}{2} \log(T)$. By similar reasoning, we have $s_{i,j+1}(a) - s_{i,j}(a) \geq c_{28}K$. From this, we conclude for $c_{24} > 0$ large enough:

$$\log(T) \cdot (s_{i,j+1}(a) - s_{i,j}(a))^{\frac{1}{2+d}} \cdot K^{\frac{1+d}{2+d}} \geq \log^{3/2}(T) + K \log(T) + \log^{5/4}(T) \cdot K^{\frac{1+3d/4}{2+d}} \cdot W^{\frac{1/2}{2+d}}.$$

Thus, (32) is shown.

$\square$

Now, we define a well-timed or *perfect replay* which, if scheduled, will detect the badness of arm $a$ (in the sense of (5)) in bin $B$ over a proper bad segment $[s_{i,j}(a), s_{i,j+1}(a))$. The simplest such perfect replay is one which is scheduled directly from rounds $s_{i,j}(a)$ to $s_{i,j+1}(a)$. We in fact show there is a spectrum of replays (of size the length of the segment $\Omega(s_{i,j+1}(a) - s_{i,j}(a))$) each of which can detect arm $a$ is bad in bin $B$, possibly by using a larger ancestor bin $B' \supseteq B$.

**Definition 13** (Perfect Replay). *For a fixed proper bad segment $[s_{i,j}(a), s_{i,j+1}(a))$, define a perfect replay as a* Base-Alg $(t_{\text{start}}, M)$ *with* $t_{\text{start}} \in [s_{i,j+1}(a) - K2^{(m-2)(2+d)} + 1, s_{i,j+1}(a) - K2^{(m-2)(2+d)-1}]$ *(where* $m \in \mathbb{N} \cup \{0\}$ *is as in Lemma 13) and* $t_{\text{start}} + M \geq s_{i,j+1}(a)$.

The following proposition analyzes the behavior of a perfect replay and shows, if scheduled, it will in fact evict arm $a$ from $\mathcal{A}(B)$ within a proper bad segment $[s_{i,j}(a), s_{i,j+1}(a))$.

**Proposition 15** (Perfect Replay Evicts Bad Arm in Proper Bad Segment). *Suppose event $\mathcal{E}_1 \cap \mathcal{E}_2$ holds (see Notation 2). Fix a bin $B$ at level $r$. Let $[s_{i,j}(a), s_{i,j+1}(a))$ be a proper bad segment defined with respect to arm $a$. Let* Base-Alg $(t_{\text{start}}, M)$ *be a perfect replay as defined above which becomes active at $t_{\text{start}}$ (i.e., $Z_{t_{\text{start}}, M} = 1$) for a fixed integer $M \geq s_{i,j+1}(a) - s_{i,j}(a)$. Then:*

(i) *Let $B'$ be the bin at level $r_{\tilde{s}-s_{i,j}(a)}$ where $\tilde{s} \doteq s_{i,j+1}(a) - K2^{(m-2)(2+d)-1}$ ($m \in \mathbb{N} \cup \{0\}$ is as in Lemma 13), as in Lemma 14. Then, there is a "safe arm" $a^\sharp(B')$ which will not be evicted from $\mathcal{A}(B')$ by* Base-Alg $(t_{\text{start}}, M)$ *(or any of its children) before round $s_{i,j+1}(a)+1$.*

(ii) *If $a \in \mathcal{A}_t$ for all rounds $t \in [\tilde{s}, s_{i,j+1}(a))$ where $X_t \in B$, w then arm $a$ will be excluded from $\mathcal{A}(B)$ by round $s_{i,j+1}(a)$.*

*Proof.* For (i), we can define the "safe arm" $a^\sharp(B')$ in a similar fashion to how $a_t^\sharp$ was defined. Let $t_i(B')$ be the last round in $[\tilde{s}, s_{i,j+1}(a)]$ such that $X_{t_i(B')} \in B'$. Then, since $s_{i,j+1}(a) < \tau_{i+1}$, we have that at round $t_i(B')$, there is a safe arm $a^\sharp(B')$ which does not satisfy $(\star)$ for any bin $B''$ intersecting $B'$ and interval of rounds $I \subseteq [\tilde{s}, s_{i,j+1}(a)]$. Once Base-Alg $(t_{\text{start}}, M)$ is scheduled, it (or any of its children) cannot evict arm $a^\sharp(B')$ from $B'$ as doing so would imply it has significant regret in some bin intersecting $B'$ (following the same calculations as in Lemma 11).

We next turn to (ii). We first suppose that arms $a$ is active in bin $B'$ from rounds $\tilde{s}$ to $s_{i,j+1}(a)$ (we'll carefully argue later this is indeed the case). We first observe $\mathbb{E}[\hat{\delta}_t^B(a^\sharp(B), a) \mid \mathcal{F}_{t-1}] = \delta_t(a_i^\sharp(B), a)$ for any round $t \in [\tilde{s}, s_{i,j+1}(a)]$ such that $X_t \in B'$. We next observe that:

$$\sum_{t=\tilde{s}}^{s_{i,j+1}(a)} \delta_t(a^\sharp(B'), a) \cdot \mathbf{1}\{X_t \in B'\} \geq \sum_{t=\tilde{s}}^{s_{i,j+1}(a)} \delta_t(a_t^\sharp, a) \cdot \mathbf{1}\{X_t \in B'\} - \sum_{t=\tilde{s}}^{s_{i,j+1}(a)} \delta_t(a^\sharp(B')) \cdot \mathbf{1}\{X_t \in B'\}.$$

$$(33)$$

By Lemma 14, the first term on the R.H.S. is at least

$$c_{26}\left(\log(T)\sqrt{K \cdot (n_{B'}([\tilde{s}, s_{i,j+1}(a)]) \vee K)} + r(B') \cdot n_{B'}([\tilde{s}, s_{i,j+1}(a)])\right).$$

Meanwhile, the second term on the R.H.S. of (33) is at most the same order by the definition of $a^\sharp(B)$. Thus, choosing $c_{26}$ large enough gives us that

$$\sum_{t=\tilde{s}}^{s_{i,j+1}(a)} \delta_t(a^\sharp(B'), a) \cdot \mathbf{1}\{X_t \in B'\} \geq c_{29}\left(\log(T)\sqrt{K \cdot (n_{B'}([\tilde{s}, s_{i,j+1}(a)]) \vee K)}\right.$$
$$\left. + r(B') \cdot n_{B'}([\tilde{s}, s_{i,j+1}(a)])\right).$$

Then, combining the above with our eviction criterion (5) and concentration (10), we have that arm $a$ will be evicted in the bin $B' \supseteq B$ at level $r_{s_{i,j+1}(a)-\tilde{s}}$ by round $s_{i,j+1}(a)$.

Finally, it remains to show that, within Base-Alg $(t_{\text{start}}, M)$'s play, arm $a$ will **not** be evicted in any child of $B'$ before round $s_{i,j+1}(a)$. This will follow from the fact that any perfect replay must use a level in $\mathcal{R}$ of size at least $r_W$. In particular, by Definition 13, the starting round $t_{\text{start}}$ is "close enough" to the critical round $s_{i,j+1}(a) - K2^{(m-2)(2+d)-1}$ so that it will not use a different level than the perfect replay which starts exactly at this critical round.

Formally, we have that the smallest level a perfect replay can use is $r_{\tilde{W}}$ where $\tilde{W} \doteq K \cdot 2^{(m-2)(2+d)} - 1$.

Next, note that $s_{i,j+1}(a) - t_{\text{start}} \leq K \cdot 2^{(m-2)(2+d)} - 1$ and so

$$\left(\frac{K}{s_{i,j+1}(a) - t_{\text{start}}}\right)^{\frac{1}{2+d}} \geq \left(\frac{K}{K \cdot 2^{(m-2)(2+d)} - 1}\right)^{\frac{1}{2+d}} \geq 2^{-(m-2)}.$$

Thus, $r_{\tilde{W}} \geq 2^{-(m-2)}$. On the other hand,

$$\left(\frac{K}{s_{i,j+1}(a) - \tilde{s}}\right)^{\frac{1}{2+d}} = \frac{1}{2^{m-2-\frac{1}{2+d}}} \in [2^{-(m-2)}, 2^{-(m-3)}).$$

Thus, $2^{-(m-2)} = r_{s_{i,j+1}(a)-\tilde{s}}$ is also the level used to detect that arm $a$ is bad in bin $B'$. Thus, we conclude that $r_{\tilde{W}}$ is no smaller than the level $r_{s_{i,j+1}(a)-\tilde{s}}$ used to evict arm $a$ in bin $B'$. This means arm $a$ cannot be evicted in a child of $B'$ before round $s_{i,j+1}(a)$, if $a$ is not already evicted in $B$. $\square$

Next, we show for any arm $a$ (in particular, $a = a_r(B)$), a perfect replay characterized by Definition 13 is scheduled with high probability if too many bad segments w.r.t. $a$ elapse, thus bounding the regret of $a$ to $a_i^\sharp(B)$ over the phases $[\tau_i, \tau_{i+1})$ intersecting block $[s_\ell(r), e_\ell(r)]$.

### D.7 Bounding the Regret of the Last Master Arm $a_r(B)$ to the Last Safe Arm $a_t^\sharp$

Next, we bound the the regret of a fixed arm $a$ to $a_t^\sharp$ over the bad segments w.r.t. $a$ in $B$. Recall from earlier (28) that our remaining goal is to establish the following bound for every bin $B \in \mathcal{T}_r$ at level $r$:

$$\mathbb{E}\left[\max_{a \in [K]} \sum_{t=s_\ell(r)}^{e_\ell(r) \wedge E(B,a)} \delta_t(a_t^\sharp, a) \cdot \mathbf{1}\{\mathcal{E}_1 \cap \mathcal{E}_2\}\right] \lesssim \sum_{i \in \text{PHASES}(\ell,r)} (\tau_{i+1} - \tau_i)^{\frac{1+d}{2+d}} \cdot K^{\frac{1}{2+d}},$$

where $E(B, a)$ is the last round in block $[s_\ell(r), e_\ell(r)]$ for which $a \in \mathcal{A}_{\text{master}}(B)$.

Note that the bad segments (Definition 12) are defined for a level $r$, block start time $s_\ell(r)$, and phase $[\tau_i, \tau_{i+1})$. In particular, they are defined independent of any choice of bin $B$ at level $r$. We'll similarly define a *bad round* $s(a)$ which will indicate when "too many" bad segments w.r.t. $a$ have elapsed, irrespective of a choice of bin $B$. Then, we'll argue that for any bin $B$ at level $r$, $E(B, a) \leq s(a)$.

It should be understood that in what follows, we condition on the block start time $s_\ell(r)$. First, fix an arm $a$ and define the *bad round* $s(a) > s_\ell(r)$ as the smallest round which satisfies, for some fixed $c_{30} > 0$:

$$\sum_{(i,j)} (s_{i,j+1}(a) - s_{i,j}(a))^{\frac{1+d}{2+d}} > c_{30} \log(T)(s(a) - t_\ell)^{\frac{1+d}{2+d}} \tag{34}$$

where the above sum is over all pairs of indices $(i, j) \in \mathbb{N} \times \mathbb{N}$ such that $[s_{i,j}(a), s_{i,j+1}(a))$ is a proper bad segment with $s_{i,j+1}(a) < s(a)$. We will show that, for any bin $B$ at level $r$, arm $a$ is evicted from $\mathcal{A}(B)$ episode $\ell$ with high probability by the time the bad round $s(a)$ occurs.

For each proper bad segment $[s_{i,j}(a), s_{i,j+1}(a))$, let $\tilde{s}_{i,j}(a) \doteq s_{i,j+1}(a) - K2^{(m-2)(2+d)-1}$ denote the "critical point" of the bad segment as in Lemma 13 and also let $m_{i,j} \doteq 2^n$ where $n \in \mathbb{N}$ satisfies:

$$2^n \geq s_{i,j+1}(a) - s_{i,j}(a) > 2^{n-1}.$$

Next, recall that the Bernoulli $Z_{M,t}$ decides whether Base-Alg $(t, M)$ activates at round $t$ (see Line 6 of Algorithm 1). If for some $t \in [\hat{s}_{i,j}(a), \tilde{s}_{i,j}(a)]$ where $\hat{s}_{i,j}(a) \doteq s_{i,j+1}(a) - K2^{(m-2)(2+d)} + 1$, $Z_{m_{i,j},t} = 1$, i.e. a perfect replay is scheduled, then $a$ will be evicted from $\mathcal{A}(B)$ by round $s_{i,j+1}(a)$ (Proposition 15).

We will show this happens with high probability via concentration on the sum $\sum_{(i,j)} \sum_t Z_{m_{i,j},t}$ where $j, i, t$ run through all $t \in [\hat{s}_{i,j}(a), \tilde{s}_{i,j}(a))$ and all proper bad segments $[s_{i,j}(a), s_{i,j+1}(a))$ with $s_{i,j+1}(a) < s(a)$. Note that these random variables, conditional on $\mathbf{X}_T$, depend only on the fixed arm $a$, the block start time $s_\ell(r)$, and the randomness of scheduling replays on Line 6. In particular, the $Z_{m_{i,j},t}$ are independent conditional on $t_\ell$.

Then, a Chernoff bound over the randomization of CMETA on Line 6 of Algorithm 1 conditional on $t_\ell$ yields

$$\mathbb{P}\left(\sum_{(i,j)} \sum_t Z_{m_{i,j},t} \leq \frac{\mathbb{E}[\sum_{(i,j)} \sum_t Z_{m_{i,j},t} \mid s_\ell(r), \mathbf{X}_T]}{2} \middle| s_\ell(r), \mathbf{X}_T\right)$$

$$\leq \exp\left(-\frac{\mathbb{E}[\sum_{(i,j)} \sum_t Z_{m_{i,j},t} \mid s_\ell(r), \mathbf{X}_T]}{8}\right).$$

We claim the error probability on the R.H.S. above is at most $1/T^3$. To this end, we compute:

$$\mathbb{E}\left[\sum_{(i,j)} \sum_t Z_{m_{i,j},t} \middle| s_\ell(r), \mathbf{X}_T\right] \geq \sum_{(i,j)} \sum_{t=\hat{s}_{i,j}(a)}^{\tilde{s}_{i,j}(a)} \left(\frac{1}{m_{i,j}}\right)^{\frac{1}{2+d}} \left(\frac{1}{t - t_\ell}\right)^{\frac{1+d}{2+d}}$$

$$\geq \frac{1}{4} \sum_{(i,j)} m_{i,j}^{\frac{1+d}{2+d}} \left(\frac{1}{s(a) - t_\ell}\right)^{\frac{1+d}{2+d}}$$

$$\geq \frac{c_{30}}{4} \log(T),$$

where the last inequality follows from (34). The R.H.S. above is larger than $24 \log(T)$ for $c_{30}$ large enough, showing that the error probability is small. Taking a further union bound over the choice of arm $a \in [K]$ gives us that $\sum_{(i,j)} \sum_t Z_{m_{i,j},t} > 1$ for all choices of arm $a$ (define this as the good event $\mathcal{E}_3(s_\ell(r))$) with probability at least $1 - K/T^3$.

Recall on the event $\mathcal{E}_1 \cap \mathcal{E}_2$ the concentration bounds of Proposition 7 and Lemma 8 hold. Then, on $\mathcal{E}_1 \cap \mathcal{E}_2 \cap \mathcal{E}_3(s_\ell(r))$, we must have for each bin $B \in \mathcal{T}_r$ at level $r$, $E(B, a) \leq s(a)$ since otherwise $a$ would have been evicted in $\mathcal{A}(B)$ by some perfect replay before the end of the block $e_\ell(r)$ by virtue of $\sum_{(i,j)} \sum_t Z_{m_{i,j},t} > 1$ for arm $a$. Thus, by the definition of the bad round $s(a)$ (34), we must have:

$$\sum_{[s_{i,j}(a), s_{i,j+1}(a)) : s_{i,j+1}(a) < e_\ell(r)} (s_{i,j+1}(a) - s_{i,j}(a))^{\frac{1+d}{2+d}} \leq c_{30} \log(T)(e_\ell(r) - t_\ell)^{\frac{1+d}{2+d}} \qquad (35)$$

Thus, by (29) in Definition 12, over the proper bad segments $[s_{i,j}(a), s_{i,j+1}(a))$ which elapse before round $e_\ell(r) \wedge E(B, a)$ in phase $[\tau_i, \tau_{i+1})$: the regret is at most

$$\sum_{(i,j)} \log(T) \cdot K^{\frac{1}{2+d}} m_{i,j}^{\frac{1+d}{2+d}} \leq \log^2(T) \cdot K^{\frac{1}{2+d}} \cdot (e_\ell(r) - t_\ell)^{\frac{1+d}{2+d}}$$

Over each non-proper bad segment $[s_{i,j}(a), s_{i,j-1}(a))$ and the last segment $[s_{i,j}(a), e_\ell(r) \wedge E(B, a)]$, the regret of playing arm $a$ to $a_t^\sharp$ is at most $\log(T) \cdot K^{\frac{1}{2+d}} m_{i,j}^{\frac{1+d}{2+d}}$ since there is at most one non-proper bad segment per phase $[\tau_i, \tau_{i+1})$ (see (29) in Definition 12).

So, we conclude that on event $\mathcal{E}_1 \cap \mathcal{E}_2 \cap \mathcal{E}_3(s_\ell(r))$: for any bin $B \in \mathcal{T}_r$

$$\max_{a \in [K]} \sum_{t=s_\ell(r)}^{e_\ell(r) \wedge E(B,a)} \delta_t(a_t^\sharp, a) \leq 2c_{30} \log^2(T) \sum_{i \in \text{PHASES}(\ell,r)} (\tau_{i+1} - \tau_i)^{\frac{1+d}{2+d}} \cdot K^{\frac{1}{2+d}}.$$

Let event $\mathcal{G} \doteq \mathcal{E}_1 \cap \mathcal{E}_2$. Then, taking expectation, we have by conditioning first on $s_\ell(r)$ and then on event $\mathcal{G} \cap \mathcal{E}_3(s_\ell(r))$:

$$\mathbb{E}\left[ \max_{a \in [K]} \sum_{t=s_\ell(r)}^{e_\ell(r) \wedge E(B,a)} \delta_t(a_t^\sharp, a) \cdot \mathbf{1}\{\mathcal{G}\} \middle| \mathbf{X}_T \right]$$

$$\leq \mathbb{E}_{s_\ell(r)} \left[ \mathbb{E}\left[ \mathbf{1}\{\mathcal{G} \cap \mathcal{E}_3(s_\ell(r))\} \max_{a \in [K]} \sum_{t=s_\ell(r)}^{e_\ell(r) \wedge E(B,a)} \delta_t(a_t^\sharp, a_r(B)) \middle| s_\ell(r) \right] \middle| \mathbf{X}_T \right]$$

$$+ T \cdot \mathbb{E}_{s_\ell(r)} \left[ \mathbb{E}\left[ \mathbf{1}\{\mathcal{G} \cap \mathcal{E}_3^c(s_\ell(r))\} \middle| s_\ell(r) \right] \middle| \mathbf{X}_T \right]$$

We first handle the first double expectation on the R.H.S. above. We have this is at most

$$\leq 2c_{30} \log^2(T) \mathbb{E}_{s_\ell(r)} \left[ \mathbb{E}\left[ \mathbf{1}\{\mathcal{G} \cap \mathcal{E}_3(t_\ell)\} \sum_{i \in \text{PHASES}(\ell,r)} K^{\frac{1}{2+d}} (\tau_{i+1} - \tau_i)^{\frac{1+d}{2+d}} \middle| s_\ell(r) \right] \middle| \mathbf{X}_T \right]$$

$$\leq 2c_{30} \log^2(T) \mathbb{E}\left[ \mathbf{1}\{\mathcal{G}\} \sum_{i \in \text{PHASES}(\ell,r)} (\tau_{i+1} - \tau_i)^{\frac{1+d}{2+d}} K^{\frac{1}{2+d}} \middle| \mathbf{X}_T \right],$$

where in the last step we bound $\mathbf{1}\{\mathcal{G} \cap \mathcal{E}_3(s_\ell(r))\} \leq \mathbf{1}\{\mathcal{G}\}$ and apply tower law again. The above R.H.S. is of the right order w.r.t. (19). So, it remains to bound

$$T \cdot \mathbb{E}_{s_\ell(r)} \left[ \mathbb{E}\left[ \mathbf{1}\{\mathcal{G} \cap \mathcal{E}_3^c(s_\ell(r))\} \middle| s_\ell(r) \right] \middle| \mathbf{X}_T \right].$$

We first observe that events $\mathcal{G}$ and $\mathcal{E}_3^c(s_\ell(r))$ are independent conditional on $\mathbf{X}_T$ and $s_\ell(r)$ since the former event only depends on the distribution of $Y_{s_\ell(r)}, \ldots, Y_T$ while the latter event depends on the distribution of the Bernoulli's $\{Z_{M,t}\}_{t>s_\ell(r),M}$ (which are independent). Then, writing $\mathbf{1}\{\mathcal{G} \cap \mathcal{E}_3^c(s_\ell(r))\} = \mathbf{1}\{\mathcal{G}\} \cdot \mathbf{1}\{\mathcal{E}_3^c(s_\ell(r))\}$, the above becomes at most $\mathbb{E}[\mathbf{1}\{\mathcal{G}\} \cdot (K/T^2) \mid \mathbf{X}_T]$ which is also of the right order w.r.t. (19).

# E  Proof of Corollary 5

The proof of Corollary 5 will follow in a similar fashion to the proof of Corollary 2 in Suk and Kpotufe [2022], which relates the total-variation rates to significant shifts in the non-stationary MAB setting. A novel difficulty here is that our notion of significant shift $\tau_i(\mathbf{X}_T), \tilde{L}(\mathbf{X}_T)$ (Definition 6) depends on the full context sequence $\mathbf{X}_T$, and so it is not clear how the (random) significant phases $[\tau_i(\mathbf{X}_T), \tau_{i+1}(\mathbf{X}_T))$ relate to the total-variation $V_T$, which is a deterministic quantity.

Our strategy will be to first convert the regret rate of Theorem 3 into one which depends on a weaker *worst-case notion of significant shift* which does not depend on the observed $\mathbf{X}_T$. Although this notion of shift is weaker, it will be easier to relate to the total-variation quantity $V_T$.

Recall that $\delta_t^a(x) \doteq \max_{a' \in [K]} \delta_t^{a',a}(x)$ and $\delta_t^{a',a}(x) \doteq f_t^{a'}(x) - f_t^a(x)$ are the gap functions in mean rewards.

**Definition 14** (worst-case sig shift). *Let $\tau_0 = 1$. Then, recursively for $i \geq 0$, the $(i+1)$-th* **worst-case significant shift** *is recorded at time $\tilde{\tau}_{i+1}$, which denotes the earliest time $\tilde{\tau} \in (\tilde{\tau}_i, T]$ such that there exists $x \in \mathcal{X}$ such that* **for every arm** *$a \in [K]$, there exists round $s \in [\tilde{\tau}_i, \tilde{\tau}]$, such that*

$$\delta_s^a(x) \geq \left( \frac{K}{t - \tilde{\tau}_i} \right)^{\frac{1}{2+d}}.$$

*We will refer to intervals $[\tilde{\tau}_i, \tilde{\tau}_{i+1})$, $i \geq 0$, as **worst-case (significant) phases**. The unknown number of such phases (by time $T$) is denoted $\tilde{L}_{\mathrm{pop}} + 1$, whereby $[\tilde{\tau}_{\tilde{L}_{\mathrm{pop}}}, \tilde{\tau}_{\tilde{L}_{\mathrm{pop}}+1})$, for $\tau_{\tilde{L}_{\mathrm{pop}}+1} \doteq T+1$, denotes the last phase.*

We next claim that

$$\mathbb{E}_{\mathbf{X}_T}\left[\sum_{i=0}^{\tilde{L}(\mathbf{X}_T)} (\tau_{i+1}(\mathbf{X}_T) - \tau_i(\mathbf{X}_T))^{\frac{1+d}{2+d}}\right] \leq c_{24} \sum_{i=0}^{\tilde{L}_{\mathrm{pop}}} (\tilde{\tau}_{i+1} - \tilde{\tau}_i)^{\frac{1+d}{2+d}}.$$

This follows since the experienced significant phases $[\tau_i(\mathbf{X}_T), \tau_{i+1}(\mathbf{X}_T))$ interleave the population analogues $[\tilde{\tau}_i, \tilde{\tau}_{i+1})$ in the following sense: at each significant shift $\tau_{i+1}(\mathbf{X}_T)$, for each arm $a \in [K]$, there is a round $s \in [\tau_i(\mathbf{X}_T), \tau_{i+1}(\mathbf{X}_T)]$ such that for $\delta_s(X_{\tau_{i+1}}) > \left(\frac{K}{\tau_{i+1}-\tau_i}\right)^{\frac{1}{2+d}}$. This means there must be a worst-case significant shift $\tilde{\tau}_j$ in the interval $[\tau_i(\mathbf{X}_T), \tau_{i+1}(\mathbf{X}_T)]$ since the criterion of Definition 14 is triggered at $x = X_{\tau_{i+1}}$. This in turn allows us to conclude that each worst-case significant phase $[\tilde{\tau}_i, \tilde{\tau}_{i+1})$ can intersect at most two significant phases $[\tau_i(\mathbf{X}_T), \tau_{i+1}(\mathbf{X}_T))$.

Thus, by the sub-additivity of the function $x \mapsto x^{\frac{1+d}{2+d}}$ (dropping dependence on $\mathbf{X}_T$ in $\tilde{L}, \tau_i$ to ease notation):

$$\sum_{i=0}^{\tilde{L}} (\tau_{i+1} - \tau_i)^{\frac{1+d}{2+d}} \leq \sum_{i=0}^{\tilde{L}} \sum_{j:[\tilde{\tau}_j,\tilde{\tau}_{j+1})\cap[\tau_i,\tau_{i+1})\neq\emptyset} |[\tilde{\tau}_j, \tilde{\tau}_{j+1}) \cap [\tau_i, \tau_{i+1})|^{\frac{1+d}{2+d}}$$

$$\leq c_{24} \sum_{j=0}^{\tilde{L}_{\mathrm{pop}}} (\tilde{\tau}_{j+1} - \tilde{\tau}_j)^{\frac{1+d}{2+d}},$$

where we use Jensen's inequality for $a^p + b^p \leq 2^{1-p}(a+b)^p$ for $p \in (0,1)$ and $a, b \geq 0$ in the last step to re-combine the subintervals of each worst-case significant phase $[\tilde{\tau}_j, \tilde{\tau}_{j+1})$.

Then, it suffices to show

$$\sum_{j=0}^{\tilde{L}_{\mathrm{pop}}} (\tilde{\tau}_{j+1} - \tilde{\tau}_j)^{\frac{1+d}{2+d}} K^{\frac{1}{2+d}} \lesssim T^{\frac{1+d}{2+d}} \cdot K^{\frac{1}{2+d}} + (V_T \cdot K)^{\frac{1}{3+d}} \cdot T^{\frac{2+d}{3+d}}. \tag{36}$$

To start, fix a worst-case significant phase $[\tilde{\tau}_i, \tilde{\tau}_{i+1})$ such that $\tau_{i+1} < T+1$. By Definition 14, there exists a context $x_i \in \mathcal{X}$ such that for arm $a_i \in \mathrm{argmax}_{a \in [K]} f^a_{\tilde{\tau}_{i+1}}(x_i)$ we have there exists a round $t_i \in [\tau_i, \tau_{i+1}]$ such that:

$$\delta^{a_i}_{t_i}(x_i) > \left(\frac{K}{\tilde{\tau}_{i+1} - \tilde{\tau}_i}\right)^{\frac{1}{2+d}}.$$

On the other hand, $\delta^{a_i}_{\tilde{\tau}_{i+1}}(x_i) = 0$ by the definition of arm $a_i$ being the best at $x_i$ at round $\tilde{\tau}_{i+1}$. Thus, letting $\tilde{a}_i \in \mathrm{argmax}_{a \in [K]} f^a_{t_i}(x_i)$ be an optimal arm at $x_i$ at round $t_i$, we have:

$$\left(\frac{K}{\tilde{\tau}_{i+1} - \tilde{\tau}_i}\right)^{\frac{1}{2+d}} < \delta^{\tilde{a}_i, a_i}_{t_i}(x_i) - \delta^{\tilde{a}_i, a_i}_{\tilde{\tau}_{i+1}}(x_i) = \sum_{t=t_i}^{\tau_{i+1}-1} \delta^{\tilde{a}_i, a_i}_t(x_i) - \delta^{\tilde{a}_i, a_i}_{t+1}(x_i).$$

For each round $t = 2, \ldots, T$, define the function $H_i : \mathcal{X} \times [0,1]^K \to [-1,1]$ by $H_i(X, Y) \doteq Y^{\tilde{a}_i} - Y^{a_i}$ which is the realized difference in rewards between arms $\tilde{a}_i$ and $a_i$ within the reward vector $Y$. Then, using this notation and summing the above display over worst-case phases $i \in [\tilde{L}_{\mathrm{pop}}]$, we get:

$$\sum_{i=1}^{\tilde{L}_{\mathrm{pop}}} \left(\frac{K}{\tilde{\tau}_{i+1} - \tilde{\tau}_i}\right)^{\frac{1}{2+d}} < \sum_{i=1}^{\tilde{L}_{\mathrm{pop}}} \sum_{t=\tau_{i-1}}^{\tau_i} |\mathbb{E}_{(X_{t-1}, Y_{t-1}) \sim \mathcal{D}_{t-1}}[H_i(X_{t-1}, Y_{t-1})] - \mathbb{E}_{(X_t, Y_t) \sim \mathcal{D}_t}[H_i(X_t, Y_t)]|.$$
$$\tag{37}$$

We next recall from the variational representation of the total variation distance [Polyanskiy and Wu, 2022, Theorem 7.24] that for any measurable function $H : \mathcal{X} \times [0,1]^K \to [-1,1]$,

$$\|\mathcal{D}_t - \mathcal{D}_{t-1}\|_{\mathrm{TV}} \geq \frac{1}{2} \left(\mathbb{E}_{(X_{t-1}, Y_{t-1}) \sim \mathcal{D}_{t-1}}[H(X_{t-1}, Y_{t-1})] - \mathbb{E}_{(X_t, Y_t) \sim \mathcal{D}_t}[H(X_t, Y_t)]\right). \tag{38}$$

Thus, plugging (38) into (37), we get

$$\sum_{i=1}^{\tilde{L}_{\mathrm{pop}}} \left( \frac{K}{\tilde{\tau}_{i+1} - \tilde{\tau}_i} \right)^{\frac{1}{2+d}} \leq 2 \sum_{t=2}^{T} \|\mathcal{D}_t - \mathcal{D}_{t-1}\|_{\mathrm{TV}}. \tag{39}$$

**Remark 7.** *Note that it is crucial in this argument that the functions $H_i$ do not depend on the realized rewards $Y_t$ or contexts $X_t$ at any particular round $t$. Rather, $H_i$ only depends on the arms $\tilde{a}_i, a_i$ which in turn only depend on the mean reward sequence $\{f_t\}_{t \in [T]}$. In other words, the above step does not follow if $a_i, \tilde{a}_i$ were defined in terms of the experienced significant shifts $\tau_i(\mathbf{X}_T)$.*

Now, by Hölder's inequality for $p \in (0, 1)$ and $q \in \left( 0, \frac{1+d}{2+d} \right)$:

$$\sum_{i=1}^{\tilde{L}_{\mathrm{pop}}} (\tilde{\tau}_{i+1} - \tilde{\tau}_i)^{\frac{1+d}{2+d}} K^{\frac{1}{2+d}} \leq T^{\frac{1+d}{2+d}} K^{\frac{1}{2+d}}$$

$$+ \left( \sum_i K^{\frac{1}{2+d}} (\tilde{\tau}_{i+1} - \tilde{\tau}_i)^{-q/p} \right)^p \left( \sum_i K^{\frac{1}{2+d}} (\tilde{\tau}_{i+1} - \tilde{\tau}_i)^{\left( \frac{1+d}{2+d} + q \right) \cdot \frac{1}{1-p}} \right)^{1-p}.$$

In particular, letting $p = \frac{1}{3+d}$ and $q = \frac{1}{(2+d)(3+d)}$ and plugging in our earlier bound (39) makes the above R.H.S.

$$T^{\frac{1+d}{2+d}} \cdot K^{\frac{1}{2+d}} + V_T^{\frac{1}{3+d}} \cdot K^{\frac{1}{3+d}} \cdot T^{\frac{2+d}{3+d}}.$$

$\square$

# F   Proof of Theorem 1

We first note that it suffices to show (3) for integer $L \in [0, T] \cap \mathbb{N}$ as lower bounds for all other $L$ follow via approximation and modifying the constant $c > 0$ in (3). Thus, going forward, fix $V \in [0, T]$ and $L \in \mathbb{Z} \cap [0, T]$.

At a high level, our construction will repeat $L + 1$ times a hard environment for stationary contextual bandits. In particular, within each stationary phase of length $T/(L+1)$ one is forced to pay a regret of $(T/(L+1))^{\frac{1+d}{2+d}}$, summing to a total regret lower bound of $(L+1) \cdot (T/(L+1))^{\frac{1+d}{2+d}} \approx (L+1)^{\frac{1}{2+d}} \cdot T^{\frac{1+d}{2+d}}$.

To obtain the lower bound expressed in terms of total variation budget $V$ in (3), we will choose $L \propto V^{\frac{2+d}{3+d}} \cdot T^{\frac{1}{3+d}}$ and argue that the actual total variation $V_T$ (Definition 5) is at most $V$ in the constructed environments for this choice of $L$. This is similar to the arguments of the analogous dynamic regret lower bound [Besbes et al., 2019, Theorem 1] for the non-contextual bandit problem.

We start by establishing a lower bound for stationary Lipschitz contextual bandits. The construction is identical to that of Rigollet and Zeevi [2010, Theorem 4.1]. We provide the details here to (1) highlight a minor novelty in circumventing the reliance of the cited result on a positive "margin parameter" $\alpha > 0$ and (2) to assist in later calculating the total variation $V_T$.

**Remark 8.** *There are other stationary lower bound results for Lipschitz contextual bandits using similar constructions and arguments, but with context marginal measures $\mu_X$ of finite support [Foster and Rakhlin, 2020, Theorem 2; Slivkins, 2014, Theorem 7]. To contrast, our construction involves setting $\mu_X$ to be uniform on $[0, 1]^d$, thus satisfying the strong density assumption (Assumption 2).*

**Proposition 16.** *Suppose there are $K = 2$ arms. Then, there exists a finite family of stationary Lipschitz contextual bandit environments $\mathcal{E}(n)$ over $n$ rounds such that for any algorithm $\pi$ taking as input random variable $U$, we have for some constant $c > 0$ and environment $\mathcal{E}$ generated uniformly and independently (of $\pi, U$) at random from $\mathcal{E}(n)$:*

$$\mathbb{E}_{\mathcal{E} \sim \mathrm{Unif}(\mathcal{E}(n)), U}[R(\pi, \mathbf{X}_T)] \geq c \cdot n^{\frac{1+d}{2+d}}.$$

*Proof.* Let the covariates $X_t$ be uniformly distributed on $[0, 1]^d$ at each round $t \in [n]$, so that $\mu_X \equiv \mathrm{Unif}\{[0, 1]^d\}$. For ease of presentation, let us reparametrize the two arms as $+1$ and $-1$.

At each round $t \in [n]$, let arm $-1$ have reward $Y_t^{-1} \sim \text{Ber}(1/2)$ and let arm $+1$ have reward $Y_t^{+1} \sim \text{Ber}(f(X_t))$ where $f : \mathcal{X} \to [0,1]$ is some mean reward function to be defined. Let

$$M \doteq \left\lceil \left(\frac{n}{3e}\right)^{\frac{1}{2+d}} \right\rceil.$$

We next partition $\mathcal{X} = [0,1]^d$ into a regular grid of bins with centers $\mathcal{Q} = \{q_1, \ldots, q_{M^d}\}$, where $q_k$ denotes the center of bin $B_k$, $k = 1, \ldots, M^d$. Concretely, re-indexing the bins, for each index $\mathbf{k} \doteq (k_1, \ldots, k_d) \in \{1, \ldots, M\}^d$, we define the bin $B_{\mathbf{k}}$ coordinate-wise as:

$$B_{\mathbf{k}} \doteq \left\{ x \in \mathcal{X} : \frac{k_\ell - 1}{M} \leq x_\ell \leq \frac{k_\ell}{M}, \ell = 1, \ldots, d \right\}.$$

Define $C_\phi \doteq 1/4$. Then, let $\phi : \mathbb{R}^d \to \mathbb{R}_+$ be the smooth function defined by:

$$\phi(x) \doteq \begin{cases} 1 - \|x\|_\infty & 0 \leq \|x\|_\infty \leq 1 \\ 0 & \|x\|_\infty > 1 \end{cases}.$$

It's straightforward to verify $\phi$ is 1-Lipschitz over $\mathbb{R}^d$.

Next, to ease notation, define the integer $m \doteq M^d$. Also, define $\Sigma_m \doteq \{-1, 1\}^m$ and for any $\omega \in \Omega_m$, define the function $f_\omega$ on $[0,1]^d$ via

$$f_\omega(x) \doteq 1/2 + \sum_{j=1}^{m} \omega_j \cdot \phi_j(x),$$

where $\phi_j(x) \doteq M^{-1} \cdot C_\phi \cdot \phi(M \cdot (x - q_j)) \cdot \mathbf{1}\{x \in B_j\}$. Then, the optimal arm at context $x \in \mathcal{X}$ in this environment is given by $\pi_f^*(x) \doteq \text{sgn}(f(x) - 1/2)$ (where we use the convention $\text{sgn}(0) \doteq 1$). If $x \in B_j$, then $\pi_{f_\omega}^*(x) = \omega_j$.

Then, define the family $\mathcal{C}$ of environments induced by $f_\omega$ for $\omega \in \Omega_m$. Note that $f_\omega$ is also 1-Lipschitz for all $\omega$. Next, let $\text{Int}(B_j)$ be the $\ell_\infty$ ball centered at $q_k$ of radius $\frac{1}{2M}$ (i.e., $\text{Int}(B_j)$ is a ball of half the $\ell_\infty$ radius contained in $B_j$). Then, by the definition of $\phi(x)$ above, we have for any $x \in \text{Int}(B_j)$ and any $j \in [m]$:

$$|f_\omega(x) - 1/2| \geq M^{-1} \cdot C_\phi/2.$$

Next, we bound the average regret w.r.t. a uniform prior over the family $\mathcal{C}$ of environments as:

$$\mathbb{E}_{f \in \mathcal{C}} \mathbb{E} \sum_{t=1}^{n} |f(X_t) - 1/2| \cdot \mathbf{1}\{\pi_t(X_t) \neq \pi^*(X_t)\} \geq$$

$$\frac{C_\phi}{2M} \mathbb{E}_{f \in \mathcal{C}} \mathbb{E} \sum_{t=1}^{n} \sum_{j=1}^{m} \mathbf{1}\{\pi_t(X_t) \neq \pi^*(X_t), X_t \in \text{Int}(B_j)\}, \tag{40}$$

where the outer expectation is over an $f \in \mathcal{C}$ chosen uniformly at random from $\mathcal{C}$.

As recall $M \propto n^{\frac{1}{2+d}}$, it will suffice to show the expectation on the above R.H.S. is of order $\Omega(n)$. This will follow essentially the same steps as the reduction to hypothesis testing in the proof of Theorem 4.1 in Rigollet and Zeevi [2010]. In particular, we note the KL divergence calculations for the induced distributions over observed data and decisions allows for the additional randomness $U$ without consequence, as it's ignorable by use of KL chain rule. The only slight modification we make in their argument is to account for the fact that we only count rounds when $X_t \in \text{Int}(B_j)$ rather than when $X_t \in B_j$. This will, in fact, only affect constants in the lower bound as $B_j$ and $\text{Int}(B_j)$ have similar masses.

Going into details, the following notation will be useful.

**Notation 3.** *Let $\omega_{[-j]}$ be $\omega$ with the $j$-th entry removed, and let $\omega_{[-j]}^i = (\omega_1, \ldots, \omega_{j-1}, i, \omega_{j+1}, \ldots, \omega_m)$ for $i \in \{\pm 1\}$. Also, let $\mathbb{P}_{\pi,f}, \mathbb{E}_{\pi,f}$ denote the joint measure and expectation, respectively, over all the randomness of $\pi$ and observations within an environment induced by mean reward function $f$.*

Then, the aforementioned reduction to hypothesis testing in the proof of Theorem 4.1 of Rigollet and Zeevi [2010] yields the following lower bound on (40):

$$\frac{C_\phi}{2^{m+1} \cdot M} \sum_{j=1}^{m} \frac{n}{4M^d} \sum_{\omega_{[-j]} \in \Omega_{m-1}} \exp\left(-\frac{4}{3M^2} \cdot N_{j,\pi}(\omega)\right) + \tilde{N}_{j,\pi}(\omega), \qquad (41)$$

where, letting $X \sim \mu_X$ be an independently drawn context,

$$N_{j,\pi}(\omega) \doteq \mathbb{E}_{\pi, f_{\omega_{[-j]}^{-1}}} \mathbb{E}_X \left[ \sum_{t=1}^{n} \mathbf{1}\{\pi_t(X) = 1, X \in B_j\} \right]$$

$$\tilde{N}_{j,\pi}(\omega) \doteq \mathbb{E}_{\pi, f_{\omega_{[-j]}^{-1}}} \mathbb{E}_X \left[ \sum_{t=1}^{n} \mathbf{1}\{\pi_t(X) = 1, X \in \mathrm{Int}(B_j)\} \right].$$

We next claim $\tilde{N}_{j,\pi}(\omega) = N_{j,\pi}/2^d$, which will follow from swapping the order of expectations and summation in the above formulas. In particular, we have

$$\tilde{N}_{j,\pi}(\omega) = \sum_{t=1}^{n} \mathbb{P}_{\pi, f_{\omega_{[-j]}^{-1}}} (\pi_t(X) = 1 | X \in \mathrm{Int}(B_j)) \cdot \mu_X(X \in \mathrm{Int}(B_j))$$

$$= \sum_{t=1}^{n} \mathbb{P}_{\pi, f_{\omega_{[-j]}^{-1}}} (\pi_t(X) = 1 | X \in \mathrm{Int}(B_j)) \cdot \left(\frac{1}{2M}\right)^d$$

$$= \frac{1}{2^d} \sum_{t=1}^{n} \mathbb{P}_{\pi, f_{\omega_{[-j]}^{-1}}} (\pi_t(X) = 1 | X \in B_j) \cdot \mu_X(X \in B_j)$$

$$= N_{j,\pi}(\omega)/2^d$$

Thus, (41) becomes lower bounded by

$$\frac{C_\phi}{2^{m+1} \cdot M} \sum_{j=1}^{m} \frac{n}{4M^d} \sum_{\omega_{[-j]} \in \Omega_{m-1}} \exp\left(-\frac{4}{3M^2} \cdot N_{j,\pi}(\omega)\right) + \frac{N_{j,\pi}(\omega)}{2^d} \geq$$

$$\frac{C_\phi \cdot m}{8 \cdot 2^d \cdot M} \inf_{z \geq 0} \left\{ \frac{n}{4M^d} \exp\left(-\frac{4}{3M^2} \cdot z\right) + z \right\}.$$

The above R.H.S. is optimized at $z^* \doteq \frac{3M^2}{4} \log\left(\frac{n}{3M^{2+d}}\right)$. Plugging this into the above along with our choice of $M$ defined earlier gives us a lower bound of $\Omega(n^{\frac{1+d}{2+d}})$.

$\square$

Given Proposition 16, the $(L+1) \cdot \left(\frac{T}{L+1}\right)^{\frac{1+d}{2+d}}$ lower bound immediately follows by lower bounding the total regret over a random environment formed by concatenating $L+1$ i.i.d. sampled environments from $\mathrm{Unif}(\mathcal{E}(T/(L+1)))$. Any such resultant environment clearly has at most $L$ global shifts. Note that the average regret (w.r.t. the random environment) over any stationary phase of length $\frac{T}{L+1}$ is lower bounded by $\left(\frac{T}{L+1}\right)^{\frac{1+d}{2+d}}$ regardless of the information learned prior to that phase, as such information can be formalized as exogeneous randomness $U$ in Proposition 16.

Next, we tackle the lower bound $V^{\frac{1}{3+d}} \cdot T^{\frac{2+d}{3+d}}$ in terms of total-variation budget $V$. First, if $V < O(T^{-\frac{1}{2+d}})$, then we're already done as the desired rate

$$\left(T^{\frac{1+d}{2+d}} + T^{\frac{2+d}{3+d}} \cdot V^{\frac{1}{3+d}}\right) \wedge \left((L+1)^{\frac{1}{2+d}} T^{\frac{1+d}{2+d}}\right)$$

is minimized by the first term which is of order $O(T^{\frac{1+d}{2+d}})$. Thus, using Proposition 16 with a single stationary phase $\mathcal{E}(T)$ gives lower bound of the right order in this regime. Such an environment clearly has total-variation $V_T = 0 \leq V$.

Suppose then that $V \geq 2 \cdot T^{-\frac{1}{2+d}}$. Let $\Delta \doteq \left\lceil \left(\frac{T}{V}\right)^{\frac{2+d}{3+d}} \right\rceil \leq T$ and consider $L + 1 = \rho \cdot T/\Delta$ stationary phases of length $\Delta$, for some fixed constant $\rho > 0$. Then, by the previous arguments we have the regret is lower bounded by

$$(L+1)^{\frac{1}{2+d}} \cdot T^{\frac{1+d}{2+d}} = \frac{T}{\Delta^{\frac{1}{2+d}}} \geq \frac{T}{2^{\frac{1}{3+d}} (T/V)^{\frac{1}{3+d}}} \propto T^{\frac{2+d}{3+d}} \cdot V^{\frac{1}{3+d}}.$$

Additionally, $T^{\frac{2+d}{3+d}} \cdot V^{\frac{1}{3+d}}$ dominates $T^{\frac{1+d}{2+d}}$ since $V \geq T^{-\frac{1}{2+d}}$. Thus, the regret lower bound is proven in terms of $V$.

It remains to verify that the total-variation $V_T$ is at most $V$ in the above constructed environments so that any such environment lies in the family $\mathcal{P}(V, L, T)$.

Clearly, the "instantaneous total-variation" $\|\mathcal{D}_t - \mathcal{D}_{t-1}\|_{\text{TV}} = 0$ for all rounds $t$ not being the start of a new stationary phase. On the other hand, for a round $t$ marking the beginning of a new phase, we have that since conditioning increases the total-variation [Polyanskiy and Wu, 2022, Theorem 7.5(c)], the instantaneous total-variation is at most:

$$\|\mathcal{D}_t - \mathcal{D}_{t-1}\|_{\text{TV}} \leq \mathbb{E}_{x \sim \mu_X} \left[ \|\mathcal{D}_t(Y_t|X_t = x) - \mathcal{D}_{t-1}(Y_{t-1}|X_{t-1} = x)\|_{\text{TV}} \right].$$

Since $Y_t^a|X_t = x \sim \text{Ber}(f_t^a(x))$, we have the R.H.S.'s inner TV quantity is just the total variation between Bernoulli's or $\max_{a \in \{\pm 1\}} |f_t^a(x) - f_{t-1}^a(x)|$. Carefully analyzing the variations in the constructed Lipschitz reward functions in the proof of Proposition 16 reveals this TV between Bernoulli's is at most $\frac{(3e)^{\frac{1}{2+d}}}{2} \cdot \left(\frac{L+1}{T}\right)^{\frac{1}{2+d}}$. Then, summing the instantaneous total-variation over phases, we have

$$V_T \leq (L+1) \cdot \frac{(3e)^{\frac{1}{2+d}}}{2} \cdot \left(\frac{L+1}{T}\right)^{\frac{1}{2+d}}$$

$$\leq \rho^{\frac{3+d}{2+d}} (L+1)^{\frac{3+d}{2+d}} \cdot \left(\frac{(3e)^{\frac{1}{2+d}}}{2}\right) \cdot T^{-\frac{1}{2+d}}$$

$$< T \cdot \left(\frac{1}{\Delta}\right)^{\frac{3+d}{2+d}}$$

$$\leq V,$$

where the third inequality follows by letting $\rho$ be a small enough constant. $\qquad \square$

