# Tracking Most Significant Shifts in Nonparametric Contextual Bandits

## Abstract

We study nonparametric contextual bandits where Lipschitz mean reward functions may change over time. We first establish the minimax dynamic regret rate in this less understood setting in terms of number of changes $L$ and total-variation $V$, both capturing all changes in distribution over context space, and argue that state-of-the-art procedures are suboptimal in this setting.

Next, we tend to the question of an *adaptivity* for this setting, i.e. achieving the minimax rate without knowledge of $L$ or $V$. Quite importantly, we posit that the bandit problem, viewed local at a given context $X_t$, should not be affected by reward changes in other parts of context space $\mathcal{X}$. We therefore propose a notion of *change* that better accounts for locality, and thus counts significantly less changes than $L$ and $V$. Our main result is to show that this more strict notion of change, which we term *experienced significant shifts*, can in fact be adapted to. As in previous work on non-stationary MAB (Suk and Kpotufe, 2022), not only do our results capture changes only at the experienced contexts $x$, but also only the most *significant* in terms of changes in mean rewards (e.g., only count severe best-arm changes at $x$).

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

## 2 Problem Formulation

### 2.1 Contextual Bandits with Changing Rewards

**Preliminaries.** We assume a finite set of arms $[K] \doteq \{1, 2 \ldots, K\}$. Let $Y_t \in [0,1]^K$ denote the vector of rewards for arms $a \in [K]$ at round $t \in [T]$ (horizon $T$), and $X_t$ the observed context at that round, lying in $\mathcal{X} \doteq [0,1]^d$, which have joint distribution $(X_t, Y_t) \sim \mathcal{D}_t$. We let $\mathbf{X}_t \doteq \{X_s\}_{s \leq t}, \mathbf{Y}_t \doteq \{Y_s\}_{s \leq t}$ denote the observed contexts and (observed and unobserved) rewards from rounds 1 to $t$. In our setting, an oblivious adversary decides a sequence of (independent) distributions on $\{(X_t, Y_t)\}_{t \in [T]}$ before play.

**Notation.** *The* **reward function** $f_t : \mathcal{X} \rightarrow [0,1]^K$ *is* $f_t^a(x) \doteq \mathbb{E}[Y_t^a | X_t = x]$, $a \in [K]$, *and captures the mean rewards of arm $a$ at context $x$ and time $t$.*

A *policy* chooses actions at each round $t$, based on observed contexts (up to round $t$) and passed rewards, whereby at each round $t$ only the reward $Y_t^a$ of the chosen action $a$ is revealed. Formally:

**Definition 1** (Policy). *A policy* $\pi \doteq \{\pi_t\}_{t \in \mathbb{N}}$ *is a random sequence of functions* $\pi_t : \mathcal{X}^t \times [K]^{t-1} \times [0,1]^{t-1} \rightarrow [K]$. *In the case of a* **randomized** *policy, i.e., where $\pi_t$ in fact

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

**Remark 3** (Significant Shifts Depend on Contexts). *It should be understood that the significant shifts $\tau_i$ and $\tilde{L}$ depend on $\mathbf{X}_T$ and mean rewards $\{f_t^a(X_t)\}_{t \in [T], a \in [K]}$, but not the realized rewards $\mathbf{Y}_T$. For simplicity of presentation, we will not make the dependence on $\mathbf{X}_T$ explicit in most places where $\tau_i, \tilde{L}$ are mentioned.*

It's clear from Definition 6 and (⋆) that only changes in the mean rewards $f_t^a(x)$ at experienced contexts $x \in \mathbf{X}_T$ are counted, and that they are only counted when experienced. Furthermore, an experienced significant shift $\tau_i$ implies a best-arm change at $X_{\tau_i}$ since, by smoothness (Assumption 1), and (⋆) we have

$$\sum_{s \in I} \delta_s^a(X_{\tau_i}) \cdot \mathbf{1}\{X_s \in B\} \geq \sum_{s \in I} \delta_s(a) \cdot \mathbf{1}\{X_s \in B\} - r(B) \sum_{s \in I} \mathbf{1}\{X_s \in B\} > 0.$$

Thus, $\tilde{L} \leq L + 1$, the global count of shifts.

On the other hand, so long as an experienced significant shift does not occur, there will be arms safe to play at each context $X_t$. As a result, procedures need not restart exploration so long as unsafe arms can be quickly ruled out.

As a warmup to presenting our main regret bounds and algorithms, we'll first consider an oracle procedure which restarts only at experienced significant shifts.

**Definition 7** (Oracle Procedure). *For each round $t$ in phase $[\tau_i, \tau_{i+1})$, define a good arm set $\mathcal{G}_t$ as the set of **safe** arms, i.e., arms which do not yet satisfy (⋆) in bin $T_r(X_t)$ for $r = r_{\tau_{i+1} - \tau_i}$ (recall from Subsection 2.3 that this is the oracle choice of level over phase $[\tau_i, \tau_{i+1})$).*

*Then, define an **oracle procedure** $\pi$: at each round $t$, $\pi$ plays a random arm $a \in \mathcal{G}_t$ w.p. $1/|\mathcal{G}_t|$.*

We then claim such an oracle procedure attains an enhanced dynamic regret rate in terms of the significant shifts $\{\tau_i\}_i$ which recovers the minimax lower bound in terms of global number of shifts $L$ and total variation $V_T$ from before.

**Proposition 2** (Sanity Check). *We have the oracle procedure $\pi$ of Definition 7 satisfies with probability at least $1 - 1/T^2$ w.r.t. the randomness of $\mathbf{X}_T$: for some $C > 0$*

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

random schedule (stored in the variable $\{Z_{m,s}\}$). We refer to the base algorithm playing at round $t$ as the *active base algorithm*. This induces a hierarchy of base algorithms, from *parent* to *child* instances of Base-Alg .

**Choice of Level.**  Focusing on a single base algorithm now, each Base-Alg manages its own discretization of the context space $\mathcal{X} = [0,1]^d$, corresponding to a level $r \in \mathcal{R}$ (see Definition 3). Within each bin $B \in \mathcal{T}_r$ at the level $r$, candidate arms, maintained in a set $\mathcal{A}(B)$, are evicted according to estimates (4) of local gaps.

As said earlier in Subsection 2.3, key in attaining optimal regret is using the right level $r \in \mathcal{R}$. An immediate difficulty is that the oracle choice of level used in Definition 7 depends on the unknown significant phase length $\tau_{i+1} - \tau_i$. To circumvent this, as in previous works [Perchet and Rigollet, 2013, Slivkins, 2014], we rely on an adaptive time-varying choice of level $r_t$. Specifically, each base algorithm choose the level $r_{t-t_{\mathrm{start}}}$ based on the time elapsed since the time $t_{\mathrm{start}}$ it was first activated.

**Sharing Information across Base Algorithms.**  Instances of Base-Alg and CMETA share information, in the form of *global variables* as listed below:

- All variables defined in CMETA, namely $t_\ell, t, \{\mathcal{A}_{\mathrm{master}}(B)\}_{B \in \mathcal{T}}, \{Z_{m,t}\}$ (see Lines 3–6 of Algorithm 1).

- All arms played at any round $t$, along with observed rewards $Y_t^a$, and the candidate arm set $\mathcal{A}_t$ which takes the value of the set $\mathcal{A}(B)$ of the active Base-Alg at round $t$ and bin $B = T_r(X_t)$ used.

By sharing these global variables, any Base-Alg can trigger a new episode: every time an arm is evicted from $\mathcal{A}(B)$ a Base-Alg , it is also evicted from $\mathcal{A}_{\mathrm{master}}(B)$, which is essentially the candidate arm set for the current episode. A new episode is triggered at time $t$ when $\mathcal{A}_{\mathrm{master}}(B)$ becomes empty for some bin $B$ (necessarily a currently experienced bin), i.e., there is no *safe* arm left to play at the context $X_t$ in the sense of Definition 6.

Note that $\mathcal{A}(B)$ are *local variables* internal to each Base-Alg (the owner of which will be clear from context in usage).

To ensure consistent behavior while using a time-varying choice of level, we enforce further regularity in arm evictions across $\mathcal{X}$: arms evicted from $\mathcal{A}(B')$ are also evicted from child bins $B \subseteq B'$ to ensure $\mathcal{A}(B) \subseteq \mathcal{A}(B')$.

**Estimating Aggregate Local Gaps.**  The quantity $\sum_{s=s_1}^{s_2} \delta_s(a', a) \cdot \mathbf{1}\{X_s \in B\}$ is estimated as $\sum_{s=s_1}^{s_2} \hat{\delta}_s^B(a', a)$, whereby the relative gap $\delta_s(a', a) \cdot \mathbf{1}\{X_s \in B\}$ is estimated by importance weighting as:

$$\hat{\delta}_s^B(a', a) \doteq |\mathcal{A}_t| \cdot \left( Y_t^{a'} \cdot \mathbf{1}\{\pi_t = a'\} - Y_t^a \cdot \mathbf{1}\{\pi_t = a\} \right) \cdot \mathbf{1}\{a \in \mathcal{A}_t\} \cdot \mathbf{1}\{X_s \in B\}. \quad (4)$$

Note that the above is an unbiased estimate of $\delta_t(a', a) \cdot \mathbf{1}\{X_s \in B\}$ whenever $a'$ and $a$ are both in $\mathcal{A}_t$ at time $t$, conditional on the contexts $X_t$. It then follows that, conditional on $\mathbf{X}_T$, the difference $\sum_{t=s_1}^{s_2} \left( \hat{\delta}_t^B(a', a) \cdot \mathbf{1}\{X_s \in B\} - \delta_t(a', a) \right)$ is a martingale that concentrates at a rate roughly $\sqrt{K \cdot n_B([s_1, s_2])}$, where recall from earlier that $n_B(I) \doteq \sum_{s \in I} \mathbf{1}\{X_s \in I\}$ is the context count in bin $B$ over interval $I$.

ut An arm $a$ is then evicted at round $t$ if, for some fixed $C_0 > 0$ [1], $\exists$ rounds $s_1 < s_2 \leq t$ such that at level $r_{s_2 - s_1}$ and (i.e., the bin at level $r_{s_2 - s_1}$ containing $X_t$) letting $B := T_{s_2 - s_1}(X_t)$ (i.e., the bin at level $r_{s_2 - s_1}$ containing $X_t$)

$$\max_{a' \in [K]} \sum_{s=s_1}^{s_2} \hat{\delta}_s^B(a', a) > \log(T)\sqrt{C_0 \cdot (K n_B([s_1, s_2]) \vee K^2)} + r_{s_2 - s_1} \cdot n_B([s_1, s_2]). \quad (5)$$

---

[1] $C_0 > 0$ needs to be sufficiently large, but is a universal constant free of the horizon $T$ or any distributional parameters.

## 5  Key Technical Highlights of Analysis

While a full analysis is deferred to Appendix D due to space constraints, we highlight some of the key novelties and core points of the analysis.

• **Local Safety in Bins implies Safe Total Regret.**  We first argue that the notion of significant regret ($\star$) within a bin $B$ captures the total regret rates $T^{\frac{1+d}{2+d}}$ we wish to compete with. If ($\star$) holds for no intervals $[s_1, s_2]$ in all bins $B$, arm $a$ would be safe and incur little regret over any $[s_1, s_2]$. As it turns out, bounding the per-bin regret by ($\star$) implies a total regret of $T^{\frac{1+d}{2+d}}$ as seen from the following rough calculation: via concentration and the strong density assumption (Assumption 2) to conflate $n_B([1, T]) \approx r(B)^d \cdot T$ and the fact that there are $\approx r^{-d}$ bins at level $r$, we have:

$$\sum_{B \in T_r} \sqrt{K \cdot n_B([1, T])} + r \cdot n_B([1, T]) \leq K^{1/2} \cdot T^{1/2} \cdot r^{-d/2} + T \cdot r. \tag{6}$$

In particular taking $r \propto (K/T)^{\frac{1}{2+d}}$ makes the above RHS the desired rate $K^{\frac{1}{2+d}} T^{\frac{1+d}{2+d}}$.

• **Significant Regret Threshold is Estimation Error.**  At the same time, the RHS of the definition of significant regret ($\star$) is a variance and bias decomposition of the bound on the (conditional on $\mathbf{X}_T$) error of estimating the cumulative regret $\sum_{s=s_1}^{s_2} \delta_s^a(x) \cdot \mathbf{1}\{X_s \in B\}$ at any context $x \in B$. Thus, intuitively, changes of magnitude above the threshold $\sqrt{K \cdot n_B(I)} + r(B) \cdot n_B(I)$ in ($\star$) are detectable.

So, the notion of significant regret ($\star$) perfectly balances both (1) detection of unsafe arms and (2) regret minimization of playing safe arms.

• **A New Balanced Replay Scheduling.**  As mentioned earlier in Subsection 3.1, previous adaptive works on contextual bandits fail to attain the optimal regret in this setting due to an inappropriate frequency of scheduling replays. We introduce a novel scheduling (Line 6 of Algorithm 1) which carefully balances exploration and fast detection of significant regret in the sense of ($\star$). The chosen rate $(1/m)^{\frac{1}{2+d}} (1/t)^{\frac{1+d}{2+d}}$ comes from the following intuitive calculation. A scheduled replay of duration $m$ will incur an additional regret of about $m^{\frac{1+d}{2+d}}$. Then, summing over all possible replays, the extra regret incurred due to replays is in total roughly upper bounded by

$$\sum_{t=1}^{T} \sum_{m=2,4,\ldots,T} \left(\frac{1}{m}\right)^{\frac{1}{2+d}} \left(\frac{1}{t}\right)^{\frac{1+d}{2+d}} \cdot m^{\frac{1+d}{2+d}} \lesssim \sum_{t=1}^{T} T^{\frac{d}{2+d}} \cdot (1/t)^{\frac{1+d}{2+d}} \lesssim T^{\frac{1+d}{2+d}}.$$

In other words, the cost of replays only incurs extra constants in the regret. Surprisingly, this scheduling rate is also sufficient for detecting significant regret in *any* experienced subregion $B$ of the context space $\mathcal{X}$, i.e. there is no need to do additional exploration on a localized per-bin basis.

Next, a key feature of the analysis is that one need only minimize regret and detect changes at the critical level $r_{s_2-s_1} \propto (K/(s_2 - s_1))^{\frac{1}{2+d}}$. In particular, the following two observations play a major role in bounding the regret.

• **Suffices to Only Check ($\star$) at Critical Levels $r_{s_2-s_1}$.**  At first glance, detecting experienced significant shifts (Definition 6) appears difficult as an arm $a$ may incur significant regret over a different bin $B'$ from the bin $B$ that is currently being used by the algorithm.

This difficulty is further compounded by the fact there may even be missing data problems as arms $a \in \mathcal{A}(B)$ in contention at $B$ may have been evicted from sibling bins of the parent $B' \supset B$, thus preventing reliable estimation of $a$ across $B'$. We in fact show that we only require detecting significant regret in bins $B'$ at the critical level $r_{s_2-s_1}$ and only for the arms still in contention across all of $B'$. In other words, changes at other levels are all accounted for by changes at this critical level.

Additionally, we observe that the calculations in (6) would hold if we were just concerned with checking ($\star$) for intervals $[s_1, s_2]$ and bins $B_{s_2-s_1}$ at level $r_{s_2-s_1} := \left(\frac{K}{s_2-s_1}\right)^{\frac{1}{2+d}}$. Thus, the critical level $r_{s_2-s_1}$ is the key to both regret minimization and experienced significant shift detection

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

first recall a Freedman's inequality, which will help us establish concentration of our gap estimators (Proposition 7).

**Lemma 6** (Theorem 1 of Beygelzimer et al. [2011]). *Let $X_1, \ldots, X_n \in \mathbb{R}$ be a martingale difference sequence with respect to some filtration $\mathcal{F}_0, \mathcal{F}_1, \ldots$. Assume for all $t$ that $X_t \leq R$ a.s.. Then for any $\delta \in (0, 1)$, with probability at least $1 - \delta$, we have:*

$$\sum_{i=1}^{n} X_i \leq (e-1) \left( \sqrt{\log(1/\delta) \sum_{i=1}^{n} \mathbb{E}[X_i^2 | \mathcal{F}_{t-1}]} + R \log(1/\delta) \right). \quad (8)$$

Recall from Section 4 that for round $t$,

$$\hat{\delta}_t^{a',a}(B) \doteq |\mathcal{A}_t| \cdot (Y_t(a') \cdot \mathbf{1}\{\pi_t = a'\} - Y_t(a) \cdot \mathbf{1}\{\pi_t = a\}) \cdot \mathbf{1}\{a \in \mathcal{A}_t\} \cdot \mathbf{1}\{X_t \in B\}.$$

We next apply Lemma 6 to our aggregate estimator from Section 4.

**Proposition 7.** *With probability at least $1 - 1/T^2$ w.r.t. the randomness of $\mathbf{Y}_T, \{\pi_t\}_t \mid \mathbf{X}_T$, we have for all bins $B \in \mathcal{T}$ and rounds $s_1 < s_2$ and all arms $a \in [K]$ that for large enough $c_1 > 0$:*

$$\left| \sum_{s=s_1}^{s_2} \hat{\delta}_s^{i_t,a}(B) - \sum_{s=s_1}^{s_2} \mathbb{E}[\hat{\delta}_s^{a',a}(B)|\mathcal{F}_{s-1}] \right| \leq c_1 \log(T) \left( \sqrt{K \cdot n_B([s_1, s_2])} + K \right), \quad (9)$$

*where $\mathcal{F} \doteq \{\mathcal{F}_t\}_{t=1}^{T}$ is the filtration with $\mathcal{F}_t$ generated by $\{\pi_s, Y_s^{\pi_s}\}_{s=1}^{t}$.*

*Proof.* The proof is similar to the proof of Proposition 3 in Suk and Kpotufe [2022].

The martingale difference $\hat{\delta}_s^{a',a}(B) - \mathbb{E}[\hat{\delta}_s^{a',a}(B) \mid \mathcal{F}_{s-1}]$ is clearly bounded above by $2K$ for all bins $B$, rounds $s$, and all arms $a, a'$. We also have a cumulative variance bound:

$$\sum_{s=s_1}^{s_2} \mathbb{E}[(\hat{\delta}_s^{a',a}(B))^2 \mid \mathcal{F}_{s-1}] \leq \sum_{s=s_1}^{s_2} \mathbf{1}\{X_s \in B\} \cdot |\mathcal{A}_s|^2 \cdot \mathbb{E}[\mathbf{1}\{\pi_s = a \text{ or } a'\}|\mathcal{F}_{s-1}]$$

$$\leq \sum_{s=s_1}^{s_2} \mathbf{1}\{X_s \in B\} \cdot 2|\mathcal{A}_s|$$

$$\leq 2K \cdot n_B([s_1, s_2]).$$

Then, the result follows from (8), and taking union bounds over bins $B$ (at most $T$ levels and at most $T$ bins per level), arms $a, a'$, and rounds $s_1, s_2$. $\square$

Since the error probability of Proposition 7 is negligible with respect to regret, we assume going forward in the analysis that (9) holds for all arms $a, a' \in [K]$ and rounds $s_1, s_2$. Specifically, let $\mathcal{E}_1$ be the good event over which the bounds of Proposition 7 hold for all all arms and intervals $[s_1, s_2]$.

## B.2 Concentration of Covariate Counts

**Notation.** *To ease notation throughout, we'll henceforth use $\mu(\cdot)$ to refer to the context marginal distribution $\mu_X(\cdot)$.*

**Lemma 8.** *Let $\{i_t\}_{t=1}^{T}$ be a random sequence of arms whose distribution depends on $\mathbf{X}_T$. With probability at least $1 - 1/T^2$ w.r.t. the randomness of $\mathbf{X}_T$, we have for all bins $B \in \mathcal{T}$, all arms $a', a \in [K]$, and rounds $s_1 < s_2$, for some large enough $c_2 > 0$ the following inequalities hold:*

$$|n_B([s_1, s_2]) - (s_2 - s_1 + 1) \cdot \mu(B)| \leq c_2 \left( \log(T) + \sqrt{\log(T)\mu(B) \cdot (s_2 - s_1 + 1)} \right) \quad (10)$$

$$\left| \sum_{s=s_1}^{s_2} \delta_s(i_s, a) \cdot (\mathbf{1}\{X_s \in B\} - \mu_s(B)) \right| \leq c_2 \left( \log(T) + \sqrt{\log(T)\mu(B) \cdot (s_2 - s_1 + 1)} \right) \quad (11)$$

$$\left| \sum_{s=s_1}^{s_2} \delta_s(a) \cdot (\mathbf{1}\{X_s \in B\} - \mu_s(B)) \right| \leq c_2 \left( \log(T) + \sqrt{\log(T)\mu(B) \cdot (s_2 - s_1 + 1)} \right) \quad (12)$$

*Proof.* The first inequality (10) follow from Lemma 6 since $\sum_{s=s_1}^{s_2} \mathbf{1}\{X_s \in B\} - \mu(B)$ is a martingale, which has conditional variance at most $(s_2 - s_1 + 1) \cdot \mu(B)$.

The other two inequalities are trickier since $\delta_s(a)$ depends on $X_s$ (so that the summand may not be a martingale difference) while $\delta_s(i_s, a)$ may not even be adapted to the canonical filtration generated by $\mathbf{X}_T$ (i.e., $i_t$ may depend on $X_s$ for $s > t$). Nevertheless, we observe that for any random variable $W_s = W_s(\mathbf{X}_T) \in [-1, 1]$:

$$-(\mathbf{1}\{X_t \in B\} - \mu(B)) \leq W_t \cdot (\mathbf{1}\{X_t \in B\} - \mu(B)) \leq \mathbf{1}\{X_t \in B\} - \mu(B).$$

The upper and lower bounds above are both martingale differences with respect to the canonical filtration of $\mathbf{X}_T$ and thus, summing the above over $t$ we have via Lemma 6:

$$\left| \sum_{s=s_1}^{s_2} W_s \cdot (\mathbf{1}\{X_t \in B\} - \mu(B)) \right| \leq \left| \sum_{s=s_1}^{s_2} \mathbf{1}\{X_s \in B\} - \mu(B) \right|$$
$$\leq c_2 \left( \log(T) + \sqrt{\log(T)\mu(B) \cdot (s_2 - s_1 + 1)} \right).$$

Then, taking union bounds over rounds $s_1, s_2$, bins $B \in \mathcal{T}$, and arms $a \in [K]$ gives the result. $\square$

**Notation 2** (good event). *Let $\mathcal{E}_1$ be the good event over which the bounds of Proposition 7 hold for all rounds $s_1, s_2 \in [T]$ and arms $a', a \in [K]$. Thus, on $\mathcal{E}_1$, our estimated gaps in each bin will concentrate.*

*Let $\mathcal{E}_2$ be the good event on which bounds of Lemma 8 holds for all bins $B$, arms $a \in [K]$, rounds $s_1, s_2 \in [T]$. Thus, on $\mathcal{E}_2$, our covariate counts $n_B([s_1, s_2])$ will concentrate and we will be able to relate the empirical quantities $\sum_{s=s_1}^{s_2} \delta_s(a) \cdot \mathbf{1}\{X_s \in B\}$ with their expectations.*

Next, we establish a lemma which allow us to relate significant regret ($\star$) and thus our eviction criterion (5) between different bins and levels.

**Lemma 9** (Relating Aggregate Gaps Between Levels). *On event $\mathcal{E}_2$, if for rounds $s_1 < s_2$, bin $B'$ at level $r_{s_2 - s_1}$ and arm $a$, for some $c_3 > 0$:*

$$\sum_{s=s_1}^{s_2} \delta_s(a) \cdot \mathbf{1}\{X_s \in B'\} \leq c_3 \left( \sqrt{K \cdot n_{B'}([s_1, s_2]) \vee K^2} + r(B') \cdot n_{B'}([s_1, s_2]) \right),$$

*then for any bin $B \subseteq B'$ and some $c_4 > 0$:*

$$\sum_{s=s_1}^{s_2} \delta_s(a) \cdot \mathbf{1}\{X_s \in B\} \leq c_4 \left( \log^{1/2}(T) \cdot r(B)^d \cdot K^{\frac{1}{2+d}} \cdot (s_2 - s_1)^{\frac{1+d}{2+d}} + K\log(T) + \sqrt{\log(T)\mu(B)(s_2 - s_1 + 1)} \right).$$

*The same applies for $\delta_s(a)$ replaced with $\delta_s(a', a)$ with any other fixed arm $a'$.*

*Proof.* We have using (12) and the strong density assumption (Assumption 2):

$$\sum_{s=s_1}^{s_2} \delta_s(a) \cdot \mathbf{1}\{X_s \in B\} \leq \sum_{s=s_1}^{s_2} \delta_s(a) \cdot \mu(B) + c_2 \left( \log(T) + \sqrt{\log(T)(s_2 - s_1 + 1) \cdot \mu(B)} \right)$$
$$\leq \frac{r(B)^d}{r(B')^d} \sum_{s=s_1}^{s_2} \delta_s(a) \cdot \mu(B') + c_2 \left( \log(T) + \sqrt{\log(T)(s_2 - s_1 + 1) \cdot \mu(B)} \right)$$

(13)

Again using (12)

$$\sum_{s=s_1}^{s_2} \delta_s(a) \cdot \mu_s(B') \leq \sum_{s=s_1}^{s_2} \delta_s(a) \cdot \mathbf{1}\{X_s \in B'\} + c_2 \left( \log(T) + \sqrt{\log(T)(s_2 - s_1 + 1) \cdot \mu(B')} \right)$$
$$\leq c_5 \left( \sqrt{K \cdot n_{B'}([s_1, s_2]) \vee K^2} + r(B') \cdot n_{B'}([s_1, s_2]) \right.$$
$$\left. + \log(T) + \sqrt{\log(T)(s_2 - s_1 + 1) \cdot \mu(B')} \right).$$

Next, applying (10) to $n_{B'}([s_1, s_2])$ and using the strong density assumption (Assumption 2) to bound the mass $\mu(B')$ above by $C_d \cdot r(B')^d$, the above R.H.S. is further upper bounded by

$$c_6 \left( \log^{1/2}(T) K^{\frac{1+d}{2+d}} \cdot (s_2 - s_1)^{\frac{1}{2+d}} + K \log(T) \right). \tag{14}$$

Finally, plugging (14) into (13) and using the fact that $(r(B')/2)^d \geq (K/(s_2 - s_1))^{\frac{d}{2+d}}$, we have that (13) is of the desired order. The proof of the same inequalities with $\delta_s(a', a)$ is analogous. $\square$

The following lemma relating the bias and variance terms in the notion of significant regret ($\star$) will serve useful many places in the analysis. They all follow from concentration and similar calculations via the strong density assumption (Assumption 2) as done previously.

**Lemma 10** (bias-variance bound and strong density). *Let $r = r_{s_2 - s_1}$. Then, for any bin $B \in T_r$:*

$$c_7 (s_2 - s_1)^{\frac{1}{2+d}} \cdot K^{\frac{d/2}{2+d}} \leq \sqrt{(s_2 - s_1 + 1) \cdot \mu(B)} \leq c_8 (s_2 - s_1)^{\frac{1}{2+d}} \cdot K^{\frac{d/2}{2+d}}$$

$$\sqrt{n_B([s_1, s_2])} \leq c_9 (s_2 - s_1)^{\frac{1}{2+d}} \cdot K^{\frac{d/2}{2+d}}$$

$$c_{10} (s_2 - s_1)^{\frac{1}{2+d}} \cdot K^{\frac{1+d}{2+d}} \leq n_B([s_1, s_2]) \cdot r \leq c_{11} (s_2 - s_1)^{\frac{1}{2+d}} \cdot K^{\frac{1+d}{2+d}}$$

## B.3 Useful Facts about Levels $r \in \mathcal{R}$ and Blocks $[s_\ell(r), e_\ell(r)]$

The following basic facts about the level selection procedure on Line 2 of Algorithm 2 will be useful as we decompose the analysis into the blocks, or different periods of rounds, where different levels are used. The proofs all follow from Notation 1 and basic calculations.

**Fact 1** (relating level to interval length). *The level $r_{s_2 - s_1} = 2^{-m}$ satisfies for $s_2 - s_1 \geq K$:*

$$2^{-(m-1)} > \left( \frac{K}{s_2 - s_1} \right)^{\frac{1}{2+d}} \geq 2^{-m},$$

*and hence*

$$K \cdot 2^{(m-1)(2+d)} < s_2 - s_1 \leq K \cdot 2^{m(2+d)}.$$

**Fact 2** (the first block). *The first block $[s_\ell(1), e_\ell(1)]$ consists of rounds $[t_\ell, t_\ell + K]$.*

**Fact 3** (start and end times of a block). *For $r < 1$, the **start time** or first round $s_\ell(r)$ of the block corresponding to level $r$ in episode $[t_\ell, t_{\ell+1})$ is $s_\ell(r) = t_\ell + \lceil K \cdot (2r)^{-(2+d)} \rceil$ and the **anticipated end time** or last round of the block is $e_\ell(r) = t_\ell + \lceil K \cdot r^{-(2+d)} \rceil - 1$.*

**Fact 4** (length of a block). *Each block $[s_\ell(r), e_\ell(r)]$ is at least $K$ rounds long. For the first block $[s_\ell(1), e_\ell(1)]$, this is already clear. Otherwise, suppose $r < 1$ in which case:*

$$e_\ell(r) - s_\ell(r) + 1 = \lceil K \cdot r^{-(2+d)} \rceil - \lceil K \cdot (2r)^{-(2+d)} \

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

813 was active and not yet permanently interrupted by round $t$. Thus, it must be the case that $t \in$
814 $[s', M(s', m')]$.

815 Overloading notation, we'll let $\mathcal{A}_t(B)$ be the value of $\mathcal{A}(B)$ for the Base-Alg active at round $t$. Next,
816 note that every round $t \in [s, M(s, m)]$ for a proper or subproper Base-Alg $(s, m)$ is clearly a round
817 where $a \in \mathcal{A}_t(B)$ and no such round is accounted for twice by Proposition 12. Thus,

$$\{t \in (t_r^a, e_\ell(r)] : a \in \mathcal{A}_t(B)\} = \bigsqcup_{\textsf{Base-Alg}\,(s,m)\in\textsc{Proper}(s_\ell(r),e_\ell(r))\cup\textsc{SubProper}(s_\ell(r),s_\ell(r))} [s, M(s, m)].$$

818 Then, we can rewrite the second double sum in (21) as:

$$\sum_{a=1}^{K} \sum_{\textsf{Base-Alg}\,(s,m)\in\textsc{Proper}(s_\ell(r),e_\ell(r))\cup\textsc{SubProper}(s_\ell(r),s_\ell(r))} Z_{m,s} \cdot \sum_{t=s\vee t_r^a}^{M(s,m)} \frac{\delta_t(a_r(B), a)}{|\mathcal{A}_t|} \cdot \mathbf{1}\{X_t \in B\}.$$

819 Recall in the above that the Bernoulli $Z_{m,s}$ (see Line 6 of Algorithm 1) decides whether
820 Base-Alg $(s, m)$ is scheduled.

821 Further bounding the sum over $t$ above by its positive part, we can expand the sum over
822 Base-Alg $(s, m) \in \textsc{Proper}(t_\ell, t_{\ell+1}) \cup \textsc{SubProper}(s_\ell(r), s_\ell(r))$ to be over all Base-Alg $(s, m)$,
823 or obtain:

$$\sum_{a=1}^{K} \sum_{\textsf{Base-Alg}\,(s,m)} Z_{m,s} \cdot \left(\sum_{t=s\vee t_r^a}^{M(s,m)} \frac{\delta_t(a_r(B), a)}{|\mathcal{A}_t|} \cdot \mathbf{1}\{X_t \in B\}\right)_+,$$

824 where the sum is over all replays Base-Alg $(s, m)$, i.e. $s \in \{t_\ell + 1, \ldots, t_{\ell+1} - 1\}$ and $m \in$
825 $\{2, 4, \ldots, 2^{\lceil\log(T)\rceil}\}$. It then remains to bound the contributed relative regret of each Base-Alg $(s, m)$
826 in the interval $[s \vee t_r^a, M(s, m)]$, which will follow similarly to the previous steps.

827 We first have using similar arguments as before (now overloading the notation $M(s, m)$ as $M(s, m, a)$
828 for clarity), i.e. combining our concentration bound (9) with the eviction criterion (5) and applying
829 Lemma 9:

$$\sum_{t=s\vee t_r^a}^{M(s,m)} \frac{\delta_t(a_r(B), a)}{|\mathcal{A}_t|} \cdot \mathbf{1}\{X_t \in B\} \leq \frac{c_5 \left(\log^{1/2}(T) \cdot r^d \cdot K^{\frac{1}{2+d}} \cdot m^{\frac{1+d}{2+d}} + K\log(T) + \sqrt{\log(T)(M(s,m) - s)\mu(B)}\right)}{\min_{t\in[s,M(s,m,a)]} |\mathcal{A}_t|}$$

830 Thus, it remains to bound

$$\sum_{a=1}^{K} \sum_{\textsf{Base-Alg}\,(s,m)} Z_{m,s} \cdot \left(\frac{c_5 \left(\log^{1/2}(T) \cdot r^d \cdot K^{\frac{1}{2+d}} \cdot m^{\frac{1+d}{2+d}} + K\log(T) + \sqrt{\log(T)(M(s,m) - s)\cdot\mu(B)}\right)}{\min_{t\in[s,M(s,m,a)]} |\mathcal{A}_t|}\right).$$

831 Swapping the outer two sums and recognizing that $\sum_{a=1}^{K} \frac{1}{\min_{t\in[s,M(s,m,a)]} |\mathcal{A}_t|} \leq \log(K)$ by similar
832 arguments to beforeby summing over arms in the order they are evicted by Base-Alg $(s, m)$, we have
833 that it remains to bound

$$\sum_{\textsf{Base-Alg}\,(s,m)} Z_{m,s} \cdot c_5 \left(\log^{1/2}(T) \cdot r^d \cdot K^{\frac{1}{2+d}} \cdot \tilde{m}^{\frac{1+d}{2+d}} + K\log(T) + \sqrt{\log(T)(m - s)\mu(B)}\right),$$

(23)

834 where $\tilde{m} \doteq m \wedge (e_\ell(r) - s_\ell(r))$ (note we may restrict attention to the part of replays in the current
835 block $[s_\ell(r), e_\ell(r)]$). Let

$$R(m, B) \doteq \left(c_5 \left(\log^{1/2}(T) \cdot r^d \cdot K^{\frac{1}{2+d}} \cdot \tilde{m}^{\frac{1+d}{2+d}} + K\log(T) + \sqrt{\log(T) \cdot \tilde{m} \cdot r^d}\right)\right) \wedge n_B([s, s+m]).$$

836 Then, in light of the previous calculations, $R(m, B)$ is an upper bound on the within-bin $B$ regret
837 contributed by a replay of total duration $m$ (note we can always coarsely upper bound this regret by
838 $n_B([s, s + m])$.

839 Then, plugging $R(m, B)$ into (23) gives via tower law (we remove the "conditional on $\mathbf{X}_T$" part for
840 ease of presentation):

$$\mathbb{E}\left[\mathbb{E}\left[\sum_{\textsf{Base-Alg}\,(s,m)} Z_{m,s} \cdot R(m, B)\,\bigg|\,s_\ell(r)\right]\right] = \mathbb{E}\left[\sum_{s=s_\ell(r)}^{T} \sum_{m} \mathbb{E}[Z_{m,s} \cdot \mathbf{1}\{s \leq e_\ell(r)\} \mid s_\ell(r)] \cdot R(m, B)\right]$$

Next, we observe that $Z_{m,s}$ and $\mathbf{1}\{s \leq e_\ell(r)\}$ are independent conditional on $t_\ell$ since $\mathbf{1}\{s \leq e_\ell(r)\}$ only depends on the scheduling and observations of base algorithms scheduled before round $s$. Thus, recalling that $\mathbb{P}(Z_{m,s} = 1) = \mathbb{E}[Z_{m,s} \mid t_\ell] = \left(\frac{1}{m}\right)^{\frac{1}{2+d}} \cdot \left(\frac{1}{s-t_\ell}\right)^{\frac{1+d}{2+d}}$,

$$
\mathbb{E}[Z_{m,s} \cdot \mathbf{1}\{s \leq e_\ell(r)\} \mid t_\ell] = \mathbb{E}[Z_{m,s} \mid t_\ell] \cdot \mathbb{E}[\mathbf{1}\{s \leq e_\ell(r)\} \mid s_\ell(r)]
$$
$$
= \left(\frac{1}{m}\right)^{\frac{1}{2+d}} \cdot \left(\frac{1}{s - t_\ell}\right)^{\frac{1+d}{2+d}} \cdot \mathbb{E}[\mathbf{1}\{s \leq e_\ell(r)\} \mid s_\ell(r)].
$$

Plugging this into our expectation from before and unconditioning, we obtain:

$$
\mathbb{E}\left[ \sum_{s=s_\ell(r)}^{e_\ell(r)} \sum_{n=1}^{\lceil \log(T) \rceil} \left(\frac{1}{2^n}\right)^{\frac{1}{2+d}} \left(\frac{1}{s - t_\ell}\right)^{\frac{1+d}{2+d}} \cdot R(2^n, B) \right] \tag{24}
$$

We first evaluate the inner sum over $n$. Note that

$$
\sum_{n=1}^{\lceil \log(T) \rceil} \left(\frac{1}{2^n}\right)^{\frac{1}{2+d}} \cdot (2^n \wedge (e_\ell(r) - s_\ell(r))^{\frac{1+d}{2+d}} \leq \log(T) \cdot (e_\ell(r) - s_\ell(r))^{\frac{d}{2+d}}
$$
$$
\sum_{n=1}^{\lceil \log(T) \rceil} \left(\frac{1}{2^n}\right)^{\frac{1}{2+d}} \sqrt{2^n \wedge (e_\ell(r) - s_\ell(r))} \leq (e_\ell(r) - s_\ell(r))^{\frac{d/2}{2+d}}
$$
$$
\sum_{n=1}^{\lceil \log(T) \rceil} \left(\frac{1}{2^n}\right)^{\frac{1}{2+d}} (K \wedge 2^n) \leq \log(T) \cdot K^{\frac{1+d}{2+d}}.
$$

Multiplying by $(s - t_\ell)^{-\frac{1+d}{2+d}}$ and taking a further sum over $s \in [s_\ell(r), e_\ell(r)]$ in the above display, (24) becomes

$$
(e_\ell(r) - t_\ell)^{\frac{1}{2+d}} \left( (e_\ell(r) - s_\ell(r))^{\frac{d}{2+d}} K^{\frac{1}{2+d}} \cdot r^d + (e_\ell(r) - s_\ell(r))^{\frac{d/2}{2+d}} \sqrt{\log(T) \cdot r^d} + K^{\frac{1+d}{2+d}} \log(T) \right).
$$

We have the first term inside the paranetheses above inside dominates the second term as long as $K \geq \log(T)$.

Next, note from Fact 4 that $e_\ell(r) - t_\ell \leq c_{13}(e_\ell(r) - s_\ell(r))$ and so the above is at most:

$$
r^d \cdot (e_\ell(r) - s_\ell(r))^{\frac{1+d}{2+d}} K^{\frac{1}{2+d}} + \log(T) K^{\frac{1+d}{2+d}} \cdot (e_\ell(r) - s_\ell(r))^{\frac{1}{2+d}}. \tag{25}
$$

We next recall from Fact 4 that each block $[s_\ell(r), e_\ell(r)]$ is at least $K$ rounds long. Thus,

$$
C_d \cdot r^d \cdot (e_\ell(r) - s_\ell(r))^{\frac{1+d}{2+d}} \cdot K^{\frac{1}{2+d}} \geq c_{21} \cdot (e_\ell(r) - s_\ell(r))^{\frac{1}{2+d}} \cdot K^{\frac{1+d}{2+d}}.
$$

Thus, the second term of (25) is at most $\log(T)$ times the first term.

Showing (a) is order (19) then follows from upper bounding $e_\ell(r) - s_\ell(r)$ by the combined length of all phases $[\tau_i, \tau_{i+1})$ intersecting block $[s_\ell(r), e_\ell(r)]$, and using the sub-additivity of $x \mapsto x^{\frac{1+d}{2+d}}$.

• **Bounding the Regret of the Last Master Arm $a_r(B)$ to the Last Safe Arm $a_t^\sharp$.** Before we proceed, we first convert $\sum_{t=s_\ell(r)}^{e_\ell(r)} \delta_t(a_t^\sharp, a_r(B)) \cdot \mathbf{1}\{X_t \in B\}$ into a more convenient form in terms of the bin-masses $\mu(B)$. By concentration (11) of Proposition 7, we have

$$
\sum_t \delta_t(a_t^\sharp, a_r(B)) \cdot \mathbf{1}\{X_t \in B\} \leq \sum_t \delta_t(a_t^\sharp, a_r(B)) \cdot \mu(B) + c_1 \left( \log(T) + \sqrt{\log(T)(e_\ell(r) - s_\ell(r)) \cdot \mu(B)} \right).
$$

By Lemma 10, we have

$$
\sqrt{(e_\ell(r) - s_\ell(r)) \cdot \mu(B)} \leq r^d \cdot (e_\ell(r) - s_\ell(r))^{\frac{1+d}{2+d}} K^{\frac{1}{2+d}}.
$$

Additionally, $\log(T)$ is of the right order with respect to (20). Thus, the concentration error terms from Proposition 7 above are negligible.

Moving forward, by the strong density assumption and in light of (19), it suffices to show

$$\sum_{t=s_\ell(r)}^{e_\ell(r)} \delta_t(a_t^\sharp, a_r(B)) \lesssim \sum_{i \in \text{PHASES}(\ell,r)} (\tau_{i+1} - \tau_i)^{\frac{1+d}{2+d}} K^{\frac{1}{2+d}}.$$

This is the most difficult quantity to bound since arm $a_t^\sharp$ may have been evicted from $\mathcal{A}_{\text{master}}(B)$ before round $t$ and, thus, we rely on our replay scheduling to bound the regret incurred while waiting to detect a large aggregate value of $\delta_t(a_t^\sharp, a_r(B))$.

For each phase $[\tau_i, \tau_{i+1})$ which intersects the remaining rounds $[s_\ell(r), e_\ell(r)]$ (in an abuse of notation, we'll conflate $e_\ell(r)$ with the *anticipated block end time* based on $s_\ell(r)$; that is, the end block time if no episode restart occurs within the block).

Then, our strategy will be to map out in time the *local bad segments* or subintervals of $[\tau_i, \tau_{i+1})$ where arm $a_r(B)$ incurs significant regret to arm $a_t^\sharp$ in bin $B$, roughly in the sense of ($\star$). The argument will conclude by arguing that a well-timed replay is scheduled to detect some local bad segment in $B$, before too many elapse.

As mentioned above, the difficulty here is that $a_r(B)$ is a random variable which depends on all the randomness up to time $e_\ell(r)$. However, conditional on just the block start time $s_\ell(r)$, we define the bad segments for a fixed arm $a$ and then argue that if too many bad segments w.r.t. $a$ elapse in the block, arm $a$ will be evicted in bin $B$. Crucially, this will hold uniformly over all arms $a$ and thus for arm $a = a_r(B)$, which bounds the regret of $a_r(B)$ in block $[s_\ell(r), e_\ell(r)]$.

**Notation.** *Going forward, we will drop the dependence on the bin $B$, level $r$, block $[s_\ell(r), e_\ell(r)]$, and episode $[t_\ell, t_{\ell+1})$ in certain definitions as they are fixed in the remainder of the analysis. We will let $a_i^\sharp(B)$ denote the last safe of bin $B$ in phase $[\tau_i, \tau_{i+1})$ (see Definition 8).*

**Definition 11.** *Fix an arm $a$ and $s_\ell(r)$, and let $[\tau_i, \tau_{i+1})$ be any phase intersecting $[s_\ell(r), e_\ell(r)]$. Define rounds $s_{i,0}(a), s_{i,1}(a), s_{i,2}(a) \ldots \in [t_\ell \vee \tau_i, \tau_{i+1})$ recursively as follows: let $s_{i,0}(a) \doteq t_\ell \vee \tau_i$ and define $s_{i,j}(a)$ as the smallest round in $(s_{i,j-1}(a), \tau_{i+1} \wedge e_\ell(r))$ such that arm $a$ satisfies for some fixed $c_{21} > 0$:*

$$\sum_{t=s_{i,j-1}(a)}^{s_{i,j}(a)} \delta_t(a_i^\sharp(B), a) \geq c_{21} \log(T) \cdot (s_{i,j}(a) - s_{i,j-1}(a))^{\frac{1+d}{2+d}} \cdot K^{\frac{1}{2+d}}. \tag{26}$$

*where $B' \supseteq B$ is the bin at level $r_{s_{i,j}(a)-s_{i,j-1}(a)}$, if such a round $s_{i,j}(a)$ exists. Otherwise, we let the $s_{i,j}(a) \doteq \tau_{i+1} - 1$. We refer to the interval $[s_{i,j-1}(a), s_{i,j}(a))$ as a **bad segment**. We call $[s_{i,j-1}(a), s_{i,j}(a))$ a **proper bad segment** if (26) above holds.*

It will in fact suffice to constrain our attention to proper bad segments, since non-proper bad segments $[s_{i,j-1}(a), s_{i,j}(a))$ (where $s_{i,j}(a) = \tau_{i+1} - 1$ and (26) is reversed) will be negligible in the regret analysis since there is at most one non-proper bad segment per phase $[\tau_i, \tau_{i+1})$ (i.e., the regret of such non-proper bad segments is at most (19)). In what follows, we let $B' \supseteq B$ be the bin at level $r_{s_{i,j}(a)-s_{i,j-1}(a)}$ where $[s_{i,j-1}(a), s_{i,j}(a))$ will be some proper bad segment, known from context.

**Lemma 13.** *Any proper bad segment is at least $K$ rounds long.*

*Proof.* We have

$$\begin{aligned}
n_{B'}([s_{i,j}(a), s_{i,j+1}(a)]) &\geq \sum_{s=s_{i,j}(a)}^{s_{i,j+1}(a)} \delta_t(a_i^\sharp(B), a) \cdot \mathbf{1}\{X_t \in B'\} \\
&\geq \sum_{s=s_{i,j}(a)}^{s_{i,j+1}(a)} \delta_t(a_i^\sharp(B), a) \cdot \mu(B') - c_2 \left( \log(T) + \sqrt{\log(T)(s_{i,j+1}(a) - s_{i,j}(a)) \cdot \mu(B')} \right) \\
&\geq c_{21} \log(T)(s_{i,j+1}(a) - s_{i,j}(a))^{\frac{1}{2+d}} \cdot K^{\frac{1+d}{2+d}} - c_2 \left( \log(T) + \sqrt{\log(T)(s_{i,j+1}(a) - s_{i,j}(a))\mu(B')} \right) \\
&\geq \sqrt{K \cdot n_{B'}([s_{i,j}(a), s_{i,j+1}(a)])},
\end{aligned}$$

where the last inequality follows from Lemma 10 and choosing $c_{21}$ large enough. $\square$

First, we relate our concentration bound (9) to (26), giving us control of the behavior of CMETA on proper bad segments. But, even before this, we establish an elementary lemma.

**Lemma 14.** *Let $[s_{i,j}(a), s_{i,j+1}(a))$ be a proper bad segment defined w.r.t. arm $a$. Let $m \in \mathbb{N}$ be such that $r_{s_{i,j+1}(a)-s_{i,j}(a)} = 2^{-m}$. Then, for some $c_{22} = c_{22}(d) > 0$ depending on the dimension $d$:*

$$\sum_{t=s_{i,j+1}(a)-K2^{(m-2)(2+d)-1}}^{s_{i,j+1}(a)} \delta_t(a_i^\sharp(B), a) \geq c_{22} \log(T) \cdot K^{\frac{1}{2+d}} \left(s_{i,j+1}(a) - s_{i,j}(a)\right)^{\frac{1+d}{2+d}}. \quad (27)$$

*Proof.* First, we may assume $s_{i,j+1}(a) - s_{i,j}(a) \geq 4 \cdot K$ by choosing $c_4$ in (26) large enough (this will make $m - 1$ sensible).

First, observe $K2^{(m-1)(2+d)} \leq s_{i,j+1}(a) - s_{i,j}(a) < K2^{m(2+d)}$. Let $\tilde{s} = s_{i,j+1}(a) - K2^{(m-2)(2+d)-1}$. Then, we have by (26) in the construction of the $s_{i,j}(a)$'s (Definition 11) that:

$$\sum_{t=\tilde{s}}^{s_{i,j+1}(a)} \delta_t(a_i^\sharp(B), a) = \sum_{t=s_{i,j}(a)}^{s_{i,j+1}(a)} \delta_t(a_i^\sharp(B), a) - \sum_{t=s_{i,j}(a)}^{\tilde{s}} \delta_t(a_i^\sharp(B), a)$$
$$\geq c_{21} \log(T) K^{\frac{1}{2+d}} \left((s_{i,j+1}(a) - s_{i,j}(a))^{\frac{1+d}{2+d}} - (\tilde{s} - s_{i,j}(a))^{\frac{1+d}{2+d}}\right)$$

Let $m_{i,j}(a) \doteq s_{i,j+1}(a) - s_{i,j}(a)$. Then, we have

$$m_{i,j}(a) \leq K2^{m(2+d)} \implies \tilde{s} - s_{i,j}(a) = m_{i,j}(a) - K2^{(m-2)(2+d)-1} \leq m_{i,j}(a)(1 - 2^{-2(2+d)-1}).$$

Plugging this into our earlier bound the constants become

$$1 - \left(1 - \frac{1}{2^{2(2+d)+1}}\right)^{\frac{1+d}{2+d}} > 0.$$

Note this last term is positive and only depends on $d$. $\qquad\square$

**Lemma 15** (Bin-Count Dominates Concentration Error on Bad Segment). *On event $\mathcal{E}_1$, letting $\tilde{s} = s_{i,j+1}(a) - K2^{(m-2)(2+d)-1}$, we have for bin $B' \supseteq B$ at level $r_{s_{i,j+1}(a)-\tilde{s}}$:*

$$n_{B'}([\tilde{s}, s_{i,j+1}(a)]) \geq 2c_1 \left(\log(T) + \sqrt{\log(T)(s_{i,j+1}(a) - \tilde{s})\mu(B')}\right).$$

*Proof.* Let $W = s_{i,j+1}(a) - \tilde{s}$. We first claim that $W \geq 2^{-2(2+d)} \cdot (s_{i,j+1}(a) - s_{i,j}(a))$. this follows from $s_{i,j+1}(a) - s_{i,j}(a) \leq K \cdot 2^{m(2+d)}$ and

$$s_{i,j+1}(a) - \tilde{s} = K \cdot 2^{(m-2)(2+d)-1} = 2^{-2(2+d)-1} \cdot (K \cdot 2^{m(2+d)}) \geq 2^{-2(2+d)-1} \cdot (s_{i,j+1}(a) - s_{i,j}(a)).$$

This will allow us to conflate $W$ and $s_{i,j+1}(a) - s_{i,j}(a)$ up to constants.

Since $\overline{\delta}_t^B(a_i^\sharp(B), a) \leq 1$, we have that (27) of the previous lemma and concentration (namely, (11) of Proposition 7; note that although $a_i^\sharp(B)$ is a random variable, it is a fixed and unchanging arm within $[\tau_i, \tau_{i+1})$ and hence $[\tilde{s}, s_{i,j}(a)])$ on $n_{B'}([\tilde{s}, s_{i,j+1}(a)])$ gives

$$n_{B'}([\tilde{s}, s_{i,j+1}(a)]) \geq \sum_{t=\tilde{s}}^{s_{i,j+1}(a)} \delta_t(a_i^\sharp(B), a) \cdot \mathbf{1}\{X_t \in B'\}$$
$$\geq c_4 \log(T) \cdot K^{\frac{1+d}{2+d}} \left(s_{i,j+1}(a) - s_{i,j}(a)\right)^{\frac{1}{2+d}}$$
$$\geq c_4 \left(\log(T) + \sqrt{\log(T) \cdot W \cdot (K/W)^{\frac{d}{2+d}}}\right)$$
$$\geq c_1 \left(\log(T) + \sqrt{\log(T)(s_{i,j+1}(a) - \tilde{s})\mu(B')}\right),$$

where the last inequality follows from the strong density assumption (Assumption 2). $\qquad\square$

Now, we define a well-timed or perfect replay which, if scheduled, will be able to detect the badness of arm $a$ in bin $B$ over a proper bad segment $[s_{i,j}(a), s_{i,j+1}(a))$.

**Definition 12** (Perfect Replay). *For a fixed proper bad segment $[s_{i,j}(a), s_{i,j+1}(a))$, define a perfect replay as a* Base-Alg $(t_{\text{start}}, m)$ *with* $t_{\text{start}} \in [s_{i,j+1}(a) - K2^{(m-2)(2+d)} + 1, s_{i,j+1}(a) - K2^{(m-2)(2+d)-1}]$ *and* $t_{\text{start}} + m \geq s_{i,j+1}(a)$.

The following proposition analyzes the behavior of a perfect replay and shows it will in fact evict arm $a$ from $\mathcal{A}(B)$ within a proper bad segment $[s_{i,j}(a), s_{i,j+1}(a))$.

**Proposition 16.** *Suppose event $\mathcal{E}_1$ holds. Let $[s_{i,j}(a), s_{i,j+1}(a))$ be a proper bad segment defined with respect to arm $a$. Let* Base-Alg $(t_{\text{start}}, m)$ *be a perfect replay as defined above which becomes active at $t_{\text{start}}$ (i.e., $Z_{t_{\text{start}}, m} = 1$). Fix an integer $m \geq s_{i,j+1}(a) - s_{i,j}(a)$. Then:*

  *(i)* Base-Alg $(t_{\text{start}}, m)$ **will not** *evict arm $a_i^\sharp(B)$ from $\mathcal{A}(B)$ before round $s_{i,j+1}(a) + 1$ while active.*

  *(ii) If $a \in \mathcal{A}_t$ for all rounds $t \in [\tilde{s}, s_{i,j+1}(a))$ where $X_t \in B$, where $\tilde{s} = s_{i,j+1}(a) - K2^{(m-2)(2+d)-1}$, then arm $a$ will be excluded from $\mathcal{A}(B)$ by round $s_{i,j+1}(a)$.*

*Proof.* Suppose event $\mathcal{E}_1$ (i.e., our concentration bound (9) holds). For (i), if $a_i^\sharp(B)$ is evicted over $[s_1, s_2] \subseteq [s_{i,j}(a), s_{i,j+1}(a)]$ from $\mathcal{A}(B')$ for bin $B' \supseteq B$ at level $r_{s_2 - s_1}$ by Line 11 of Algorithm 2, then $a_i^\sharp(B)$ incurs significant regret in bin $B'$ over $[s_1, s_2]$ (following same reasoning as in Lemma 11). This is a contradiction to the definition of the last safe arm $a_i^\sharp(B)$ (Definition 8). This shows (i).

For (ii), we first observe $\mathbb{E}[\hat{\delta}_t^B(a_i^\sharp(B), a) \mid \mathcal{F}_{t-1}] = \delta_t(a_i^\sharp(B), a)$ for any round $t \in [\tilde{s}, s_{i,j+1}(a)]$ such that $X_t \in B$ if $a_i^\sharp(B), a \in \mathcal{A}_t$. Let $B' \supseteq B$ be the bin at level $r_{s_{i,j+1}(a) - \tilde{s}}$.

Let $W = s_{i,j+1} - \tilde{s}$. Then, by Lemma 14, we have by smoothness that:

$$\sum_{t=\tilde{s}}^{s_{i,j+1}(a)} \delta_t(a_i^\sharp(B), a) \cdot \mathbf{1}\{X_t \in B'\} \geq c_4 \log(T) K^{\frac{1+d}{2+d}} \cdot W^{\frac{1}{2+d}} - n_{B'}([\tilde{s}, s_{i,j+1}(a)]) \cdot r(B')$$

Next, note that

$$\log(T)\sqrt{K \sum_{s=\tilde{s}}^{s_{i,j+1}} \mu_s(B')} + \log(T) \cdot r(B') \sum_{s=\tilde{s}}^{s_{i,j+1}} \mu_s(B'), \tag{28}$$

is bounded above by the same order.

Next, we bound (28) below by an empirical analogue. Applying concentration on $n_{B'}([\tilde{s}, s_{i,j+1}(a)])$ which dominates the Bernstein error by the previous lemma, the above is further lower bounded by

$$\log(T) \left( \sqrt{K \cdot n_{B'}([\tilde{s}, s_{i,j+1}(a)])} + r(B') \cdot n_{B'}([\tilde{s}, s_{i,j+1}(a)]) \right),$$

meaning arm $a$ will be evicted in $B'$ over $[\tilde{s}, s_{i,j+1}(a)]$.

Furthermore, within Base-Alg $(t_{\text{start}}, m)$'s play, arms $a$ and $a_i^\sharp(B)$ will **not** be evicted in any child of $B'$ before round $s_{i,j+1}(a)$ because such an eviction can only happen through a child base algorithm of Base-Alg $(t_{\text{start}}, m)$ which will necessarily use a level at least $r_W$. This is because of the way perfect replays are defined. By definition, the $

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

_r(B)), s_{i,j+1}(a_r(B))) : s_{i,j+1}(a_r(B)) < e_\ell(r)} (s_{i,j+1}(a_r(B)) - s_{i,j}(a_r(B)))^{\frac{1+d}{2+d}} \leq c_{23} \log(T)(e_\ell(r) - t_\ell)^{\frac{1+d}{2+d}} \tag{30}$$

Thus, by (26) in Definition 11, over the proper bad segments $[s_{i,j}(a_r(B)), s_{i,j+1}(a_r(B)))$ which elapse before the end of the block $e_\ell(r)$ in phase $[\tau_i, \tau_{i+1})$: the regret is at most

$$\sum_{t=s_\ell(r)}^{e_\ell(r)} \delta_t(a_t^\sharp, a_r(B)) \leq \sum_{(i,j)} \log(T) \cdot K^{\frac{1}{2+d}} m_{i,j}^{\frac{1+d}{2+d}}$$

$$\leq \log^2(T) \cdot K^{\frac{1}{2+d}} \cdot (e_\ell(r) - t_\ell)^{\frac{1+d}{2+d}}$$

Over each non-proper bad segment $[s_{i,j}(a_r(B)), s_{i,j-1}(a_r(B)))$ and the last segment $[s_{i,j}(a_r(B)), e_\ell(r)]$, the regret of playing arm $a_r(B)$ to $a_i^\sharp$ is at most $\log(T) \cdot r(B)^d \cdot K^{\frac{1}{2+d}} m_{i,j}^{\frac{1+d}{2+d}}$ by a similar series of calcuations and since there is at most one non-proper bad segment per phase $[\tau_i, \tau_{i+1})$ (see (26) in Definition 11).

So, we conclude that on event $\mathcal{E}_1 \cap \mathcal{E}_3(s_\ell(r))$:

$$\sum_{t=s_\ell(r)}^{e_\ell(r)} \delta_t(a_t^\sharp, a_r(B)) \leq 2c_{23} \log^2(T) \sum_{i \in \text{PHASES}(r,\ell)} K^{\frac{1}{2+d}} \cdot (\tau_{i+1} - \tau_i)^{\frac{1+d}{2+d}}.$$

Taking expectation (all expectations below are conditional on $\mathbf{X}_T$ and the good event $\mathcal{E}_2$ over which we have concentration of covariate counts), we have by conditioning first on $s_\ell(r)$ and then on event $\mathcal{E}_1 \cap \mathcal{E}_3(s_\ell(r))$:

$$\mathbb{E}\left[\sum_{t=s_\ell(r)}^{e_\ell(r)} \delta_t(a_t^\sharp, a_r(B))\right]$$

$$\leq \mathbb{E}_{s_\ell(r)}\left[\mathbb{E}\left[\mathbf{1}\{\mathcal{E}_1 \cap \mathcal{E}_3(s_\ell(r))\} \sum_{t=s_\ell(r)}^{e_\ell(r)} \delta_t(a_t^\sharp, a_r(B))\bigg| s_\ell(r)\right]\right] + T \cdot \mathbb{E}_{t_\ell}\left[\mathbb{E}\left[\mathbf{1}\{\mathcal{E}_1^c \cup \mathcal{E}_2^c(s_\ell(r))\}\bigg| s_\ell(r)\right]\right]$$

$$\leq 2c_{23} \log^2(T)\mathbb{E}_{s_\ell(r)}\left[\mathbb{E}\left[\mathbf{1}\{\mathcal{E}_1 \cap \mathcal{E}_3(t_\ell)\} \sum_{i \in \text{PHASES}(\ell,r)} K^{\frac{1}{2+d}}(\tau_{i+1} - \tau_i)^{\frac{1+d}{2+d}}\bigg| s_\ell(r)\right]\right] + \frac{K}{T^2}$$

$$\leq 2c_{23} \log^2(T)\mathbb{E}\left[\mathbf{1}\{\mathcal{E}_1\} \sum_{i \in \text{PHASES}(\ell,r)} (\tau_{i+1} - \tau_i)^{\frac{1+d}{2+d}} K^{\frac{1}{2+d}}\right] + \frac{1}{T},$$

where in the last step we bound $\mathbf{1}\{\mathcal{E}_1 \cap \mathcal{E}_3(t_\ell)\} \leq \mathbf{1}\{\mathcal{E}_1\}$ and apply tower law again. Plugging this into our earlier concentration bound on $\sum_{t=s_\ell(r)}^{e_\ell(r)} \delta_t(a_t^\sharp, a_r(B)) \cdot \mathbf{1}\{X_t \in B\}$, we conclude this part.

$\square$

# E  Proof of Corollary 5

The proof of Corollary 5 will follow in a similar fashion to the proof of Corollary 2 in Suk and Kpotufe [2022], which relates the total-variation rates to significant shifts in the non-stationary MAB setting. A novel difficulty here is that our notion of significant shift $\tau_i(\mathbf{X}_T), \tilde{L}(\mathbf{X}_T)$ (Definition 6) depends on the full context sequence $\mathbf{X}_T$, and so it is not clear how the (random) significant phases $[\tau_i(\mathbf{X}_T), \tau_{i+1}(\mathbf{X}_T))$ relate to the total-variation $V_T$, which is a deterministic quantity.

Our strategy will be to first convert the regret rate of Theorem 3 into one which depends on a weaker *worst-case notion of significant shift* which does not depend on the observed $\mathbf{X}_T$. Although this notion of shift is weaker, it will be easier to relate to the total-variation quantity $V_T$.

Let $\delta_t^a(x) := \max_{a' \in [K]} f_t^{a'}(x) - f_t^a(x)$ be the gap in mean rewards at the fixed context $x \in \mathcal{X}$.

**Definition 13** (worst-case sig shift). *Let $\tau_0 = 1$. Then, recursively for $i \geq 0$, the $(i+1)$-th **worst-case significant shift** is recorded at time $\tilde{\tau}_{i+1}$, which denotes the earliest time $\tilde{\tau} \in (\tilde{\tau}_i, T]$ such that there exists $x \in \mathcal{X}$ such that **for every arm** $a \in [K]$, there exists round $s \in [\tilde{\tau}_i, \tilde{\tau}]$, such that*

$$\delta_s^a(x) \geq \left(\frac{K}{t - \tilde{\tau}_i}\right)^{\frac{1}{2+d}}.$$

*We will refer to intervals* $[\tilde{\tau}_i, \tilde{\tau}_{i+1}), i \geq 0,$ *as* **worst-case (significant) phases**. *The unknown number of such phases (by time $T$) is denoted $\tilde{L}_{\text{pop}} + 1$, whereby $[\tilde{\tau}_{\tilde{L}_{\text{pop}}}, \tilde{\tau}_{\tilde{L}_{\text{pop}}+1})$, for $\tau_{\tilde{L}_{\text{pop}}+1} \doteq T + 1$, denotes the last phase.*

We next claim that

$$\mathbb{E}_{\mathbf{X}_T}\left[\sum_{i=0}^{\tilde{L}(\mathbf{X}_T)} (\tau_{i+1}(\mathbf{X}_T) - \tau_i(\mathbf{X}_T))^{\frac{1+d}{2+d}}\right] \leq c_{24} \sum_{i=0}^{\tilde{L}_{\text{pop}}} (\tilde{\tau}_{i+1} - \tilde{\tau}_i)^{\frac{1+d}{2+d}}.$$

This follows since the empirical significant phases $[\tau_i(\mathbf{X}_T), \tau_{i+1}(\mathbf{X}_T))$ interleave the population analogues $[\tilde{\tau}_i, \tilde{\tau}_{i+1})$ in the following sense: at each significant shift $\tau_{i+1}(\mathbf{X}_T)$, for each arm $a \in [K]$, there is around $s \in [\tau_i(\mathbf{X}_T), \tau_{i+1}(\mathbf{X}_T)]$ such that for $\delta_s(X_{\tau_{i+1}}) > \left(\frac{K}{\tau_{i+1} - \tau_i}\right)^{\frac{1}{2+d}}$. This means there must be a worst-case significant shift $\tilde{\tau}_j$ in the interval $[\tau_i(\mathbf{X}_T), \tau_{i+1}(\mathbf{X}_T)]$ since the criterion of Definition 13 is triggered at $x = X_{\tau_{i+1}}$. Thus, by the sub-additivity of the function $x \mapsto x^{\frac{1+d}{2+d}}$. This also allows us to conclude that each worst-case significant phase $[\tilde{\tau}_i, \tilde{\tau}_{i+1})$ can intersect at most two significant phases $[\tau_i(\mathbf{X}_T), \tau_{i+1}(\mathbf{X}_T))$.

Thus,

$$\sum_{i=0}^{\tilde{L}(\mathbf{X}_T)} (\tau_{i+1}(\mathbf{X}_T) - \tau_i(\mathbf{X}_T))^{\frac{1+d}{2+d}} \leq \sum_{i=0}^{\tilde{L}(\mathbf{X}_T)} \sum_{j: [\tilde{\tau}_j, \tilde{\tau}_{j+1}) \cap [\tau_i(\mathbf{X}_T), \tau_{i+1}(\mathbf{X}_T)) \neq \emptyset} |[\tilde{\tau}_j, \tilde{\tau}_{j+1}) \cap [\tau_i(\mathbf{X}_T), \tau_{i+1}(\mathbf{X}_T))|^{\frac{1+d}{2+d}}$$

$$\leq c_{24} \sum_{j=0}^{\tilde{L}_{\text{pop}}} (\tilde{\tau}_{j+1} - \tilde{\tau}_j)^{\frac{1+d}{2+d}},$$

where we use Jensen's inequality for $a^p + b^p \leq 2^{1-p}(a+b)^p$ for $p \in (0, 1)$ and $a, b \geq 0$ in the last step to re-combine the subintervals of each worst-case significant phase $[\tilde{\tau}_j, \tilde{\tau}_{j+1})$.

Then, it suffices to show

$$\sum_{j=0}^{\tilde{L}_{\text{pop}}} (\tilde{\tau}_{j+1} - \tilde{\tau}_j)^{\frac{1+d}{2+d}} K^{\frac{1}{2+d}} \lesssim T^{\frac{1+d}{2+d}} \cdot K^{\frac{1}{2+d}} + (V_T \cdot K)^{\frac{1}{3+d}} \cdot T^{\frac{2+d}{3+d}}. \tag{31}$$

We first transform the total variation into a more flexible quantity depending on the reward functions $f_t^a(\cdot)$ and the full sequence $\mathbf{X}_T$.

**Lemma 17.** *Let $G_t : \mathcal{X} \times [0,1]^K \to [-1,1]$ be any measurable function which takes the mean reward vector $f_t : \mathcal{X} \to [0,1]^K$ at round $t$ as input, and outputs a real number in $[-1,1]$. Then, recalling $\mathcal{D}_t$ is the joint distribution of $X_t$ and $Y_t$, we have for $t = 2, \ldots, T$:*

$$\|\mathcal{D}_t - \mathcal{D}_{t-1}\|_{\text{TV}} \geq \frac{1}{2} \left(G_t(f_t) - G_t(f_{t-1})\right).$$

*Proof.* This follows from the variational representation of the total variation distance [Polyanskiy and Wu, 2022, Theorem 7], which says for any measurable function $H : \mathcal{X} \times [0,1]^K \to [-1,1]$,

$$\|\mathcal{D}_t - \mathcal{D}_{t-1}\|_{\text{TV}} \geq \frac{1}{2} \left(\mathbb{E}_{(X_t, Y_t) \sim \mathcal{D}_t}[H(X_t, Y_t)] - \mathbb{E}_{(X_{t-1}, Y_{t-1}) \sim \mathcal{D}_{t-1}}[H(X_{t-1}, Y_{t-1})]\right). \tag{32}$$

In particular, we can take $H$ to only depend on the mean reward functions. $\square$

Now, fix a worst-case significant phase $[\tilde{\tau}_i, \tilde{\tau}_{i+1})$ such that $\tau_{i+1} < T + 1$. By Definition 13, there exists a context $x_i \in \mathcal{X}$ such that for arm $a_i \in \text{argmax}_{a \in [K]} f_{\tilde{\tau}_{i+1}}^a(x_i)$ we have there exists a round $t_i \in [\tau_i, \tau_{i+1}]$ such that:

$$\delta_{t_i}^{a_i}(x_i) > \left(\frac{K}{\tilde{\tau}_{i+1} - \tilde{\tau}_i}\right)^{\frac{1}{2+d}}.$$

On the other hand, $\delta^{a_i}_{\tilde{\tau}_{i+1}}(x_i) = 0$ by the definition of arm $a_i$ being the best at $x_i$ at round $\tilde{\tau}_{i+1}$. Thus,

$$\left(\frac{K}{\tilde{\tau}_{i+1} - \tilde{\tau}_i}\right)^{\frac{1}{2+d}} < \delta^{a_i}_{t_i}(x_i) - \delta^{a_i}_{\tilde{\tau}_{i+1}}(x_i) = \sum_{t=t_i+1}^{\tau_{i+1}} \delta_t(a_i, x_i) - \delta_{t-1}(a_i, x_i).$$

For each round $t = 2, \ldots, T$, let $G_t(f_t) := \delta_t(a_i, x_i)$, where $x_i$ is the associated context of the unique worst-case significant shift $\tilde{\tau}_{i+1}$ such that $t \in [\tilde{\tau}_i, \tilde{\tau}_{i+1})$ and where $a_i$ is defined as above. Then, $G_t$ only depends on the mean reward function $f_t : \mathcal{X} \to [0,1]^K$ at round $t$ and *not* on the observed contexts $\mathbf{X}_T$. Then, since $G_t(\cdot)$ satisfies the condition of Lemma 17, we must have

$$\sum_{i=1}^{\tilde{L}_{\text{pop}}} \left(\frac{K}{\tilde{\tau}_{i+1} - \tilde{\tau}_i}\right)^{\frac{1}{2+d}} < \sum_{t=2}^{T} G_t(f_t) - G_{t-1}(f_{t-1}) \leq \sum_{t=2}^{T} \|\mathcal{D}_t - \mathcal{D}_{t-1}\|_{\text{TV}}. \tag{33}$$

Now, by Hölder's inequality for $p \in (0,1)$ and $q \in \left(0, \frac{1+d}{2+d}\right)$:

$$\sum_{i=1}^{\tilde{L}_{\text{pop}}} (\tilde{\tau}_{i+1} - \tilde{\tau}_i)^{\frac{1+d}{2+d}} K^{\frac{1}{2+d}} \leq T^{\frac{1+d}{2+d}} K^{\frac{1}{2+d}} + \left(\sum_i K^{\frac{1}{2+d}}(\tilde{\tau}_{i+1} - \tilde{\tau}_i)^{-q/p}\right)^p \left(\sum_i K^{\frac{1}{2+d}}(\tilde{\tau}_{i+1} - \tilde{\tau}_i)^{\left(\frac{1+d}{2+d}+q\right) \cdot \frac{1}{1-p}}\right)^{1-p}.$$

In particular, letting $p = \frac{1}{3+d}$ and $q = \frac{1}{(2+d)(3+d)}$ and plugging in our earlier bound (33) makes the above RHS

$$V_T^{\frac{1}{3+d}} \cdot K^{\frac{1}{3+d}} \cdot T^{\frac{2+d}{3+d}}.$$

$\square$

# F    Proof of Theorem 1

We first note that it suffices to show (3) for integer $L \in [0, T] \cap \mathbb{N}$ as lower bounds for all other $L$ follow via approximation and modifying the constant $c > 0$ in (3). Thus, going forward, fix $V \in [0, T]$ and $L \in \mathbb{Z} \cap [0, T]$.

At a high level, our construction will repeat $L + 1$ a hard environment for stationary contextual bandits. In particular, within each stationary phase of length $T/(L+1)$ one is forced to pay a regret of $\left(\frac{T}{L+1}\right)^{\frac{1+d}{2+d}}$, summing to a total regret lower bound of $(L+1) \cdot \left(\frac{T}{L+1}\right)^{\frac{1+d}{2+d}} \approx L^{\frac{1}{2+d}} \cdot T^{\frac{1+d}{2+d}}$.

To get the rate in terms of $V$ in (3), we will choose $L \propto V^{\frac{2+d}{3+d}} \cdot T^{\frac{1}{3+d}}$ appropriately and argue that the total-variation $V_T$ is less than $V$, so that our constructed environment indeed lies in the family $\mathcal{P}(V, L, T)$. This is similar to the arguments of the analogous lower bound [Besbes et al., 2019, Theorem 1] for the non-contextual non-stationary bandit problem.

We start by establishing a lower bound for stationary Lipschitz context bandits. The construction is identical to that of Rigollet and Zeevi [2010, Theorem 4.1]. We only highlight a minor novelty in circumventing the reliance of the cited result on a positive "margin parameter" $\alpha > 0$.

**Proposition 18.** *Suppose there are $K = 2$ arms. Then, there exists a stationary Lipschitz contextual bandit environment $\mathcal{E}(n)$ over $n$ rounds such that for any algorithm $\pi$ taking as input random variable $U$, independent of $\mathcal{E}(n)$, we have for some constant $c > 0$:*

$$\mathbb{E}_{\mathcal{E}(n),U}[R(\pi, \mathbf{X}_T)] \geq c \cdot n^{\frac{1+d}{2+d}}.$$

*Proof.* Let the covariates $X_t$ be uniformly distributed on $[0,1]^d$ at each round $t \in [n]$, so that $\mu_X \equiv \text{Unif}\{[0,1]^d\}$. For ease of presentation, let us reparametrize the two arms as $+1$ and $-1$.

At each round $t \in [n]$, let arm $-1$ have reward $Y_t^{-1} \sim \text{Ber}(1/2)$ and let arm $1$ have reward $Y_t^1 \sim \text{Ber}(f(X_t))$ where $f : \mathcal{X} \to [0,1]$ is some function to be defined. Let

$$M := \left\lceil \left(\frac{n}{8e}\right)^{\frac{1}{2+d}} \right\rceil.$$

We next partition $\mathcal{X} = [0,1]^d$ into a regular grid $\mathcal{Q} = \{q_1, \ldots, q_{M^d}\}$, where $q_k$ denotes the center of bin $B_k$, $k = 1, \ldots, M^d$. Specifically, for each index $\mathbf{k} = (k_1, \ldots, k_d) \in \{1, \ldots, M\}^d$, we define the bin $B_k$ as:

$$B_k = \left\{ x \in \mathcal{X} : \frac{k_\ell - 1}{M} \leq x_\ell \leq \frac{k_\ell}{M}, \ell = 1, \ldots, d \right\}.$$

Define $C_\phi \doteq 1/4$. Then, let $\phi : \mathbb{R}^d \to \mathbb{R}_+$ be a smooth function defined by:

$$\phi(x) = \begin{cases} 1 - \|x\|_\infty & 0 \leq \|x\|_\infty \leq 1 \\ 0 & \|x\|_\infty > 1 \end{cases}.$$

It's straightforward to verify $\phi$ is 1-Lipschitz over $\mathbb{R}^d$.

Now, define the integer $m = \lceil \mu \cdot M^d \rceil$ where $\mu \in (0,1)$ is chosen small enough to ensure $m \leq M^d$.

Define $\Sigma_m = \{-1, 1\}^m$ and for any $\omega \in \Omega_m$, define the function $f_\omega$ on $[0,1]^d$ via

$$f_\omega(x) = 1/2 + \sum_{j=1}^m \omega_j \cdot \phi_j(x),$$

where $\phi_j(x) \doteq M^{-1} \cdot C_\phi \cdot \phi(M \cdot (x - q_j)) \cdot \mathbf{1}\{x \in B_j\}$. Then, the optimal arm at context $x \in \mathcal{X}$ in this environment is given by $\pi_f^*(x) \doteq \mathrm{sgn}(f(x) - 1/2)$.

Then, define the family $\mathcal{C}$ of environments induced by $f_\omega$ for $\omega \in \Omega_m$. Next, let $\mathrm{Int}(B_k)$ be the $\ell_\infty$ ball centered at $q_k$ of radius $\frac{1}{2M}$. Then, we have for any $x \in \mathrm{Int}(B_k)$,

$$|f_\omega(x) - 1/2| \geq M^{-1} \cdot C_\phi/2.$$

Then, the worst-case regret over the family of environments in $\mathcal{C}$ is at least

$$\sup_{f \in \mathcal{C}} \mathbb{E} \sum_{t=1}^n |f^{(1)}(X_t) - f^{(2)}(X_t)| \cdot \mathbf{1}\{\pi_t(X_t) \neq \pi^*(X_t)\}$$

$$\geq \frac{C_\phi}{2M} \sup_{f \in \mathcal{C}} \mathbb{E} \sum_{t=1}^n \sum_{j=1}^m \mathbf{1}\{\pi_t(X_t) \neq \pi^*(X_t), X_t \in \mathrm{Int}(B_j)\}.$$

Lower bounding the remaining supremum on the above RHS display by $\Omega(n)$ follows the same steps as the proof of Theorem 4.1 in Rigollet and Zeevi [2010]. In particular, the algorithm $\pi$ may depend on additional randomness $U$, independent of the environment, which is ignorable in the KL calculations by use of chain rule. Plugging in the earlier choice of $M$ this makes the above RHS at least $\Omega(n^{\frac{1+d}{2+d}})$.

$\square$

Given Proposition 18, the $(L+1) \cdot \left( \frac{T}{L+1} \right)^{\frac{1+d}{2+d}}$ lower bound immediately follows by constructing a random environment which consists of $L + 1$ independent repetitions of the stationary environment $\mathcal{E}(T/(L+1))$. Any such constructed environment clearly has at most $L$ global shifts. Note that the regret over a given stationary phase of length $\frac{T}{L+1}$ is lower bounded by $\left( \frac{T}{L+1} \right)^{\frac{1+d}{2+d}}$ regardless of the information learned prior to that phase, as such information can be formalized as exogenous randomness $U$ in Proposition 18 w.r.t. the fixed stationary phase.

Next, we tackle the lower bound $V^{\frac{1}{3+d}} \cdot T^{\frac{2+d}{3+d}}$ in terms of total-variation budget $V$. First, if $V < \left( \frac{1}{T} \right)^{\frac{3+d}{2+d}}$, then we're already done as

$$\left( T^{\frac{1+d}{2+d}} + T^{\frac{2+d}{3+d}} \cdot V^{\frac{1}{3+d}} \right) \wedge \left( (L+1)^{\frac{1}{2+d}} T^{\frac{1+d}{2+d}} \right)$$

is minimized by the first term which is of order $T^{\frac{1+d}{2+d}}$. Thus, using Proposition 18 with a single stationary phase $\mathcal{E}(T)$ gives lower bound of the right order. Such an environment clearly has total-variation $V_T = 0 \leq V$.

Let $\Delta \doteq \left\lceil \left(\frac{T}{V}\right)^{\frac{2+d}{3+d}} \right\rceil \leq \left\lceil T^{\frac{1}{3+d}} \right\rceil$ and consider $L+1 = T/\Delta$ stationary phases of length $\Delta$. Then, by the previous arguments we have the regret is lower bounded by

$$(L+1)^{\frac{1}{2+d}} \cdot T^{\frac{1+d}{2+d}} = \frac{T}{\Delta^{\frac{1}{2+d}}} \geq \frac{T}{2^{\frac{1}{3+d}} (T/V)^{\frac{1}{3+d}}} \propto T^{\frac{2+d}{3+d}} \cdot V^{\frac{1+d}{3+d}}.$$

Additionally, $T^{\frac{2+d}{3+d}} \cdot V^{\frac{1+d}{3+d}}$ dominates $T^{\frac{1+d}{2+d}}$ since $V \geq \left(\frac{1}{T}\right)^{\frac{3+d}{2+d}}$. Thus, the regret lower bound is proven in terms of $V$.

It remains to show the total-variation $V_T$ is at most $V$ in the above constructed environments so that it lies in the family $\mathcal{P}(V, L, T)$.

Clearly, the instantaneous total-variation $\|\mathcal{D}_t - \mathcal{D}_{t-1}\|_{\text{TV}} = 0$ for all rounds $t$ not being the start of a new stationary phase. On the other hand, for such a round $t$, we have that since conditioning increases the TV [Polyanskiy and Wu, 2022, Theorem 7.5(c)], the instantaneous TV is at most:

$$\|\mathcal{D}_t - \mathcal{D}_{t-1}\|_{\text{TV}} \leq \mathbb{E}_{x \sim \mu_X} \left[ \|\mathcal{D}_t(Y_t|X_t = x) - \mathcal{D}_{t-1}(Y_{t-1}|X_{t-1} = x)\|_{\text{TV}} \right].$$

Since $Y_t^a|X_t = x \sim \text{Ber}(f_t^a(x))$, we have the RHS' inner TV quantity is just the total variation between Bernoulli's or $\max_{a \in [2]} |f_t^a(x) - f_{t-1}^a(x)|$. Carefully analyzing the variations int he constructed Lipechitz reward functions in the proof of Proposition 18 reveals this TV between Bernoulli's is at most $\frac{e^{\frac{1}{2+d}}}{8^{\frac{1+d}{2+d}}} \cdot \left(\frac{L+1}{T}\right)^{\frac{1}{2+d}}$ (note the attached constant is $< 1$ for all $d \in \mathbb{N} \cup \{0\}$).

Summing over phases, we have

$$V_T \leq (L+1)^{\frac{3+d}{2+d}} \cdot T^{-\frac{1}{2+d}} = T \cdot \left(\frac{1}{\Delta}\right)^{\frac{3+d}{2+d}} = V.$$

$\square$