# OpenReview forum: "Tracking Most Significant Shifts in Nonparametric Contextual Bandits"
_NeurIPS.cc/2023/Conference — NeurIPS 2023 poster_

### Official Review · Reviewer_aF2K · 2023-07-04

**Soundness:** 3 good
**Presentation:** 3 good
**Contribution:** 4 excellent
**Rating:** 7
**Confidence:** 4

**Summary:**

This paper studies nonparametric contextual bandit problems with distributional shifts. This paper proposes a new notion of distributional changes called the experienced significant shifts. Based on this notion, the authors develop new algorithms that achieve the minimax rates without knowing some problem-dependent parameters.

====After rebuttal====
I have read the rebuttal. I'd like to keep my scores. I also encourage the authors to add more discussions regarding the adaptivity issue.

**Strengths:**

The authors developed a new notion called the experience significant shifts that better capture the distribution shifts in contextual bandits. Based on this new notion, the authors show that the minimax optimal regret in contextual bandits with distributional shifts can be achieved, even without knowing problem-dependent parameters. In my opinion, this result is significant to the community. Also, along the way, the authors develop several new techniques to achieve this result (e.g., those have been highlighted in Section 5), which can be of independent interest.

**Weaknesses:**

In the paper, the authors mention several generalizations of the existing setting, e.g., (i) generalization to H\"older continuity, and (ii) the one mentioned in Remark 1. However, no formal results are provided for these generalizations. It will be great if the authors can provide some formal statements.

Also, for the oracle procedure described in Definition 7, it's better formally state the power of the oracle, e.g., what is known to the oracle and what is unknown.

**Questions:**

In bandit learning, when the goal is to minimize the cumulative regret, researchers have previously shown that adaptivity to the usual minimax rate is usually impossible and the best one can hope for is the Pareto optimality, e.g., for the model selection problem [1, 2]. However, in this paper, the authors show the opposite in this paper for learning with distributional shifts. Can authors elaborate more on this? Is this because of the common assumptions made in this setting, e.g., Assumption 2?

Citations:

[1] Teodor Marinov and Julian Zimmert. The Pareto frontier of model selection for general contextual bandits.
[2] Yinglun Zhu and Robert Nowak. Pareto optimal model selection in linear bandits.

---

> ### Author Rebuttal · Authors · 2023-08-08
>
> Thanks for supportive comments!
>
> _Generalizations to Hölder continuity and Remark 1_: We apologize for the vagueness; we'll add explicit statements showcasing such statements.
>
> _On Adaptivity_: The cited papers in fact study other notions of adaptivity, which are not applicable to to our problem for the following reasons. [1] considers a stronger notion of adaptivity than the one considered here: the impossibility of adaptive switching regret against an adaptive adversary, which is a harder problem than our task of dynamic regret minimization with obliviously decided rewards. [2] is concerned with adaptivity to the unknown intrinsic dimension of linear representations of arms; here we focus on optimal regret in terms of the context dimension which is known (indeed, by Assumption 2, as you point out).
>
> Also, note that adaptivity to the minimax rate for an unknown number of distribution shifts is recently shown to be possible in many bandit and RL settings (see Auer et al., 2019; Chen et al., 2019; Wei \& Luo, 2021). So, it is a not a new phenomenon. However, we find your comment very insightful and will add discussion as such.

---

### Official Review · Reviewer_GBqh · 2023-07-10

**Soundness:** 3 good
**Presentation:** 4 excellent
**Contribution:** 4 excellent
**Rating:** 8
**Confidence:** 3

**Summary:**

This paper studies the contextual bandits problem with changing Lipschitz reward function and proves a *minimax optimal* regret bounds for this problem, which includes both upper and lower bounds. For the upper bound, this paper comes up with an algorithm that achieves it. The algorithm is based on carefully maintaining a hierarchical partition tree that discretizes the context space.

**Strengths:**

1. This paper achieves minimax optimality for the problem, which closes the existing gap and solves the problem.
2. The novel idea of significant shifts and the algorithm design of maintaining the partition tree, could be of independent interest.
3. This paper provides excellent plain word explanation to highlight the key ideas and steps in their proof.

**Weaknesses:**

1. The algorithm is recursive and complicated, so it makes the audience hard to understand the algorithm, even with the algorithm explanation in Sec. 4. Maybe it is better to replace Line 9 in Algorithm 2 with some while-loop to improve readability.

**Questions:**

Typos:
1. Line 178, "Miminimax".
2. Below Line 259, Algorithm 1, "tree $T$" -> tree $\mathcal T$.
3. Line 305, "ut".
4. Line 306, "(i.e., the bin at level $r_{s_2-s_1}$ containing $X_t$)" duplicates.
4. Line 13 in Algorithm 2, trailing ; after :
The authors may spend some time to polish the paper before finalizing.

---

> ### Author Rebuttal · Authors · 2023-08-08
>
> Thanks for encouraging comments as well as pointing out typos and writing suggestions!

---

> > ### Comment · Reviewer_GBqh · 2023-08-17
> >
> > I have read the rebuttal.

---

### Official Review · Reviewer_o8kG · 2023-07-11

**Soundness:** 3 good
**Presentation:** 3 good
**Contribution:** 3 good
**Rating:** 7
**Confidence:** 3

**Summary:**

This paper studies nonstationary contextual bandits. In particular, a new notion of "significant shift" is introduced (Definition 6) which accounts for shifts in the distribution localized at the possible actions and with significant magnitude. First, the authors derive a lower bound for existing definitions of shifts (Theorem 1) and provide an oracle algorithm to achieve it. Then, after introducing the new notion of shifts, they derive an algorithm which adapts to it (i.e., which does not require the time indexes of the shifts), see Theorem 3.

**Strengths:**

- The topic is of interest to the NeurIPS community
- The paper is globally clear and well written
- The intuition and analysis are sound
- Authors constantly compares their results with existing ones to position their contribution

**Weaknesses:**

- The paper lacks experiments. Since the authors provided a detailed algorithm, I find it disappointing that no experiment is carried out. It would be of particular interest to study the behavior of the different algorithm depending on the types of switches
- I feel previous paper on significant switches could be discussed a bit more, to highlight the changes and challenges raised by the contextual setting
- No lower bound for the proposed definition of switches is provided


Minor:
- Abstract: MAB is not defined yet
- Section 1.1: linebreaks do not seem necessary to me and make the reading less fluid
- When citing several works, the chronological order is preferable
- Equation ($\star$): what is $r(B)$?
- Lines 242, 252: are the log factors omitted?
- l. 305: ut
- l. 316: $\approx$ could be avoided

**Questions:**

- In this paper, the available actions are the same at every time steps. Could the authors think about a generalization when the action set changes over time? In particular, would it be possible to restrict the significant changes only to the playable arms?
- Could the Lipschitz assumption be removed?
- In Definition 7: does $\cal{G}_t$ always exist?
- Do authors have in mind simple examples of $f_a^t$ where both switches characterization drastically differ?
- If I'm not missing anything, it seems to me that no definition of switch implies the other. Then, I find it a bit misleading to compare results all along. It should be made clear that the two are two different parameterizations incomparable in general (I agree on the identity $\tilde{L} \le L$)
- Assume that changes have small magnitude (e.g., small drift at each time step) or do not apply to every arms (but only to the best let say), is the regret of CMETA not impacted by those switches?

---

> ### Author Rebuttal · Authors · 2023-08-08
>
> Thanks for pointing out typos, writing suggestions, and many careful questions.
>
> **Weaknesses**:
>
> _Experiments_: We'd like to emphasize the main contribution of the paper is theoretical, rather than proposing a new algorithm. In fact, the algorithm in the paper is of a theoretical nature that only serves to drive the main theoretical message: for the well-studied setting of contextual bandits with Lipschitz rewards, both the form of the optimal regret and whether optimality can be adaptively achieved have remained open; we have resolved these questions.
>
>  We agree with you, however, that an eventual goal of this fledgling line of work is to develop practical procedures, and we admit the state-of-the-art is still far from this.
>
> _Lower Bound in terms of $\tilde{L}$_: The construction of the lower bound (Theorem 1) in terms of $L$ global shifts and total-variation $V_T$ in fact satisfies $\tilde{L}=\Omega(L)$. Thus, this implies a matching lower bound $T^{\frac{1+d}{2+d}}\cdot \tilde{L}^{\frac{1}{2+d}}$ of the same order as our upper bound (Theorem 3), up to log factors. We'll make this more clear.
>
> **Questions**:
>
> _Changing Action Sets_: This is an interesting future direction which is beyond the scope of our paper. Even in the simpler MAB setting, such results are unknown as there are added difficulties with changing action sets. For instance, with changing action sets, the _safe arm_ (i.e., the arm not yet incurring significant regret within a phase) and a bad arm with large regret may not be available on the same rounds, a fact crucial to the significant shift analysis of Suk and Kpotufe, 2022. This makes it unclear how to generalize key parts of the regret analysis.
>
> _Removing Lipschitz Assumptions_: We can generalize all the results to the setting of  $\alpha$-Hölder continuous rewards with $\alpha\leq 1$. We'll include a remark to this.
>
> _Existence of Good Arm Set_ $\mathcal{G}_t$: Yes, $\mathcal{G}_t$ always exists and contains at least one arm by the definition of experienced significant shift (Definition 6).
>
> _Instances where Switch Characterizations Drastically Differ_: As a simple example, if there were no changes in best arm at any context $x$ but changes in rewards at every round (e.g., the rewards of all arms change together by the same amount) then we'd have large total-variation $V_T=T$ and global count of shifts $L=T$ versus $\tilde{L} = 0$ experienced sig. shifts. Going even beyond this, even a tighter global count of best-arm changes $S := \sum_{t=2}^T {\bf 1}(\exists x \in \mathcal{X}: \text{best arm changes at $x$ from $t-1$ to $t$})$ could still be large $S=T$ while $\tilde{L}  \ll T$ if the changes in best arm were constrained to a small subregion of context space.
>
> _Comparability of Definitions of Switches_: In fact, the lower and upper bound results **are comparable**.  Our main adaptive upper bound always achieves the minimax optimal rate under all the parametrizations mentioned in the paper: $L \geq \tilde{L}$ (Corollary 4) and $V_T$ (Corollary 5).
>
> _Small Magnitude Changes and Regret of CMETA_: You're correct in that small enough magnitude changes do not affect the regret of CMETA. Additionally, changes constrained to subsets of arms which do not change the best arm do not affect the regret of CMETA.

---

### Official Review · Reviewer_PUca · 2023-07-13

**Soundness:** 3 good
**Presentation:** 2 fair
**Contribution:** 3 good
**Rating:** 4
**Confidence:** 3

**Summary:**

This paper studies nonparametric contextual bandits where the mean reward functions can change over time. A key assumption is that the rewards are Lipchitz in context. The authors then adopt a typical approach to discretize the context space into bins. The notions “significant regret”, “unsafe at context”, and “experienced significant shift,”; in particular, an experienced significant shift implies a change in the optimal arm in a particular bin. The authors then propose an algorithm Contextual Meta-Elimination while Tracking (CMETA) and establish regret bounds in terms of the total number of “experienced significant shifts.”

**Strengths:**

This paper is well-organized and introduces both a notion of “experienced significant shift” and an algorithm, CMETA, accompanied with theoretical guarantees.

**Weaknesses:**

My major concerns centers around the comparison of CMETA and its analysis to the algorithm and regret analysis introduced by Suk and  Kpotufe (2022), and the presentation of the theoretical results.
1. Besides discretization of the context space into bins, how does CMETA and its analysis differ from that introduced by Suk and  Kpotufe (2022)?
2. As for the presentation of the results, we take Theorem 3 as an example. First, the notation is inconsistent: the notation $\tilde{L}$ is introduced in line 214, and in Theorem 3 it is appeared as $\tilde{L}$ in line 248 but $\tilde{L}(\mathbb{X}_T)$ in line 250. In addition, $\tilde{L}$ is dependent on the discretization of context space into bins, which is determined by level $r$ in Algorithm 1, yet $r$ does not appear in Theorem 3.1? Moreover, could the authors elaborate on “choice of level” in Section 4 on how the level is adaptively chosen?


**Questions:**

My main concerns were raised in the “Weakness” section. To reiterate:
1. Besides discretization of the context space into bins, how does CMETA and its analysis differ from that introduced by Suk and  Kpotufe (2022)?
2. It would be very helpful if the authors could elaborate on the “choice of level” in Algorithm 1 (CMETA) and how the regret bound established in Theorem 3.1 depends on the level.

A remaining question is:
-  In Corollary 4, it seems that if $\tilde{L}$ grows linearly in $T$, the bound on cumulative regret becomes linear in $\log^3(T) T$, in bandits where rewards are bounded in $[0, 1]$ (and cumulative regret is at most $T$)?

---

> ### Author Rebuttal · Authors · 2023-08-08
>
> _Key Difficulties_: Although we agree with the reviewer that discretization is a natural approach appearing in fact in all past works on Lipschitz contextual bandits, this is not the main technical difficulty, neither algorithmically nor analytically: the main difficulty is to understand the level of discretization required for the specific setting, i.e., what cell width to employ in different parts of space (e.g., Rigollet & Zeevi, 2010; Perchet & Rigollet, 2013), and how to automatically infer such cell-width from data (Slivkins, 2014). This is in fact the focus of papers on the subject, and is by now well-understood in the stationary setting to tightly depend on the time horizon. In the non-stationary setting considered here, however, such automatic choice of discretization is now even more difficult: we need to not only make a choice of level in stationary phases, but also, in order to detect stationary phases of varying length, we need separate choices of levels commensurate with the unknown starts of new phases. The main challenge therefore is in understanding how to design and schedule such automatic choices of levels while maintaining optimal performance w.r.t. the unknown number and positions of stationary phases. This is explained lines 318-341 of Section 5.
>
> _Notation and Choice of Level_: We apologize for confusing notation; we drop the dependence on ${\bf X}_T$ in $\tilde{L}$ in some places for ease of presentation (see Lines 211--212).
>
> Note that the notion $\tilde{L}$ of experienced sig. shift (Definition 6) is independent of any fixed level (and thus of the level used by the algorithm), which is part of its appeal. This misunderstanding is our fault as we often discuss $\tilde{L}$ in terms of levels used by the algorithm. As said above, such choices of level are the main technical apport of our analysis.
>
> The "choice of level'' is adaptive in the sense that it does not depend on a fixed horizon $T$ and so can "adapt'' to the minimax regret over unknown episode durations (the earlier Perchet & Rigollet, 2013 employ such a time-varying level in the stationary setup).
>
> _On $\log$ Factors and Sublinear Regret_: It has unfortunately become common in bandits to write regret bounds this way. We apologize for the confusion this may have caused. Our regret is always upper bounded by $T$ and not by $\log^3(T) \cdot T$ in the worst case, and we'll add an indicator to our bound to express when there's $\log$ factors.

---

### Author Rebuttal · Authors · 2023-08-10

We thank reviewers for their time and useful comments. Please see individual rebuttals to each reviewer.

---

### Decision · Program_Chairs · 2023-09-21

**Decision:**

Accept (poster)

**Comment:**

The authors consider a non-parametric contextual bandit problem under non-stationarity conditions. While non-parametric contextual bandits is well-understood in the stationary setting, and non-stationarity in a variety of settings have come to be well-understood only recently (e.g., Wei and Luo, Suk and Kpotufe), the intersection has not been explored. As some reviewers suggest, the trope of combining two well-known concepts and calling it novel is not very inspiring. However, the authors make a number of novel contributions, including the experienced significant shift which may be useful elsewhere, and clearly describe that the analysis of the adaptive gridding common to non-parametric stationary settings must be completely rethought in the context of this non-stationary setting.